# THE INVISIBLE LEASH?
# WHY RLVR MAY OR MAY NOT ESCAPE ITS ORIGIN

## ABSTRACT

Recent advances in large reasoning models highlight Reinforcement Learning with Verifiable Rewards (RLVR) as a promising method for enhancing AI's capabilities, particularly in solving complex logical tasks. However, it remains unclear whether the current practice of RLVR truly expands a model's reasoning boundary or mainly amplifies high-reward outputs that the base model already knows for improved precision. This study presents an empirical investigation that provides fresh insights into the potential limits of the common practice of RLVR. We examine how, under current training conditions, RLVR can operate as a support-constrained optimization mechanism that may restrict the discovery of entirely original solutions, remaining constrained by the base model's initial distribution. We also identify an entropy–reward trade-off: while the current RLVR recipe reliably enhances precision, it may progressively narrow exploration and potentially overlook correct yet underrepresented solutions. Extensive empirical experiments validate that while the current RLVR recipe consistently improves `pass@1`, *the shrinkage of empirical support generally outweighs the expansion of empirical support under larger sampling budgets*, failing to recover correct answers that were previously accessible to the base model. Interestingly, we also observe that while RLVR sometimes increases token-level entropy, it results in greater uncertainty at each generation step and declining answer-level entropy. This indicates that these seemingly more uncertain paths ultimately converge onto a smaller set of distinct answers. Taken together, we reveal potential limits of the current RLVR recipe in extending reasoning horizons. Breaking this invisible leash may require future algorithmic innovations such as explicit exploration mechanisms or hybrid strategies that seed probability mass into underrepresented solution regions.

## 1 INTRODUCTION

The rise of large reasoning models, such as DeepSeek-R1 (Guo et al., 2025) and OpenAI-o1 (Jaech et al., 2024), marks a breakthrough in AI capabilities, particularly in solving complex logical tasks involving mathematics (Luo et al., 2025c; Zeng et al., 2025) and programming (Luo et al., 2025b; Liu & Zhang, 2025). The key ingredient behind this remarkable progress is large-scale **Reinforcement Learning with Verifiable Rewards** (**RLVR**), where a pretrained base model or one fine-tuned on Chain-of-Thought data is optimized via RL using simple, automatically computed rewards. Prior work has also explored stronger notions of verifiability, such as models that can generate interactive proofs of their own correctness (Amit et al., 2025). Despite empirical success, a fundamental question remains under active debate within the research community: *Does the current practice of RLVR expand base models' reasoning capabilities, or simply reinforce patterns base models already knows, sometimes at the expense of exploring alternative correct solutions?*

Recent studies have revealed a puzzling pattern that hints at this limitation. While models trained with the common RLVR recipe consistently outperform base models when evaluated with a single attempt (`pass@1`), base models often perform better given multiple attempts. It is even reported that RLVR benefits from seemingly random or spurious reward signals, raising questions about whether observed improvements genuinely reflect enhanced reasoning (Shao et al., 2025). While `pass@k` may have limitations as a comprehensive measure of reasoning boundaries, as it primarily captures solution retrieval rather than novel reasoning capacity (Wen et al., 2025), we adopt it here as a practical proxy metric following Chen et al. (2021); Shao et al. (2024); Chen et al. (2025b). This

metric provides a useful lens for examining how RLVR affects solution accessibility, though future work should explore more nuanced measures of reasoning capability expansion. Besides, prior work examines RLVR only through before/after snapshots, leaving unexplored how RLVR reshapes the model's effective reasoning support throughout training.

Some studies interpret pass@k as evidence that the current RLVR recipe primarily performs conservative optimization within the base model's existing capabilities (Yue et al., 2025a; Zhao et al., 2025; Shah et al., 2025; Ma et al., 2025; He et al., 2025). Others argue that this pattern only appears in specialized domains where base models were already well-trained, and that RLVR can substantially expand reasoning in other domains (Liu et al., 2025).

Seeking a definitive answer to this debate remains an open challenge. In the extreme case, it seems unlikely that the current RLVR recipe can unlock advanced reasoning capabilities for any model out of the box, such as GPT-2 (Radford et al., 2019). We are curious if there may exist inherent limitations in the current RLVR practice. This paper provides a systematic empirical investigation into the fundamental capabilities and potential limitations of the current RLVR practice. We introduce the concept of *empirical support*: the set of correct solutions that a model can realistically discover under finite sampling. Using this framework, we show that:

1. **The current RLVR recipe primarily preserves rather than expands the base model's solution coverage.** Across diverse reasoning benchmarks, RLVR consistently loses access to more correct solutions than it gains, even while improving single-sample accuracy. We additionally analyze the temporal evolution of support dynamics across RLVR training steps, uncovering how RLVR progressively narrows the accessible solution space over time.

2. **The precision-diversity trade-off is fundamental, not domain-specific.** This pattern appears across mathematics, logical reasoning, factual QA, and code generation, suggesting it reflects inherent properties of current RLVR methods rather than domain-specific quirks.

3. **Local uncertainty and global diversity can diverge.** The current RLVR recipe sometimes increases token-level entropy (appearing more "uncertain" during generation) while simultaneously reducing answer-level entropy (converging to fewer final solutions).

These findings show that current RLVR may face an "invisible leash". They remain fundamentally constrained by their initialization and cannot discover reasoning patterns that lie outside the base model's effective reach. To break this invisible leash, RLVR may need augmenting with explicit exploration or hybrid strategies that seed probability mass into underrepresented regions of the solution space. We hope this work offers novel insights into the strengths and limitations of the current RLVR recipe, guiding future efforts in improving RLVR practice and building LLM systems that can unlock genuinely new reasoning capacity.

## 2 PRELIMINARIES

### 2.1 FORMALIZING SOLUTION ACCESSIBILITY

**Effective Support of Correct Completions.** Let $\mathcal{X}$ denote the space of natural language prompts, and $\mathcal{Y}$ denote the space of token sequences (*e.g.*, reasoning traces or completions). For a fixed prompt $x \in \mathcal{X}$, $q(y \mid x)$ is the output distribution of the base model, and $R(x, y) \in \{0, 1\}$ is a verifiable reward function indicating whether $y$ is a correct solution. Various RLVR algorithms, including PPO (Schulman et al., 2017), RLOO (Kool et al., 2019), GRPO (Guo et al., 2025), DAPO (Yu et al., 2025), or REINFORCE++ (Hu, 2025), learn a new distribution $\pi_\theta(y \mid x)$ to optimize different variants of the following regularized objective: $\max_\theta \mathbb{E}_{y \sim \pi_\theta(\cdot \mid x), x \sim \mathcal{D}} \left[ R(x, y) - \beta^{-1} \log \frac{\pi_\theta(y \mid x)}{q(y \mid x)} \right]$, where $\mathcal{D}$ is the distribution of prompts. An optional log ratio corresponds to a regularized policy update that penalizes divergence from the base model $q$ controlled by a hyperparameter $\beta > 0$.

**Definition 2.1** (Support of Correct Completions). *Let* $\mathcal{C} = \{y \in \mathcal{Y} \mid R(x, y) = 1\}$ *denote the set of correct completions under the reward function* $R$. *Then the* effective support on correct completions *of a distribution* $p(y \mid x)$ *is defined as*

$$\text{supp}(p) := \{y \in \mathcal{C} \mid p(y \mid x) > 0\}.$$

**Empirical Support Relaxation.** The effective support assumes that $q$ has exact zeros in its support, which, however, rarely holds in practice. Softmax layers yield strictly positive probabilities across all tokens, making the nominal support of $q$ span the entire space $\mathcal{Y}$. This factor, along with sampling noise or temperature scaling, contributes to what we refer to as *empirical support diffusion*: over time, the model may assign growing probability mass to completions that initially had negligible—but still nonzero—probability under the base model.

While $q(y \mid x)$ is technically positive for all $y$ due to the softmax, many completions lie so deep in the tail that they are effectively invisible to the training algorithm under finite sampling. To formalize this, we develop a relaxation and define the *empirical support under* $\epsilon$ as

$$\text{supp}_\epsilon(q) := \{y \in \mathcal{C} \mid q(y \mid x) > \epsilon\},$$

where $\epsilon > 0$, with $\epsilon \to 0$, denotes a minimal cutoff that separates completions with practically observable likelihood from those that are statistically negligible. Completions outside this threshold are unlikely to be sampled in typical on-policy RL settings with finite rollouts. The choice of $\epsilon$ is thus crucial for assessing which completions are empirically reachable. Intuitively, $\epsilon$ should correspond to the minimum probability required for a correct completion to appear within finite samples. We derive a principled estimate for this threshold based on sampling confidence bounds in Appx. C.4.

## 2.2 CHARACTERIZING HOW RLVR CHANGES SOLUTION ACCESS

With empirical support defined, we categorize what happens to correct solutions under RLVR:

**Definition 2.2** (Empirical Support Dynamics). *For a given threshold* $\epsilon > 0$,

- *We say RLVR achieves* empirical support expansion *under threshold* $\epsilon$ *if* $\text{supp}_\epsilon(\pi_\theta) \setminus \text{supp}_\epsilon(q) \neq \emptyset$, *i.e. there exists at least one completion* $y^* \in \mathcal{C}$ *such that*

$$q(y^* \mid x) \leq \epsilon \quad \text{but} \quad \pi_\theta(y^* \mid x) > \epsilon.$$

  *That is, the RLVR-trained model assigns non-negligible probability mass to correct completions that were effectively negligible under the base model.*

- *We say RLVR exhibits* empirical support shrinkage *under threshold* $\epsilon$ *if* $\text{supp}_\epsilon(q) \setminus \text{supp}_\epsilon(\pi_\theta) \neq \emptyset$, *i.e. there exists at least one completion* $y^* \in \mathcal{C}$ *such that*

$$q(y^* \mid x) > \epsilon \quad \text{but} \quad \pi_\theta(y^* \mid x) \leq \epsilon.$$

  *This formalizes the phenomenon where RLVR concentrates probability mass onto a narrower subset of outputs, effectively excluding correct solutions that were previously accessible under the base model.*

**Support Dynamics Metrics.** To quantify RLVR's impact on solution accessibility, we introduce the following precision and recall-inspired metrics based on these four support categories.

**Definition 2.3** (Support Dynamics Metrics). *Let $P$, $E$, $S$, and $O$ denote the number of correct completions in preservation, expansion, shrinkage, and out-of-support, respectively.*

- *Support Retention Rate (SRR) measures how well RLVR preserves the base model's accessible correct solutions:*

$$SRR = \frac{P}{P + S}$$

- *Net Discovery Rate (NDR) measures the fraction of RLVR's accessible solutions that represent genuine discoveries:*

$$NDR = \frac{E}{P + E}$$

- *Support Dynamic Score (SDS) provides a balanced measure combining retention and discovery:*

$$SDS = \frac{2 \cdot SRR \cdot NDR}{SRR + NDR} = \frac{2PE}{P^2 + 2PE + ES}$$

- *Net Support Change Rate (NSCR) captures the net expansion or shrinkage of empirical support:*

$$NSCR = \frac{E - S}{P + E + S}$$

These metrics provide complementary perspectives on RLVR's behavior:

- **SRR** $\in [0, 1]$: Higher values indicate better preservation of base model solutions. SRR $= 1$ means no shrinkage occurred.

- **NDR** $\in [0, 1]$: Higher values indicate more genuine discovery. NDR $= 0$ means no new solutions were found; NDR $= 1$ means all accessible solutions are discoveries.

- **SDS** $\in [0, 1]$: Harmonic mean balancing retention and discovery. High SDS requires both good retention *and* meaningful expansion.

- **NSCR** $\in [-1, 1]$: Positive values indicate net expansion, negative values indicate net shrinkage.

These metrics enable us to distinguish between different RLVR behaviors: *support-constrained optimization* (high SRR, low NDR), *genuine capability expansion* (high SRR, high NDR), *inefficient redistribution* (low SRR, low NDR), and *aggressive exploration* (low SRR, high NDR). We also provide theoretical foundations to understand RLVR's support-bounded behavior in Appx. C.

## 3 EVIDENCE OF HIDDEN-SUPPORT DYNAMICS

### 3.1 EXPERIMENTAL SETUP

We adopt **ProRL-1.5B-v1** (Liu et al., 2025) as our main RLVR method due to its robust long-horizon training framework. Starting from `DeepSeek-R1-Distill-Qwen-1.5B` as the base model, ProRL's `Nemotron-Research-Reasoning` series leverages GRPO enhanced with decoupled clipping, dynamic sampling, KL divergence regularization, and periodic reference resets to sustain exploration and prevent entropy collapse during extended RL training. In addition, we evaluate other RLVR variants at multiple scales (7B–14B parameters), including `Skywork` (Wei et al., 2023), `AceReason-Nemotron` (Chen et al., 2025a), and `Phi4-Reason` (Abdin et al., 2025), alongside a visual LLM (`Kangheng-OVR-7B` (Wei et al., 2025)).

Performance is measured across two categories. (1) **Math reasoning tasks**: MATH500 (Hendrycks et al., 2021), Minerva (Lewkowycz et al., 2022), OlympiadBench (He et al., 2024), AIME 2024, AIME 2025, and AMC 2023. (2) **Non-math reasoning tasks**: SimpleQA (Wei et al., 2024) (factuality), LiveBench (White et al., 2025) (logic, coding, and language comprehension), SciBench (Wang et al., 2023) (multi-domain scientific problem-solving), and Reasoning Gym (Stojanovski et al., 2025) (cognition, geometry, graph theory, games). In Reasoning Gym, we especially focus on tasks that ProRL explicitly highlighted as challenging for the base model. For SimpleQA, we employ GPT-4.1 (Achiam et al., 2023) as the judge. Sampling budgets are $k \in \{4096, 8192\}$ for math,

Table 1: Aggregate support dynamics across diverse models and domains. Each completion is categorized by correctness and support status: **Preservation** indicates both base and RLVR find the solution; **Shrinkage** indicates only the base model found it; **Expansion** indicates only RLVR found it; and Out of Support denotes solutions found by neither. Higher SRR, NDR, and SDS reflect stronger preservation, genuine discovery, and balanced optimization, respectively. NSCR values closer to zero indicate more balanced support change. `Kangheng-OVR-7B` is included as a vision-language model (VLM). Full detailed statistics for each model are provided in Appx. A.

| Model | Domain | SRR | NDR | SDS | NSCR | P | E | S | O |
|---|---|---|---|---|---|---|---|---|---|
| PRORL-1.5B-V1 | Math | 0.96 | 0.00 | 0.01 | -0.04 | 1355 | 5 | 56 | 131 |
| | Non-Math | 0.91 | 0.03 | 0.06 | -0.06 | 1045 | 31 | 107 | 674 |
| | Overall | 0.94 | 0.02 | 0.03 | -0.05 | 2400 | 36 | 163 | 805 |
| PRORL-1.5B-V2 | Math | 0.96 | 0.01 | 0.01 | -0.04 | 1349 | 9 | 62 | 127 |
| | Non-Math | 0.90 | 0.04 | 0.07 | -0.06 | 1039 | 39 | 113 | 666 |
| | Overall | 0.93 | 0.02 | 0.04 | -0.05 | 2388 | 48 | 175 | 793 |
| NEMOTRON-1-7B | Math | 0.99 | 0.00 | 0.01 | -0.00 | 1431 | 5 | 9 | 102 |
| | Non-Math | 0.97 | 0.02 | 0.04 | -0.02 | 1284 | 23 | 47 | 503 |
| | Overall | 0.98 | 0.01 | 0.02 | -0.01 | 2715 | 28 | 56 | 605 |
| SKYWORK-OR1-7B | Math | 0.98 | 0.00 | 0.00 | -0.02 | 1406 | 2 | 34 | 105 |
| | Non-Math | 0.96 | 0.02 | 0.04 | -0.02 | 1279 | 24 | 52 | 502 |
| | Overall | 0.97 | 0.01 | 0.02 | -0.02 | 2685 | 26 | 86 | 607 |
| NEMOTRON-1-14B | Math | 0.99 | 0.00 | 0.01 | -0.01 | 1425 | 5 | 15 | 102 |
| | Non-Math | 0.99 | 0.00 | 0.01 | -0.01 | 993 | 3 | 8 | 399 |
| | Overall | 0.99 | 0.00 | 0.01 | -0.01 | 2418 | 8 | 23 | 501 |
| PHI4-REASON-PLUS-14B | Math | 0.99 | 0.01 | 0.01 | -0.00 | 1407 | 8 | 12 | 120 |
| | Non-Math | 0.99 | 0.01 | 0.01 | -0.00 | 1067 | 8 | 11 | 317 |
| | Overall | 0.99 | 0.01 | 0.01 | -0.00 | 2474 | 16 | 23 | 437 |
| OLMO-2-0425-1B | Math | 0.887 | 0.110 | 0.166 | -0.072 | 761 | 83 | 104 | 599 |
| KANGHENG-OVR-7B (VLM) | Math | 1.00 | 0.00 | 0.01 | -0.00 | 781 | 3 | 4 | 516 |

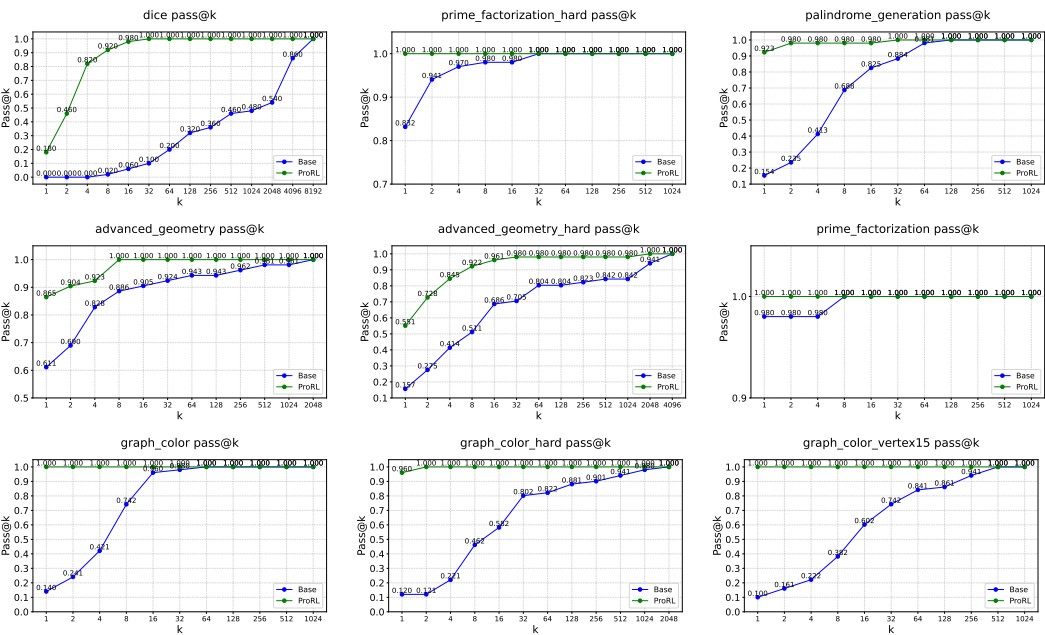

Figure 1: Typical empirical support preservation in Reasoning Gym tasks, like Graph Coloring, Palindrome Generation, and Advanced Geometry.

$k \in \{1024, 2048, 4096, 8192, 16384\}$ for Reasoning Gym, and $k \in \{1024, 2048\}$ for other non-math datasets, ensuring that any unreachable solution $y^* \in \mathcal{C}$ remains below the empirical support threshold of the base model. More implementation details appear in Appx. B.

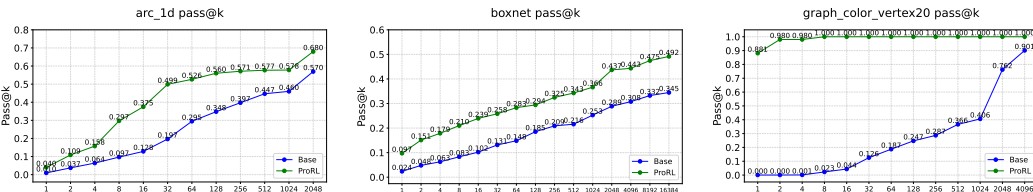

Figure 2: Instances of empirical support expansion, as seen in Boxnet, Dice, and Arc 1D tasks.

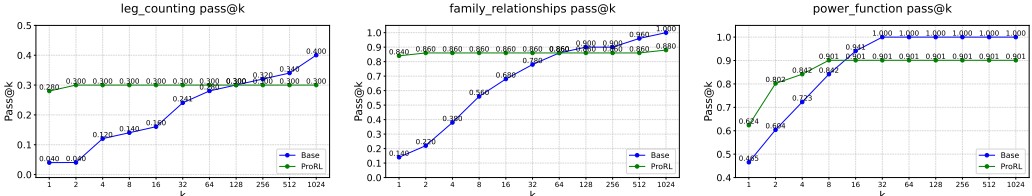

Figure 3: Examples of empirical support shrinkage on Reasoning Gym tasks such as Leg Counting, Family Relationships, and Power Function.

### 3.2 RESULTS: PREDOMINANT PRESERVATION WITH LIMITED EXPANSION

**Support preservation dominates across all domains.** Table 1 shows that across diverse model scales and families, RLVR predominantly acts as a support-constrained optimization mechanism. All models achieve very high support retention (overall SRR ≈ 0.93–0.99) while genuine discovery remains rare (NDR ≤ 0.04). For example, Nemotron-7B and Nemotron-14B retain nearly all base-model solutions (SRR ≥ 0.98) with negligible expansion. Even smaller-scale ProRL-1.5B achieves SRR = 0.93 with only modest gains (NDR = 0.02). These patterns persist across math (SRR = 0.96–0.99, NDR ≈ 0.00–0.01) and non-math domains (SRR = 0.90–0.99, NDR ≤ 0.04). Preservation-dominated behavior is especially clear in Reasoning Gym tasks such as graph_color and palindrome, where ProRL accelerates convergence toward near-perfect pass@k with large budgets (Fig. 1). Support counts confirm this: *most correct completions remain shared between RLVR and base models.*

**Selective but limited empirical support expansion.** Despite the strong conservation, RLVR occasionally recovers solutions negligible to the base model. Expansion is consistently small: ProRL-1.5B discovers 48 new completions across 11 benchmarks, while larger models (e.g., Phi4-14B, Nemotron-14B) add fewer than 10. Non-math datasets exhibit the highest relative discovery (NDR ≤ 0.04), whereas math datasets are virtually stagnant (NDR ≤ 0.01). Some Reasoning Gym tasks, such as graph_color_vertex20 and arc_1d, show genuine expansion (Fig. 2), but remain isolated exceptions rather than the dominant trend. These suggest that while RLVR can occasionally redistribute mass into underexplored solution modes, such expansion remains the exception rather than the rule, challenging assumptions about RLVR's capacity for genuine reasoning horizon extension.

**Empirical support shrinkage outweighs expansion.** Across all models and domains, shrinkage consistently exceeds expansion. ProRL-1.5B loses 175 completions while gaining only 48 (ratio ≈ 3.6:1), while Nemotron-7B and Skywork-OR1-7B display similar patterns (ratios ≈ 2:1–3:1). Even large models (Nemotron-14B, Phi4-14B) show net shrinkage despite near-perfect preservation. Overall NSCR values remain slightly negative (−0.01 to −0.06), showing that RLVR systematically narrows the accessible solution set. This explains paradoxical outcomes: while RLVR models outperform bases at low $k$, base models dominate at high $k$ due to broader solution coverage (e.g., AIME2024 base pass@8192 = 93.3% vs. ProRL-1.5B's 83.3%). Reasoning Gym tasks like leg_counting, family_relationships, and power_function illustrate this vividly (Fig. 3).

**Support dynamic score confirms imbalance.** SDS values remain consistently low across all scales ($\leq 0.07$), reflecting poor balance between preservation and discovery. The highest observed SDS

Table 2: **Comparison of support dynamics between SFT and DAPO training on Qwen2.5-Math-7B.** SFT moderately expands support, while DAPO sharpens support but often reduces stability and increases shrinkage (lower NSCR).

| Dataset | Training | pass@k | | Support Dynamics Metrics | | | | Support Counts | | | |
|---|---|---|---|---|---|---|---|---|---|---|---|
| | | Base | Target | SRR | NDR | SDS | NSCR | P | E | S | O |
| *Math Reasoning Benchmarks (pass@256)* | | | | | | | | | | | |
| AIME2024 | SFT | 63.33% | 63.33% | 0.789 | 0.211 | 0.332 | 0.000 | 15 | 4 | 4 | 7 |
| | DAPO | 63.33% | 66.67% | 0.895 | 0.150 | 0.257 | 0.045 | 17 | 3 | 2 | 8 |
| AIME2025 | SFT | 50.00% | 63.33% | 0.933 | 0.263 | 0.411 | 0.200 | 14 | 5 | 1 | 10 |
| | DAPO | 50.00% | 56.67% | 1.000 | 0.118 | 0.211 | 0.118 | 15 | 2 | 0 | 13 |
| AMC23 | SFT | 100.00% | 100.00% | 1.000 | 0.000 | 0.000 | 0.000 | 40 | 0 | 0 | 0 |
| | DAPO | 100.00% | 92.50% | 0.925 | 0.000 | 0.000 | -0.075 | 37 | 0 | 3 | 0 |
| MATH500 | SFT | 96.40% | 99.00% | 0.996 | 0.030 | 0.059 | 0.026 | 480 | 15 | 2 | 3 |
| | DAPO | 96.40% | 95.20% | 0.975 | 0.013 | 0.025 | -0.012 | 470 | 6 | 12 | 12 |
| Minerva | SFT | 63.24% | 66.91% | 0.895 | 0.154 | 0.263 | 0.050 | 154 | 28 | 18 | 72 |
| | DAPO | 63.24% | 55.15% | 0.802 | 0.080 | 0.145 | -0.120 | 138 | 12 | 34 | 88 |
| Olympiad | SFT | 78.22% | 81.78% | 0.953 | 0.089 | 0.162 | 0.042 | 503 | 49 | 25 | 98 |
| | DAPO | 78.22% | 72.89% | 0.884 | 0.051 | 0.096 | -0.065 | 467 | 25 | 61 | 122 |

is LiveBench-L at $0.288$, but even this corresponds to 2 expansions against 4 shrinkages. Math benchmarks are particularly imbalanced (SDS $\approx 0.00$–$0.01$), while non-math domains fare only marginally better. Thus, RLVR's improvements arise primarily from mass concentration, not meaningful solution expansion.

**Perplexity analysis on support constraints.** Tab. 3 reports perplexity where base and ProRL models are evaluated against *external* reasoning traces from DeepSeek-R1 and Claude Sonnet 4. ProRL consistently shows higher perplexity. For instance, on AIME2024, perplexity against Claude Sonnet 4 rises from 8.76 (Base) to 14.91 (ProRL), which indicates RLVR reduces the model's ability to assign probability to diverse external reasoning styles. While format differences contribute, the dominant effect is structural: RLVR concentrates probability around narrower solution trajectories. Correctness patterns further reveal the precision-coverage trade-off. In *shrinkage* cases, ProRL shows higher perplexity even when the base model succeeds, confirming RLVR collapses mass away from viable solution pathways. Conversely, modest perplexity improvements in rare *expansion* cases suggest ProRL's new successful trajectories originate from the base model's low-density tails rather than genuinely novel reasoning structures.

**Comparison between SFT and RLVR**. We fix the base model (Qwen2.5-Math-7B), dataset (DeepMath-103K), sampling protocol, and optimization hyperparameters, varying only the training objective (SFT vs. DAPO). As shown in Tab. 3.2 and A, SFT consistently produces *moderate support expansion* with positive NSCR values across benchmarks, whereas DAPO produces *sharply concentrated* distributions with mixed or negative support change. While DAPO improves precision at low sampling budgets, its support dynamics reflect a *high-SRR but low-NDR* regime: preservation of known correct solutions but limited discovery of new ones. On several benchmarks such as MATH500, Minerva, and Olympiad, DAPO reduces accessible correct solutions, causing *net shrinkage despite identical training conditions*. These results confirm that support-constrained behavior is not an artifact of scaling, data mixture, or procedure, but emerges from the objective itself. By contrast, SFT expands the reachable solution set without collapsing existing modes, underscoring that shrinkage is intrinsic to RLVR-style multiplicative updates rather than SFt.

**Support dynamics during RLVR training.** Tab, A summarizes how support-dynamics metrics evolve across RLVR training steps. Across benchmarks, SRR remains consistently high, showing that RLVR reliably preserves previously correct reasoning paths, while NSCR gradually decreases, indicating a steady contraction of the model's solution support. On harder datasets, mild fluctuations in NDR and SDS reflect transient exploration that is subsequently pruned. Overall, these checkpoint-level statistics reveal that RLVR reshapes rather than monotonically improves the reasoning distribution. It reinforces a narrow set of stable trajectories over time, which explains both the early-stage gains and the late-stage degradation observed on more diverse math tasks.

Table 3: Perplexity of reasoning tokens from base and RLVR across math benchmarks, segmented by correctness patterns and reference types. For different problem categories (e.g., shrinkage and expansion), perplexity is computed against external references (DeepSeek-R1 and Claude Sonnet 4), reflecting each model's compatibility with diverse and broad solution modes.

| Category | Correctness | Reference | Target | AIME 2024 | AIME 2025 | Olympiad |
|---|---|---|---|---|---|---|
| Shrinkage | ✓ Base, ✗ ProRL | DeepSeek-R1 | Base | 1.24 | 1.39 | 1.17 |
| | | | ProRL | 1.39 | 1.70 | 1.25 |
| | | Claude Sonnet 4 | Base | 1.70 | 1.54 | 1.51 |
| | | | ProRL | 2.12 | 1.98 | 1.83 |
| Expansion | ✗ Base, ✓ ProRL | DeepSeek-R1 | Base | - | - | 1.41 |
| | | | ProRL | - | - | 1.28 |
| | | Claude Sonnet 4 | Base | - | - | 1.65 |
| | | | ProRL | - | - | 1.38 |
| – – | ✗ Base, ✗ ProRL | DeepSeek-R1 | Base | 1.82 | 1.75 | 1.62 |
| | | | ProRL | 2.20 | 2.15 | 1.94 |
| | | Claude Sonnet 4 | Base | 8.76 | 6.05 | 5.98 |
| | | | ProRL | 14.91 | 9.76 | 9.55 |

**Overall takeaway: RLVR as precision enhancer, not capability expander.** Across model scales (1.5B–14B) and domains (math, non-math, multimodal), RLVR consistently behaves as a support-bounded optimizer. With SRR near one but NDR near zero, and uniformly negative NSCR, RLVR enhances precision by concentrating mass on known high-reward solutions but rarely discovers new reasoning paths. This aligns with the *Temporal Forgetting* effect (Li et al., 2025). Breaking RLVR's *invisible leash* may thus require explicit exploration or hybrid strategies that deliberately seed probability mass into underrepresented solution regions.

### 3.3 WHEN EMPIRICAL SUPPORT EXPANSION OCCURS

Although rare, empirical support expansion follows clear patterns that align with the base model's latent capabilities. Across Reasoning Gym tasks, we identify two primary mechanisms that explain why expansion arises in a small set of cases such as `dice`, `arc_1d`, `boxnet`, and `graph_color_vertex20`.

**(1) RLVR Recomposes Subskills the Base Model Already Possesses** These expansion tasks exhibit modular or compositional structure, such as local moves in graph coloring, element-wise updates in `arc_1d`, or JSON-style key-value fragments in `boxnet`. The base model assigns non-negligible mass to many of these fragments individually, but not to their correct global combination. RLVR magnifies these weakly represented components and helps the model assemble them coherently. This matches our empirical patterns: all observed expansions occur in tasks where the base model's `pass@1` is low, but its `pass@k` curve rises steadily (Fig. 2), indicating that all fragments needed for correct reasoning already lie in the base model's long tail. RLVR amplifies these long-tail valid completions, elevating a few above the empirical support threshold $\epsilon$.

**(2) RLVR Corrects Prompt-Format Misalignment and Extracts Latent Ability** A second driver is the sensitivity to prompt format. In several expanding tasks, especially `dice` and `boxnet`, the base model demonstrates partial competence but fails to follow the exact response format required by the reward function. RLVR reshapes the distribution to follow instructions more effectively, unlocking capabilities that were previously present but suppressed. Consistent with this interpretation, the perplexity gaps in Table 3 remain modest in expansion cases, showing that the "new" completions are stylistic or formatting variants of reasoning patterns already accessible to the base model, not fundamentally novel solutions beyond its support.

**Why Expansion Is Small and Bounded** Even when these two favorable mechanisms align, expansion remains sharply limited. Across all benchmarks, NDR never exceeds 0.04 (Tab. 1), NSCR remains negative in all models. Thus, RLVR's gains arise from amplifying low-probability but *exist-*

*ing* solution fragments or format-correct variants, but never from discovering solutions truly absent from the base distribution.

## 4 ENTROPY REDUCTION AND THE PASS@K TRADE-OFF

### 4.1 EXPERIMENTAL SETUP

To study how RLVR reshapes the sampling distribution, we examine the base model and RLVR with a medium sampling budget $k = 32$ on the math reasoning benchmarks. We quantify changes in the output distribution using two entropy metrics:

- **Token-Level Entropy:** Let $\mathcal{V}$ denote the vocabulary and $y^{(i)} = (y_1^{(i)}, y_2^{(i)}, \ldots, y_{T^{(i)}}^{(i)})$ denote the $i$-th generated sequence of length $T^{(i)}$ for $1 \leq i \leq N$. At each timestep $t$, the model outputs a probability distribution $p_t^{(i)}(v)$ over vocabulary tokens $v \in \mathcal{V}$. The entropy of this distribution is given by: $H_t^{(i)} = -\sum_{v \in \mathcal{V}} p_t^{(i)}(v) \log p_t^{(i)}(v)$. The average token-level entropy over all $N$ sequences and their timesteps is computed as: TokenEntropy $= \frac{1}{N} \sum_{i=1}^{N} \left( \frac{1}{T^{(i)}} \sum_{t=1}^{T^{(i)}} H_t^{(i)} \right)$, capturing the local uncertainty at each generation step.

- **Answer-level Entropy:** Let $\{o^{(1)}, \ldots, o^{(N)}\}$ denote the answers extracted from each generated sequence $y^{(i)}$ (using NA for incomplete outputs), and let $\{o_1^*, \ldots, o_M^*\}$ be the $M$ unique answers. Let $f_j$ be the frequency of answer $o_j^*$, with empirical probability $p_j = \frac{f_j}{N}$. Then: AnswerEntropy $= -\sum_{j=1}^{M} p_j \log p_j$. This captures global diversity over output completions, with lower values indicating increased mode collapse.

### 4.2 RESULTS: PRECISION GAINS, ENTROPY DYNAMICS, AND TRADE-OFFS

**Consistent gains in precision, but sharper global distributions.** Tab. 4 shows that RLVR consistently improves `avg@32` across all benchmarks, raising average performance from 54.5% to 65.4% for ProRL and from 43.0% to 61.3% for DAPO (Yu et al., 2025). However, this increased precision comes at a cost: RLVR systematically reduces *answer-level entropy*, indicating a collapse onto fewer distinct solutions and empirically validating our theoretical prediction that reward optimization sharpens output distributions around known modes, thereby reducing effective support coverage. Notably, intrinsically harder tasks, such as AIME or Minerva, still exhibit higher absolute answer-level entropy for both the base and RLVR models, suggesting that challenging problems inherently foster broader solution spaces that require exploration over more diverse completions.

**Decoupled local uncertainty and global diversity.** While answer-level entropy consistently declines, token-level entropy exhibits more varied behavior. In models like ProRL and DAPO, it increases, suggesting greater local uncertainty during generation, possibly due to longer or more elaborated reasoning chains that introduce additional decision points or "forking" tokens (Wang et al., 2025). However, this pattern is far from universal: other RLVR models like AceReason and Skywork display similar or even lower token-level entropy relative to their base counterparts, and prior work has documented sharp entropy collapse in early training phases (Cui et al., 2025).

More importantly, increased token-level entropy does not imply greater exploration of the output space. Despite appearing more stochastic at the step level, RLVR models frequently converge onto a smaller set of final answers—reflected in lower answer-level entropy. Notably, even between two models built on the same base (DeepSeek-7B), Skywork-OR1-7B shows *lower* token-level entropy than AceReason-7B, yet exhibits *higher* answer-level entropy. This contrast highlights that local uncertainty does not reliably predict the diversity of final solutions, revealing a critical decoupling between local uncertainty and global diversity. We refer to this phenomenon as *local stochasticity without global exploration*: the model exhibits variability in generation but ultimately collapses to a narrow set of solutions. Thus, token-level entropy should not be conflated with genuine exploratory behavior, and interpreting entropy dynamics in RLVR requires distinguishing between stepwise uncertainty and overall support expansion.

Table 4: Summary of `avg@32` accuracy, response length, and entropy metrics across math reasoning benchmarks (row colors: ▢ base models, ▢ RLVR models). RLVR consistently improves accuracy and alters distributional properties. While answer-level entropy consistently decreases, token-level entropy shows more varied behavior across models.

| Metric | Model | AIME 2024 | AMC 2023 | MATH 500 | Minerva | Olympiad | Avg. |
|---|---|---|---|---|---|---|---|
| avg@32 Acc. (%) | DeepSeek-1.5B | 31.15 | 72.81 | 85.01 | 32.18 | 51.55 | 54.54 |
| | ProRL-1.5B | 45.62 | 85.70 | 92.01 | 39.27 | 64.56 | 65.43 |
| | DeepSeek-7B | 53.23 | 89.30 | 93.95 | 43.07 | 66.67 | 69.24 |
| | AceReason-7B | 65.83 | 95.08 | 95.81 | 45.35 | 73.92 | 75.20 |
| | Skywork-OR1-7B | 67.40 | 93.59 | 95.73 | 43.81 | 73.05 | 74.71 |
| | DeepSeek-14B | 67.81 | 95.39 | 95.28 | 46.43 | 72.06 | 75.39 |
| | AceReason-14B | 77.29 | 98.67 | 97.01 | 47.20 | 77.74 | 79.58 |
| | Qwen2.5-32B | 18.12 | 55.23 | 75.84 | 24.55 | 41.40 | 43.03 |
| | DAPO-32B | 51.25 | 92.81 | 80.75 | 32.50 | 49.15 | 61.29 |
| Response Length | DeepSeek-1.5B | 16363 | 9979 | 5700 | 8194 | 11873 | 10422 |
| | ProRL-1.5B | 7786 | 6294 | 5070 | 6569 | 6678 | 6479 |
| | DeepSeek-7B | 13613 | 6402 | 4125 | 5595 | 8988 | 7745 |
| | AceReason-7B | 10740 | 5961 | 4313 | 6261 | 7703 | 6995 |
| | Skywork-OR1-7B | 15628 | 8282 | 5735 | 8742 | 12094 | 10096 |
| | DeepSeek-14B | 11295 | 5735 | 3781 | 4919 | 8042 | 6755 |
| | AceReason-14B | 13871 | 7239 | 4622 | 7720 | 10033 | 8697 |
| | Qwen2.5-32B | 1247 | 874 | 585 | 3544 | 881 | 1426 |
| | DAPO-32B | 6908 | 3157 | 3386 | 5665 | 5827 | 4989 |
| Token-Level Entropy | DeepSeek-1.5B | 0.45 | 0.40 | 0.42 | 0.49 | 0.44 | 0.44 |
| | ProRL-1.5B | 0.47▲ | 0.51▲ | 0.54▲ | 0.55▲ | 0.52▲ | 0.52▲ |
| | DeepSeek-7B | 0.38 | 0.34 | 0.35 | 0.39 | 0.38 | 0.37 |
| | AceReason-7B | 0.18▼ | 0.23▼ | 0.27▼ | 0.24▼ | 0.23▼ | 0.23▼ |
| | Skywork-OR1-7B | 0.14▼ | 0.16▼ | 0.19▼ | 0.17▼ | 0.16▼ | 0.16▼ |
| | DeepSeek-14B | 0.33 | 0.30 | 0.32 | 0.35 | 0.33 | 0.33 |
| | AceReason-14B | 0.12▼ | 0.13▼ | 0.15▼ | 0.15▼ | 0.14▼ | 0.14▼ |
| | Qwen2.5-32B | 0.17 | 0.16 | 0.15 | 0.28 | 0.15 | 0.18 |
| | DAPO-32B | 0.26▲ | 0.19▲ | 0.27▲ | 0.44▲ | 0.30▲ | 0.29▲ |
| Answer-Level Entropy | DeepSeek-1.5B | 2.15 | 0.91 | 0.46 | 1.65 | 1.33 | 1.30 |
| | ProRL-1.5B | 1.24 | 0.35 | 0.18 | 0.90 | 0.63 | 0.66 |
| | DeepSeek-7B | 1.47 | 0.36 | 0.18 | 0.96 | 0.80 | 0.75 |
| | AceReason-7B | 0.96 | 0.14 | 0.11 | 0.77 | 0.53 | 0.50 |
| | Skywork-OR1-7B | 0.97 | 0.20 | 0.12 | 0.80 | 0.58 | 0.54 |
| | DeepSeek-14B | 1.01 | 0.14 | 0.13 | 0.83 | 0.59 | 0.54 |
| | AceReason-14B | 0.66 | 0.06 | 0.07 | 0.67 | 0.44 | 0.38 |
| | Qwen2.5-32B | 2.37 | 1.32 | 0.68 | 2.27 | 1.41 | 1.61 |
| | DAPO-32B | 1.12 | 0.09 | 0.26 | 0.96 | 0.63 | 0.61 |

**Implications.** Our empirical analysis reveals a trade-off in RLVR: it improves precision by amplifying high-reward outputs, but simultaneously narrows the diversity of global solutions. This limitation is especially consequential in domains that admit multiple valid answers or benefit from creative reasoning, underscoring the need for explicit exploration mechanisms or diversity-promoting strategies to complement standard RLVR. Moreover, the observed divergence between token-level and answer-level entropy highlights the need for a more nuanced interpretation of stochasticity in reward-optimized models—showing that precision gains often come at the expense of global diversity, and that maintaining controlled variability is critical for sustaining effective exploration.

## 5 CONCLUSION

We reveal that the current RLVR improves precision by sharpening distributions around known high-reward trajectories, yet largely preserves the base model's support. Importantly, we found that this sharpening does not merely prune incorrect outputs—it can also concentrate probability mass on a narrower subset of correct solutions, occasionally excluding valid alternatives that the more diverse base model could still recover. Meanwhile, the divergence between token-level uncertainty and answer-level diversity indicates that stepwise stochasticity alone is insufficient for global exploration, motivating future work to bridge this gap. We suggest that to expand reasoning capabilities beyond the base model's scope, RLVR must be coupled with explicit exploration strategies or off-policy mechanisms that seed probability mass into underrepresented regions of the solution space.

ETHICS STATEMENT

This research was conducted in accordance with established academic ethical standards using publicly available models and appropriate computational resources. All models were accessed through proper channels with necessary permissions, and outputs were analyzed solely for research purposes to understand fundamental properties of reinforcement learning techniques. We acknowledge the significant computational resources required for LLM evaluation and made efforts to optimize experimental design while maintaining scientific rigor. This work investigates limitations of current RL approaches in AI systems, and we believe transparent reporting of these constraints is essential for guiding effective future research directions, helping practitioners set appropriate expectations, and contributing to broader scientific understanding of AI capabilities. While our analysis focuses on specific model families and current benchmarks, which may limit generalizability, we encourage continued research into these limitations and the development of methods that can genuinely expand model reasoning capabilities while maintaining the beneficial precision improvements that RLVR provides.

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

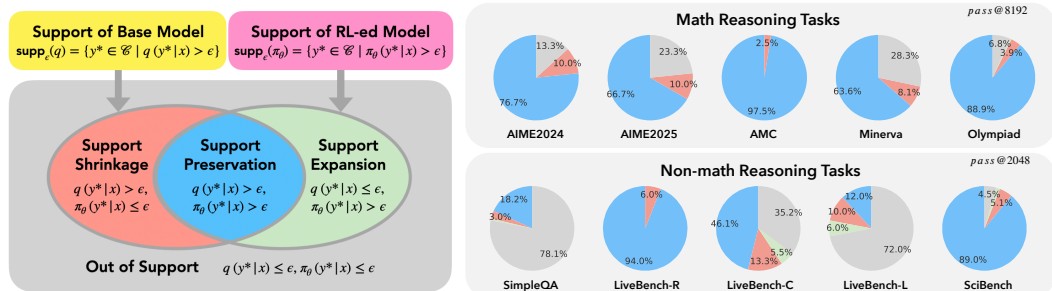

Figure 4: **Left:** Conceptual illustration of empirical support. We define four regions based on whether a correct completion $y^* \in \mathcal{C}$ is assigned non-negligible probability mass by the base and RLVR models, $q$ and $\pi_\theta$ *Support Preservation* covers completions with $q(y^*|x) > \epsilon$ and $\pi_\theta(y^*|x) > \epsilon$; *Support Shrinkage* includes correct completions downweighted by RLVR below $\epsilon$; *Support Expansion* includes completions that RLVR newly upweights above $\epsilon$ despite negligible base model mass; and *Out of Support* refers to completions missed by both. **Right:** Pie charts showing the proportion of completions in each category across diverse reasoning tasks.

## A  DETAILED STATISTICS FOR SUPPORT DYNAMICS

We provide full per-model statistics that underlie the aggregate values in Table 1. For each model and domain (Math, Non-Math, and Overall), we report the raw counts of correct completions across the four empirical support categories: *Preservation (P)*, *Expansion (E)*, *Shrinkage (S)*, and *Out-of-Support (O)*. From these counts, we compute the derived metrics: Support Retention Rate (SRR), Net Discovery Rate (NDR), Support Dynamic Score (SDS), and Net Support Change Rate (NSCR).

These expanded tables enable a fine-grained comparison of how different RLVR variants and model scales redistribute probability mass across correct solutions. In particular, they clarify whether improvements in single-sample accuracy stem from strong preservation of the base model's support, from genuine discovery of new solutions, or from trade-offs between expansion and shrinkage.

We include results for all evaluated models: `ProRL-1.5B-V2`, `Nemotron-1-7B`, `Skywork-OR1-7B`, `AceReason-Nemotron-1-14B`, `Phi4-Reason-Plus-14B`, and the visual reasoning model `Kangheng-OVR-7B (VLM)`. These tables serve as the ground truth for the aggregate summaries in the main text and substantiate the claims about predominant preservation, limited expansion, and consistent shrinkage observed across both math and non-math domains.

## B  EXPERIMENTAL DETAILS

We provide comprehensive details of the experimental setup, including dataset descriptions and evaluation methodologies. A key aspect of our evaluation approach is the answer processing enhancement framework for Reasoning Gym, which addresses format compatibility challenges between base and ProRL models to ensure fair evaluation.

### B.1  EVALUATION SETTINGS

We employed vLLM (Kwon et al., 2023) as the inference backend. For all models, we utilized a sampling temperature of $0.6$, a $top\_p$ value of $0.95$, and a maximum response length of 32768.

### B.2  DATASETS

**Math benchmarks.** We utilized the complete datasets from MATH500 (Hendrycks et al., 2021), Minerva (Lewkowycz et al., 2022), OlympiadBench (He et al., 2024), AIME 2024, AIME 2025, and AMC 2023 for evaluating LLMs. For vision-language models, we evaluated on the testmini sets of MathVision (Wang et al., 2024) and MathVista (Lu et al., 2023).

Table 5: Support dynamics metrics and `pass@k` performance of ProRL-1.5B-v1, compared with its base model, DeepSeek-R1-Distill-Qwen-1.5B.

| Dataset | pass@k Performance | | Support Dynamics Metrics | | | | Support Counts | | | |
|---|---|---|---|---|---|---|---|---|---|---|
| | Base | RLVR | SRR | NDR | SDS | NSCR | P | E | S | O |
| *Math Reasoning Benchmarks (pass@8192)* | | | | | | | | | | |
| AIME2024 | 93.3% | 83.3% | 0.893 | 0.000 | 0.000 | -0.107 | 25 | 0 | 3 | 2 |
| AIME2025 | 80.0% | 73.3% | 0.833 | 0.091 | 0.164 | -0.077 | 20 | 2 | 4 | 4 |
| AMC | 100.0% | 100.0% | 1.000 | 0.000 | 0.000 | 0.000 | 40 | 0 | 0 | 0 |
| Math | 99.6% | 99.4% | 0.998 | 0.000 | 0.000 | -0.002 | 497 | 0 | 1 | 2 |
| Minerva | 71.7% | 63.6% | 0.887 | 0.000 | 0.000 | -0.113 | 173 | 0 | 22 | 77 |
| Olympiad | 92.7% | 89.3% | 0.958 | 0.005 | 0.010 | -0.037 | 600 | 3 | 26 | 46 |
| *Non-Math Reasoning Benchmarks (pass@2048)* | | | | | | | | | | |
| SimpleQA | 23.3% | 18.0% | 0.743 | 0.038 | 0.073 | -0.221 | 75 | 3 | 26 | 329 |
| LiveBench-R | 100.0% | 94.0% | 0.940 | 0.000 | 0.000 | -0.060 | 94 | 0 | 6 | 0 |
| LiveBench-C | 62.5% | 56.2% | 0.838 | 0.069 | 0.128 | -0.094 | 67 | 5 | 13 | 43 |
| LiveBench-L | 26.0% | 24.0% | 0.769 | 0.167 | 0.274 | -0.067 | 10 | 2 | 3 | 35 |
| SciBench | 94.1% | 90.5% | 0.946 | 0.016 | 0.031 | -0.038 | 616 | 10 | 35 | 31 |
| LiveCodeBench v5 | 46.4% | 43.0% | 0.860 | 0.072 | 0.133 | -0.069 | 129 | 10 | 21 | 163 |
| LiveCodeBench v6 | 43.5% | 42.0% | 0.947 | 0.018 | 0.036 | -0.034 | 54 | 1 | 3 | 73 |
| *Aggregate Statistics* | | | | | | | | | | |
| **Math Benchmarks** | – | – | 0.960 | 0.0037 | 0.0073 | -0.036 | 1355 | 5 | 56 | 131 |
| **Non-Math Benchmarks** | – | – | 0.907 | 0.0288 | 0.0558 | -0.064 | 1045 | 31 | 107 | 674 |
| **Overall** | – | – | 0.936 | 0.0148 | 0.0291 | -0.049 | 2400 | 36 | 163 | 805 |

Table 6: Support dynamics metrics and `pass@k` performance of AceReason-Nemotron-1-7B, compared with its base model, DeepSeek-7B.

| Dataset | pass@k Performance | | Support Dynamics Metrics | | | | Support Counts | | | |
|---|---|---|---|---|---|---|---|---|---|---|
| | Base | RLVR | SRR | NDR | SDS | NSCR | P | E | S | O |
| *Math Reasoning Benchmarks (pass@8192)* | | | | | | | | | | |
| AIME2024 | 93.3% | 93.3% | 1.000 | 0.000 | 0.000 | 0.000 | 28 | 0 | 0 | 2 |
| AIME2025 | 100.0% | 100.0% | 1.000 | 0.000 | 0.000 | 0.000 | 30 | 0 | 0 | 0 |
| AMC | 100.0% | 100.0% | 1.000 | 0.000 | 0.000 | 0.000 | 40 | 0 | 0 | 0 |
| Math | 99.8% | 99.8% | 1.000 | 0.000 | 0.000 | 0.000 | 499 | 0 | 0 | 1 |
| Minerva | 71.7% | 71.0% | 0.985 | 0.005 | 0.010 | -0.010 | 192 | 1 | 3 | 76 |
| Olympiad | 96.0% | 95.7% | 0.991 | 0.006 | 0.012 | -0.003 | 642 | 4 | 6 | 23 |
| *Non-Math Reasoning Benchmarks (pass@2048)* | | | | | | | | | | |
| SimpleQA | 38.6% | 35.6% | 0.862 | 0.065 | 0.121 | -0.073 | 144 | 10 | 23 | 256 |
| LiveBench-R | 100.0% | 99.0% | 0.990 | 0.000 | 0.000 | -0.010 | 99 | 0 | 1 | 0 |
| LiveBench-C | 85.9% | 85.9% | 0.991 | 0.009 | 0.018 | 0.000 | 109 | 1 | 1 | 17 |
| LiveBench-L | 24.0% | 24.0% | 0.833 | 0.167 | 0.278 | 0.000 | 10 | 2 | 2 | 36 |
| SciBench | 94.7% | 93.5% | 0.982 | 0.006 | 0.012 | -0.012 | 643 | 4 | 12 | 33 |
| LiveCodeBench v5 | 62.8% | 62.5% | 0.970 | 0.025 | 0.048 | -0.005 | 197 | 5 | 6 | 115 |
| LiveCodeBench v6 | 64.1% | 63.4% | 0.976 | 0.012 | 0.024 | -0.012 | 82 | 1 | 2 | 46 |
| *Aggregate Statistics* | | | | | | | | | | |
| **Math Benchmarks** | – | – | 0.994 | 0.003 | 0.007 | -0.003 | 1431 | 5 | 9 | 102 |
| **Non-Math Benchmarks** | – | – | 0.965 | 0.018 | 0.035 | -0.018 | 1284 | 23 | 47 | 503 |
| **Overall** | – | – | 0.981 | 0.010 | 0.020 | -0.010 | 2715 | 28 | 56 | 605 |

**Non-math benchmarks.** For SimpleQA (Wei et al., 2024), we uniformly sampled 10% of the original dataset (433 samples) to enable efficient large-scale evaluation under high-pass conditions. For LiveBench (White et al., 2025), we used the 2024-11-25 version available on HuggingFace. To ensure unambiguous evaluation, we focused exclusively on tasks with binary correct/incorrect judgments and excluded tasks involving intermediate floating-point judgments, as these lack clear correctness criteria. Based on this selection criterion, we evaluated the following subsets: $web\_of\_lies\_v2$ and $spatial$ subsets for Reasoning tasks (LiveBench-R), the $typos$ subset for Language tasks (LiveBench-L), and all available data for Coding tasks (LiveBench-C). For SciBench (Wang et al., 2023), we evaluated on the complete dataset. For LiveCodeBench (Jain et al.,

Table 7: Support dynamics metrics and `pass@k` performance of Skywork-OR1-7B, compared with its base model, DeepSeek-R1-Distill-Qwen-7B.

| Dataset | pass@k Performance | | Support Dynamics Metrics | | | | Support Counts | | | |
|---|---|---|---|---|---|---|---|---|---|---|
| | Base | RLVR | SRR | NDR | SDS | NSCR | P | E | S | O |
| *Math Reasoning Benchmarks (pass@8192)* | | | | | | | | | | |
| AIME2024 | 93.3% | 93.3% | 1.000 | 0.000 | 0.000 | 0.000 | 28 | 0 | 0 | 2 |
| AIME2025 | 100.0% | 100.0% | 1.000 | 0.000 | 0.000 | 0.000 | 30 | 0 | 0 | 0 |
| AMC | 100.0% | 100.0% | 1.000 | 0.000 | 0.000 | 0.000 | 40 | 0 | 0 | 0 |
| Math | 99.8% | 99.8% | 1.000 | 0.000 | 0.000 | 0.000 | 499 | 0 | 0 | 1 |
| Minerva | 71.7% | 71.3% | 0.985 | 0.010 | 0.020 | -0.005 | 192 | 2 | 3 | 75 |
| Olympiad | 96.0% | 91.4% | 0.952 | 0.000 | 0.000 | -0.048 | 617 | 0 | 31 | 27 |
| *Non-Math Reasoning Benchmarks (pass@2048)* | | | | | | | | | | |
| SimpleQA | 38.6% | 37.0% | 0.880 | 0.081 | 0.149 | -0.039 | 147 | 13 | 20 | 253 |
| LiveBench-R | 100.0% | 98.0% | 0.980 | 0.000 | 0.000 | -0.020 | 98 | 0 | 2 | 0 |
| LiveBench-C | 85.9% | 85.2% | 0.991 | 0.000 | 0.000 | -0.009 | 109 | 0 | 1 | 18 |
| LiveBench-L | 24.0% | 22.0% | 0.917 | 0.000 | 0.000 | -0.083 | 11 | 0 | 1 | 38 |
| SciBench | 94.7% | 92.8% | 0.974 | 0.006 | 0.012 | -0.020 | 638 | 4 | 17 | 33 |
| LiveCodeBench v5 | 62.8% | 62.2% | 0.966 | 0.025 | 0.049 | -0.010 | 196 | 5 | 7 | 115 |
| LiveCodeBench v6 | 64.1% | 62.6% | 0.952 | 0.024 | 0.048 | -0.023 | 80 | 2 | 4 | 45 |
| **Aggregate Statistics** | | | | | | | | | | |
| **Math Benchmarks** | – | – | 0.976 | 0.001 | 0.003 | -0.023 | 1406 | 2 | 34 | 105 |
| **Non-Math Benchmarks** | – | – | 0.961 | 0.018 | 0.036 | -0.021 | 1279 | 24 | 52 | 502 |
| **Overall** | – | – | 0.969 | 0.010 | 0.020 | -0.022 | 2685 | 26 | 86 | 607 |

Table 8: Support dynamics metrics and `pass@k` performance of Nemotron-1-14B, compared with its base model, DeepSeek-R1-Distill-Qwen-14B.

| Dataset | pass@k Performance | | Support Dynamics Metrics | | | | Support Counts | | | |
|---|---|---|---|---|---|---|---|---|---|---|
| | Base | RLVR | SRR | NDR | SDS | NSCR | P | E | S | O |
| *Math Reasoning Benchmarks (pass@4096)* | | | | | | | | | | |
| AIME2024 | 96.7% | 93.3% | 0.966 | 0.000 | 0.000 | -0.034 | 28 | 0 | 1 | 1 |
| AIME2025 | 100.0% | 96.7% | 0.967 | 0.000 | 0.000 | -0.033 | 29 | 0 | 1 | 0 |
| AMC | 100.0% | 100.0% | 1.000 | 0.000 | 0.000 | 0.000 | 40 | 0 | 0 | 0 |
| Math | 99.8% | 99.8% | 1.000 | 0.000 | 0.000 | 0.000 | 499 | 0 | 0 | 1 |
| Minerva | 71.7% | 69.5% | 0.959 | 0.011 | 0.021 | -0.030 | 187 | 2 | 8 | 75 |
| Olympiad | 95.9% | 95.6% | 0.992 | 0.005 | 0.009 | -0.003 | 642 | 3 | 5 | 25 |
| *Non-Math Reasoning Benchmarks (pass@1024)* | | | | | | | | | | |
| SimpleQA | 27.0% | 26.8% | 0.983 | 0.009 | 0.017 | -0.008 | 115 | 1 | 2 | 315 |
| LiveBench-R | 99.0% | 99.0% | 1.000 | 0.000 | 0.000 | 0.000 | 99 | 0 | 0 | 1 |
| LiveBench-C | 92.2% | 92.2% | 1.000 | 0.000 | 0.000 | 0.000 | 118 | 0 | 0 | 10 |
| LiveBench-L | 46.0% | 44.0% | 0.957 | 0.000 | 0.000 | -0.043 | 22 | 0 | 1 | 27 |
| SciBench | 93.1% | 92.6% | 0.992 | 0.003 | 0.006 | -0.005 | 639 | 2 | 5 | 46 |
| **Aggregate Statistics** | | | | | | | | | | |
| **Math Benchmarks** | – | – | 0.990 | 0.0035 | 0.0070 | -0.0069 | 1425 | 5 | 15 | 102 |
| **Non-Math Benchmarks** | – | – | 0.992 | 0.0030 | 0.0060 | -0.0050 | 993 | 3 | 8 | 399 |
| **Overall** | – | – | 0.991 | 0.0033 | 0.0066 | -0.0061 | 2418 | 8 | 23 | 501 |

2024), we evaluated the dataset on both v5 and v6 versions. Due to computational efficiency considerations, we conducted LiveCodeBench evaluation exclusively on 1.5B and 7B models, excluding the 14B variants from this particular benchmark.

**Reasoning Gym.** For Reasoning Gym (Stojanovski et al., 2025), we employ the `easy` set from the version updated after commit *17a8431* in its repository as our `default` task configuration. This choice ensures consistency with the `default` task configuration used in prior evaluations, maintaining comparable experimental conditions. Additionally, we utilize the `hard` set as our challenging evaluation benchmark for further evaluations.

Table 9: Support dynamics metrics and `pass@k` performance of Phi4-Reason-Plus-14B, compared with its base model – Phi4-Reason-14B.

| Dataset | pass@k Performance | | Support Dynamics Metrics | | | | Support Counts | | | |
|---|---|---|---|---|---|---|---|---|---|---|
| | Base | RLVR | SRR | NDR | SDS | NSCR | P | E | S | O |
| *Math Reasoning Benchmarks (pass@4096)* | | | | | | | | | | |
| AIME2024 | 96.7% | 96.7% | 1.000 | 0.000 | 0.000 | 0.000 | 29 | 0 | 0 | 1 |
| AIME2025 | 100.0% | 100.0% | 1.000 | 0.000 | 0.000 | 0.000 | 30 | 0 | 0 | 0 |
| AMC | 100.0% | 100.0% | 1.000 | 0.000 | 0.000 | 0.000 | 40 | 0 | 0 | 0 |
| Math | 100.0% | 99.8% | 0.998 | 0.000 | 0.000 | -0.002 | 499 | 0 | 1 | 0 |
| Minerva | 66.2% | 65.4% | 0.972 | 0.017 | 0.033 | -0.011 | 175 | 3 | 5 | 89 |
| Olympiad | 94.8% | 94.7% | 0.991 | 0.008 | 0.016 | -0.002 | 634 | 5 | 6 | 30 |
| *Non-Math Reasoning Benchmarks (pass@1024)* | | | | | | | | | | |
| SimpleQA | 37.9% | 37.4% | 0.970 | 0.019 | 0.036 | -0.012 | 159 | 3 | 5 | 266 |
| LiveBench-R | 100.0% | 100.0% | 1.000 | 0.000 | 0.000 | 0.000 | 100 | 0 | 0 | 0 |
| LiveBench-C | 97.7% | 96.9% | 0.992 | 0.000 | 0.000 | -0.008 | 124 | 0 | 1 | 3 |
| LiveBench-L | 74.0% | 74.0% | 1.000 | 0.000 | 0.000 | 0.000 | 37 | 0 | 0 | 13 |
| SciBench | 94.2% | 94.2% | 0.992 | 0.008 | 0.015 | 0.000 | 647 | 5 | 5 | 35 |
| *Aggregate Statistics* | | | | | | | | | | |
| **Math Benchmarks** | – | – | 0.992 | 0.0057 | 0.0112 | -0.0028 | 1407 | 8 | 12 | 120 |
| **Non-Math Benchmarks** | – | – | 0.990 | 0.0074 | 0.0148 | -0.0028 | 1067 | 8 | 11 | 317 |
| **Overall** | – | – | 0.991 | 0.0064 | 0.0128 | -0.0028 | 2474 | 16 | 23 | 437 |

Table 10: Support dynamics metrics and `pass@k` performance of vision-language model OVR-7B-RL, compared with its base model, OVR-7B-ColdStart, across visual math reasoning benchmarks.

| Dataset | pass@k Performance | | Support Dynamics Metrics | | | | Support Counts | | | |
|---|---|---|---|---|---|---|---|---|---|---|
| | Base | RLVR | SRR | NDR | SDS | NSCR | P | E | S | O |
| *Math Reasoning Benchmarks (pass@8192)* | | | | | | | | | | |
| MathVista | 49.1% | 49.1% | 0.998 | 0.002 | 0.004 | 0.000 | 490 | 1 | 1 | 508 |
| MathVision | 96.7% | 96.4% | 0.990 | 0.007 | 0.014 | -0.003 | 291 | 2 | 3 | 8 |
| *Aggregate Statistics* | | | | | | | | | | |
| **Math Benchmarks** | – | – | 0.995 | 0.0038 | 0.0076 | -0.0013 | 781 | 3 | 4 | 516 |

## B.3 ANSWER PROCESSING ENHANCEMENT IN REASONING GYM

We identified significant evaluation challenges when testing the base model on Reasoning Gym. The ProRL model, having been trained on Reasoning Gym data, predominantly produces responses that conform to the expected format, leading to much higher accuracy scores. In contrast, the base model struggled with format adherence due to insufficiently detailed prompts, and its limited 1.5B parameter capacity made it particularly susceptible to evaluation inconsistencies. To address these issues, we enhanced both the answer extraction protocol and prompt design to ensure fair and objective accuracy assessments across both models. This causes the differences of ProRL's reported performance and our evaluation results in Reasoning Gym.

### B.3.1 GENERAL ANSWER EXTRACTION PROTOCOL

First, we enhanced the answer extraction protocol with a hierarchical, priority-based extraction mechanism that processes responses through multiple fallback levels. Each level attempts to capture the model's intended answer, and successful extraction at any level bypasses subsequent processing steps.

The strategy first attempts to extract content using the Reasoning Gym's `extract_answer()` function, which captures answers within `<answer></answer>` tags. This approach receives the highest priority since these tags represent Reasoning Gym's default format. When this method fails, the system searches for content within the final `\boxed{}` formatting.

Table 11: Support dynamics metrics and `pass@256` performance of OLMo-2-0425-1B-DPO vs RLVR.

| Dataset | pass@k Performance | | Support Dynamics Metrics | | | | Support Counts | | | |
|---|---|---|---|---|---|---|---|---|---|---|
| | Base | RLVR | SRR | NDR | SDS | NSCR | P | E | S | O |
| *Math Reasoning Benchmarks (pass@256)* | | | | | | | | | | |
| AIME2024 | 20.00% | 16.67% | 0.833 | 0.000 | 0.000 | -0.167 | 5 | 0 | 1 | 24 |
| AIME2025 | 23.33% | 20.00% | 0.571 | 0.333 | 0.421 | -0.111 | 4 | 2 | 3 | 21 |
| AMC23 | 85.00% | 77.50% | 0.912 | 0.000 | 0.000 | -0.088 | 31 | 0 | 3 | 6 |
| MATH500 | 77.80% | 76.20% | 0.920 | 0.060 | 0.113 | -0.019 | 358 | 23 | 31 | 88 |
| Minerva-Math | 33.82% | 32.35% | 0.837 | 0.125 | 0.218 | -0.039 | 77 | 11 | 15 | 169 |
| Olympiad | 49.93% | 49.33% | 0.849 | 0.141 | 0.242 | -0.010 | 286 | 47 | 51 | 291 |
| **Aggregate Statistics** | | | | | | | | | | |
| **Math Benchmarks** | – | – | 0.887 | 0.110 | 0.166 | -0.072 | 761 | 83 | 104 | 599 |

Table 12: **Pass@k comparison of Qwen2.5-Math-7B (SFT vs DAPO)** across five math benchmarks. Higher values indicate better performance.

| **AIME2024** | @1 | @2 | @4 | @8 | @16 | @32 | @64 | @128 | @256 |
|---|---|---|---|---|---|---|---|---|---|
| SFT | 20.00 | 30.00 | 30.00 | 33.33 | 43.33 | 50.00 | 63.33 | 63.33 | 63.33 |
| DAPO | 20.00 | 26.67 | 26.67 | 40.00 | 46.67 | 56.67 | 60.00 | 63.33 | 66.67 |
| **AIME2025** | @1 | @2 | @4 | @8 | @16 | @32 | @64 | @128 | @256 |
| SFT | 16.67 | 16.67 | 26.67 | 30.00 | 33.33 | 40.00 | 43.33 | 50.00 | 63.33 |
| DAPO | 16.67 | 20.00 | 30.00 | 43.33 | 46.67 | 50.00 | 50.00 | 50.00 | 56.67 |
| **AMC23** | @1 | @2 | @4 | @8 | @16 | @32 | @64 | @128 | @256 |
| SFT | 60.00 | 72.50 | 77.50 | 85.00 | 90.00 | 95.00 | 97.50 | 97.50 | 100.00 |
| DAPO | 62.50 | 72.50 | 85.00 | 87.50 | 87.50 | 92.50 | 92.50 | 92.50 | 92.50 |
| **MATH500** | @1 | @2 | @4 | @8 | @16 | @32 | @64 | @128 | @256 |
| SFT | 78.20 | 85.20 | 90.00 | 92.80 | 95.00 | 96.20 | 98.00 | 98.80 | 99.00 |
| DAPO | 85.00 | 89.80 | 91.40 | 93.00 | 93.20 | 93.80 | 94.40 | 94.80 | 95.20 |
| **Minerva-Math** | @1 | @2 | @4 | @8 | @16 | @32 | @64 | @128 | @256 |
| SFT | 36.03 | 39.71 | 43.75 | 47.79 | 51.47 | 54.04 | 63.60 | 65.44 | 66.91 |
| DAPO | 40.81 | 45.96 | 48.16 | 51.10 | 51.10 | 52.57 | 53.68 | 54.41 | 55.15 |

For dice tasks using the base model, failed `extract_answer()` attempts trigger additional processing through Lighteval (Habib et al., 2023)'s `math_normalizer()` function. This function handles \boxed{} capture and converts $a/b$ fractions to LaTeX format \frac{a}{b}. When `extract_answer()` successfully captures $a/b$ fraction answers, the system applies Lighteval's `fix_a_slash_b()` function to achieve the same LaTeX conversion.

For non-dice tasks or when using ProRL models, failed `extract_answer()` attempts utilize Lighteval's `last_boxed_only_string()` and `remove_boxed()` functions. These functions locate content within the final \boxed{}, primarily addressing cases where base model prompt modifications shifted from answer tags to boxed formatting.

As a final fallback, the system extracts content following `</think>` tags when all previous methods fail and the response contains these markers. This safety mechanism captures base model responses that ignore formatting requirements in lengthy tasks.

### B.3.2    TASK-SPECIFIC PROCESSING MODIFICATIONS

Our core answer processing pipeline applies to both models, with additional processing steps designed primarily to address format compatibility issues commonly encountered with base model responses. Specifically, the processing logic for each task is enhanced as follows:

*dice* The ground truth for dice tasks uses $a/b$ fraction format. Base models frequently express fractions in LaTeX format, requiring format standardization for accurate evaluation. For base models only, we convert ground truth fractions from $a/b$ format to LaTeX format `\frac{a}{b}` to ensure both model answers and ground truth use consistent LaTeX formatting. ProRL dice processing maintains $a/b$ formatting for both model answers and ground truth, leveraging the dice samples present in its training data.

*prime_factorization* The ground truth format requires answers to be combinations of numbers and multiplication symbol (i.e., $\times$) only. We implement three key modifications to ensure compatibility with this requirement. First, we standardize LaTeX multiplication symbols by replacing `\times` with $\times$ to meet the evaluation requirements, as base models frequently use LaTeX multiplication symbols instead of standard multiplication signs. Second, we expand LaTeX exponentiation by converting formats like `a^b` into repeated multiplication ($a \times a \times \ldots \times a$ for $b$ iterations), preventing errors when base models consolidate repeated factors into exponential notation. Third, we process response equations by retaining only right-side content when answers contain equals signs, transforming responses like "$561 = 3 \times 11 \times 17$" to "$3 \times 11 \times 17$" to eliminate question restatement that base models commonly include.

*palindrome_generation* The ground truth format expects palindromic character strings (sequences that read the same forwards and backwards). We remove excess whitespace to address frequent spacing issues in base model responses. This transformation converts spaced responses like "k h g a g h k" to "khgaghk", preventing string reversibility judgment failures that occur when spaces interfere with palindrome verification.

*advanced_geometry* The ground truth format requires floating-point numbers. Our processing includes three main steps to handle LaTeX formatting issues commonly produced by base models. First, we remove redundant LaTeX expressions by eliminating `\left` and `\right` markers while converting `^\circ` to $^\circ$ symbol, addressing base models' tendency to use LaTeX for brackets and degree symbols. Second, we convert LaTeX numerical expressions, transforming fractions `\frac{a}{b}` and other LaTeX formats (`\sqrt{a}`, `\sin{a}`, `\log{a}`, `\pi`, etc.) into three-decimal floating-point numbers using the `latex2sympy2_extended` library's `latex2sympy()` function. Third, we evaluate arithmetic expressions containing radicals (such as $2\sqrt{16} + 5\sqrt{4} - 3$) by converting them into three-decimal floating-point numbers using Python's built-in mathematical functions, handling cases where base models output final results as arithmetic expressions rather than computed values.

*power_function* The ground truth format uses e-notation scientific notation. We convert mixed LaTeX and arithmetic symbol scientific notation to ensure format consistency. The system transforms patterns like "$-2.36 \times 10^{-16}$" or "$1.5 \times 10^5$" to e-notation format ("-2.36e-16", "1.5e5"), preventing numerically correct but format-incompatible evaluation errors when base models use mixed LaTeX and arithmetic symbols for scientific notation.

*arc_1d* The ground truth format requires space-separated digit sequences. We handle two types of responses to meet this grid format requirement. For pure numerical responses, we insert spaces between consecutive digits, converting sequences like "22220000000000000000111" to "2 2 2 2 0 0 0 0 0 0 0 0 0 0 0 0 0 0 0 1 1 1". For mixed numerical and textual responses, we extract digits and insert spaces, transforming LaTeX grid formats like `\begin{array}{cccc}` 0 & 0 & 0 & 0 & 0 & 0 & 0 & 0 & 7 & 3 & 0 & 0 & 4 & 6 `\\\end{array}` to "0 0 0 0 0 0 0 7 3 0 0 4 6", addressing base models' tendency to output correct answers in LaTeX grid format.

*boxnet* The ground truth format requires dictionary list formatting `[{key:  value}, ...]`. We implement comprehensive JSON format cleaning to meet these evaluation requirements. Our processing pipeline includes several steps: rejecting pure numerical responses to prevent non-JSON format interference; removing JSON markdown wrappers that eliminate ```` ```json {content} ````  ``` ``` markers; converting single dictionaries to single-element dictionary lists (`dict` $\rightarrow$ `[dict]`); and filtering illegal elements by removing non-dictionary components from JSON lists. Additionally, we clean nested structure values within individual dictionary entries. For nested lists, we extract the first element as the value (`[{key1:  [value1, value2, ...]}`,

...] $\rightarrow$ [{key1: value1}, ...]). For nested dictionaries, we select matching key values when available ([{key1: {key1: value1, key2: value2, ...}}, ...] $\rightarrow$ [{key1: value1}, ...]) or default to the first element value when keys don't match ([{key1: {key2: value2, key3: value3}}, ...] $\rightarrow$ [{key1: value2}, ...]). These modifications preserve model response content to the maximum extent while ensuring ground truth format compliance.

### B.4 ENTROPY ANALYSIS

**Setup.** In entropy analysis, we configure the models with a sampling temperature of $0.6$, a $top\_p$ value of $0.95$, and a maximum response length of $32768$ tokens to balance response diversity and quality. Each model generates 32 completions per problem following the `avg@32` evaluation protocol, and all reported metrics (accuracy, response length, token-level entropy, and answer-level entropy) are averaged across these 32 completions and across all test problems in each benchmark.

**Models.** We evaluate a diverse set of reasoning models to understand the entropy characteristics across different training paradigms and parameter scales, as summarized in the following table.

**Entropy Computation.** For token-level entropy computation, we employ teacher-forcing to obtain probability estimates. Specifically, after generating the 32 completions with the specified sampling parameters, we feed each generated sequence back to the model and perform a single forward pass to compute the probability distribution over the vocabulary at each token position. Answer-level entropy is computed by first extracting the final answer from each completion using Lighteval (Habib et al., 2023), then calculating the entropy over the distribution of unique answers across the 32 completions. This approach allows us to compute both token-level and answer-level entropy directly from the model's probability distributions without introducing additional sampling variance.

## C THEORETICAL LIMITS OF RLVR

### C.1 SUPPORT PRESERVATION: WHY RLVR RARELY DISCOVERS NEW MODES

We begin with a core limitation of RLVR: it is inherently constrained to operate within the support of the base model's distribution. Since RLVR relies on gradient signals derived from samples generated by the base model, it cannot assign a nonzero probability to any solution that can never be sampled from $q(\cdot \mid x)$. As a result, any correct output $y^*$ with $q(y^* \mid x) = 0$ remains inaccessible to policy gradient updates, regardless of reward. We formalize this intuition with Theorem C.1, which makes precise how RLVR's reliance on the base model's sampling prevents discovering truly new solutions.

**Theorem C.1** (Support Preservation under RLVR). *Let $\pi_\theta(y \mid x)$ be the RLVR-trained distribution obtained via standard on-policy gradient updates on verifiable rewards $R$. Then for all $x \in \mathcal{X}$,*

$$\mathrm{supp}(\pi_\theta(\cdot \mid x)) \subseteq \mathrm{supp}(q(\cdot \mid x)).$$

*In particular, if $q(y^* \mid x) = 0$ for some correct solution $y^*$, then RLVR cannot discover $y^*$.*

*Proof.* By construction we initialize the RLVR policy to the base model as $\pi_{\theta_0}(y \mid x) = q(y \mid x)$. Hence

$$\mathrm{supp}\big(\pi_{\theta_0}(\cdot \mid x)\big) = \mathrm{supp}\big(q(\cdot \mid x)\big).$$

**Inductive step.** Assume that at some iteration $\theta$ we have

$$\pi_\theta(y^* \mid x) = 0 \quad \text{for a particular } y^*.$$

All standard policy-gradient updates (e.g. REINFORCE, PPO, GRPO) take the form

$$\theta' = \theta + \eta \, \nabla_\theta \mathbb{E}_{y \sim \pi_\theta(\cdot \mid x)}\Big[R(x,y) - \beta^{-1} \log \tfrac{\pi_\theta(y \mid x)}{q(y \mid x)}\Big],$$

where $\eta$ is the learning rate. Since the outer expectation is over $y \sim \pi_\theta$, any $y^* \in \mathcal{C}$ with $\pi_\theta(y^* \mid x) = 0$ is never sampled and thus contributes no gradient component. Therefore

$$\pi_{\theta'}(y^* \mid x) = 0,$$

and the support of $\pi_{\theta'}$ remains a subset of that of $q$.

**Conclusion.** By induction, none of the updates can introduce positive probability mass on any $y^* \in \mathcal{C}$ for which $q(y^* \mid x) = 0$. Equivalently,

$$\mathrm{supp}\big(\pi_\theta(\cdot \mid x)\big) \subseteq \mathrm{supp}\big(q(\cdot \mid x)\big),$$

indicating that any correct solution $y^*$ with $q(y^* \mid x) = 0$ remains unreachable by the RLVR policy. $\qquad\square$

**Corollary C.2** (Asymptotic Sampling Upper Bound)**.** *Let* $pass@k_p(x)$ *be the probability that at least one out of $k$ samples $y_i \sim p(\cdot \mid x)$ is correct, i.e.* $pass@k_p(x) = 1 - \big(\mathrm{Pr}_{y \sim p}[R(x, y) = 0]\big)^k$. *Under the conditions of Thm. C.1 and the sampling independence, we have*

$$\limsup_{k \to \infty} pass@k_{\pi_\theta}(x) \leq \limsup_{k \to \infty} pass@k_q(x).$$

*Proof.* From Thm. C.1, support preservation implies $\mathrm{supp}(\pi_\theta(\cdot|x)) \subseteq \mathrm{supp}(q(\cdot|x))$. Thus, for any $y \in C$, $\pi_\theta(y|x) > 0 \implies q(y|x) > 0$.

Define the total mass on correct completions by

$$\pi_\theta(C) = \Pr_{y \sim \pi_\theta}[R(x, y) = 1], \quad q(C) = \Pr_{y \sim q}[R(x, y) = 1].$$

Here, samples are assumed independent across the different draws of LLMs; otherwise, we can only assert an upper bound using union bounds. As $k \to \infty$, the `pass@k` success probability becomes

$$\mathrm{pass@}k_{\pi_\theta}(x) = 1 - (1 - \pi_\theta(C))^k \longrightarrow \begin{cases} 1, & \pi_\theta(C) > 0, \\ 0, & \pi_\theta(C) = 0, \end{cases}$$

and similarly for $q$.

Because support preservation ensures that any correct completion reachable under $\pi_\theta$ must also be reachable under $q$,

$$\pi_\theta(C) > 0 \implies q(C) > 0.$$

Hence, the asymptotic success probability satisfies

$$\lim_{k \to \infty} \mathrm{pass@}k_{\pi_\theta}(x) \leq \lim_{k \to \infty} \mathrm{pass@}k_q(x).$$

$\qquad\square$

Theorem C.1 and Corollary C.2 prove that RLVR optimization cannot expand the search space beyond the initial support of the base model. This limitation arises because on-policy sampling means the model updates only from what it already samples — lacking representational coverage means no gradient can ever push probability mass toward truly unseen solutions. Even when rewards provide a clear training signal, RLVR cannot access or discover solutions that the base model assigns zero probability.

This manifests as a trade-off between sharpness and diversity: RLVR can improve `pass@1` by concentrating mass on known high-reward modes but tends to reduce `pass@k` performance for larger $k$, where broader coverage is beneficial. By contrast, the base model may occasionally sample correct answers from its long-tail distribution, giving it a statistical edge under high-$k$ evaluations (Yue et al., 2025a; Liu et al., 2025). This asymptotic upper bound captures a ceiling: no matter how many samples are drawn, the RLVR-trained model cannot exceed the base model's `pass@k` in the limit.

**Theorem C.3** (Empirical Support Preservation)**.** *Assume $\epsilon$ is below the finite-sample detectability threshold used in rollouts. Then, under standard sampling and update procedures with a finite sample budget, we have*

$$\mathrm{supp}_\epsilon\big(\pi_\theta(\cdot \mid x)\big) \subseteq \mathrm{supp}_\epsilon\big(q(\cdot \mid x)\big).$$

*Proof.* Following Zhu et al. (2025), the total update in RLVR training decomposes into

$$\nabla L_{\text{total}} = \nabla L_{\text{PSR}} + \nabla L_{\text{NSR}},$$

where PSR (*positive sample reinforcement*) promotes correct completions and NSR (*negative sample reinforcement*) demotes incorrect ones while redistributing mass proportionally to the current policy. If $y \notin \text{supp}_\epsilon(q)$, then $q(y \mid x) \leq \epsilon$, so $y$ is not $\epsilon$-detectable under the base model and will not be sampled as a positive example. Thus $\nabla L_{\text{PSR}}$ has *no contribution* to $y$, and its probability can only change via $\nabla L_{\text{NSR}}$.

**NSR gradient structure.** We first analyze a single decoding position. At any position with logits $z$ and probabilities $\pi_v$, for a sampled *wrong* token $y_t$ and learning rate $\eta$, the NSR gradient satisfies

$$-\frac{\partial L_{\text{NSR}}}{\partial z_v} \propto \begin{cases} -\pi_{y_t}(1 - \pi_{y_t}), & v = y_t, \\ \pi_v\, \pi_{y_t}, & v \neq y_t, \end{cases} \qquad \Delta z_v = \eta\left(-\frac{\partial L_{\text{NSR}}}{\partial z_v}\right).$$

The softmax policy update under a small NSR step $\Delta z$ has the multiplicative form

$$\pi'(v) \;=\; \frac{\pi(v)\,\exp(\Delta z_v)}{\sum_u \pi(u)\,\exp(\Delta z_u)}.$$

For a correct token $a \neq y_t$, this gives

$$\Delta z_a = \eta\pi(a)\pi(y_t), \qquad \Delta z_{y_t} = -\eta\pi(y_t)(1-\pi(y_t)), \qquad \Delta z_u = \eta\pi(u)\pi(y_t) \geq 0 \;\; (u \notin \{a, y_t\}).$$

Therefore,

$$\frac{\pi'(a)}{\pi(a)} \;=\; \frac{\exp(\Delta z_a)}{\sum_u \pi(u)\exp(\Delta z_u)} \;\leq\; \frac{\exp(\eta\pi(a)\pi(y_t))}{1 - \eta\pi(y_t)^2}.$$

Using $\exp(\eta\pi(a)\pi(y_t)) \leq \exp(\eta\pi(y_t))$ and $1/(1-x) \leq e^{2x}$ for $x \in [0, 1/2]$, we obtain

$$\frac{\pi'(a)}{\pi(a)} \;\leq\; \exp\big(2\eta\pi(y_t)\big).$$

Iterating for $K$ steps yields the token-level bound

$$\pi^{(K)}(a \mid x, y_{<t}) \;\leq\; \pi^{(0)}(a \mid x, y_{<t}) \exp\Big(2\eta \sum_{k=0}^{K-1} \pi^{(k)}(y_t \mid x, y_{<t})\Big) \;\leq\; \pi^{(0)}(a \mid x, y_{<t})\, e^{2\eta K}.$$

**Extension to sequences.** For a full sequence $y^\star = (a_1, \ldots, a_T)$, the autoregressive factorization gives

$$\pi^{(K)}(y^\star \mid x) \;=\; \prod_{t=1}^{T} \pi^{(K)}(a_t \mid x, a_{<t}).$$

Applying the token-level bound at each position $t$,

$$\pi^{(K)}(a_t \mid x, a_{<t}) \;\leq\; \pi^{(0)}(a_t \mid x, a_{<t})\, e^{2\eta K}.$$

Multiplying across all $T$ positions yields

$$\pi^{(K)}(y^\star \mid x) \;\leq\; \pi^{(0)}(y^\star \mid x) \exp(2\eta TK).$$

**Conclusion.** Thus, if a sequence $y$ lies outside $\text{supp}_\epsilon(q)$ so that $\pi^{(0)}(y \mid x) \leq \epsilon$, then even after $K$ NSR updates we have $\pi^{(K)}(y \mid x) \leq \epsilon e^{2\eta TK}$. As $\epsilon \to 0$, multiplying it by any finite constant still yields a vanishingly small quantity; thus, any finite multiple of $\epsilon$ is statistically negligible (i.e., undetectable in practice). Therefore, $\epsilon e^{2\eta TK}$ remains negligible for any finite $K$ and $T$, and

$$\text{supp}_\epsilon(\pi_\theta) \;\subseteq\; \text{supp}_\epsilon(q).$$

$\square$

In this sense, RLVR inherits both the inductive biases and structural limitations of its initialization. Without deliberate intervention or scaling, it remains confined to the functional expressivity of the base model. Our framework formalizes why RLVR often improves sampling efficiency but rarely produces qualitatively new reasoning capabilities.

## C.2 A VARIATIONAL AND SUPPORT-BOUNDED POLICY UPDATE

We now present a unified view of the RLVR objective through the lens of variational inference. This reveals why RLVR is inherently support-bounded: it makes minimal updates to the base distribution while ensuring improved performance.

**Proposition C.4** (KL Projection onto Reward-Consistent Distributions). *Let $\Delta(\mathcal{Y})$ be the probability simplex over the finite output space $\mathcal{Y}$. Define the set of feasible policies that achieve at least a target expected reward $\rho$:*

$$\mathcal{P}_\rho := \{p(y \mid x) \in \Delta(\mathcal{Y}) \mid \mathbb{E}_p[R(x,y)] \geq \rho\}.$$

*Then the solution to the variational problem, $\min_{\pi \in \mathcal{P}_\rho} \mathrm{KL}(\pi \parallel q)$, is the distribution within $\mathcal{P}_\rho$ that is closest in KL divergence to the base model. The optimal policy takes the form:*

$$\pi^*(y \mid x) \propto q(y \mid x) \cdot \exp(\beta R(x,y)),$$

*where $\beta \in \mathbb{R}_{\geq 0}$ is the dual variable associated with the reward constraint and $\beta = 0$ degenerates to the base policy $q$.*

*Proof.* We provide two closely related derivations to illuminate the same optimal solution from both a hard-constrained and a soft-regularized perspective.

**Convexity of Feasible Set $P_\rho$.** We first prove the convexity of $P_\rho$. Recall $P_\rho = \left\{ p \in \Delta(\mathcal{Y}) : \sum_y p(y)R(x,y) \geq \rho \right\}$, where $\Delta(\mathcal{Y})$ denotes the probability simplex over $\mathcal{Y}$.

Take any two distributions $p_1, p_2 \in P_\rho$ and let $\lambda \in [0,1]$. Consider the convex combination

$$p_\lambda := \lambda p_1 + (1-\lambda)p_2.$$

Since $\Delta(\mathcal{Y})$ is convex, we have $p_\lambda \in \Delta(\mathcal{Y})$.

Next, because $p_1, p_2 \in P_\rho$, it follows that

$$\sum_y p_1(y)R(x,y) \geq \rho \quad \text{and} \quad \sum_y p_2(y)R(x,y) \geq \rho.$$

Thus,

$$\sum_y p_\lambda(y)R(x,y) = \lambda \sum_y p_1(y)R(x,y) + (1-\lambda) \sum_y p_2(y)R(x,y) \geq \lambda\rho + (1-\lambda)\rho = \rho.$$

Hence $p_\lambda \in P_\rho$. This shows that $P_\rho$ is convex.

**Convexity, existence, and strong duality.** We then verify the foundational properties of the optimization problem. Recall we wish to solve

$$\min_{\pi \in P_\rho} \mathrm{KL}(\pi \| q), \quad \text{where } P_\rho = \left\{ \pi \in \Delta(\mathcal{Y}) : \sum_y \pi(y)R(x,y) \geq \rho \right\}.$$

The objective function $\mathrm{KL}(\pi \| q)$ is convex in $\pi$ over the probability simplex $\Delta(\mathcal{Y})$, since relative entropy is jointly convex and thus convex in $\pi$ for fixed $q$. The feasible set $P_\rho$ is also convex.

Moreover, if there exists a strictly feasible distribution $\pi$ such that $\sum_y \pi(y)R(x,y) > \rho$, then by *Slater's condition*, strong duality holds. This guarantees that the optimal value of the primal problem equals the optimal value of its Lagrangian dual, and the Karush-Kuhn-Tucker (KKT) conditions characterize the optimal solution. In typical applications—where $q$ arises from softmax-based models with full support—such strictly feasible distributions exist, ensuring that our subsequent Lagrangian approach is valid.

**1) Hard-constrained formulation (projection perspective).** Consider the optimization problem:

$$\min_\pi \text{KL}(\pi \| q) \quad \text{s.t.} \quad \mathbb{E}_\pi[R(x,y)] \geq \rho, \quad \sum_y \pi(y \mid x) = 1, \quad \pi(y \mid x) \geq 0.$$

Using the method of Lagrange multipliers, the Lagrangian is:

$$\mathcal{L}(\pi, \beta, \lambda) = \sum_y \pi(y \mid x) \log \frac{\pi(y \mid x)}{q(y \mid x)} - \beta \left( \sum_y \pi(y \mid x) R(x,y) - \rho \right) + \lambda \left( \sum_y \pi(y \mid x) - 1 \right).$$

Here, we compute the derivative concerning $\pi(y \mid x)$ for fixed multipliers, thereby finding the stationary points of the Lagrangian. Specifically, we take derivative with respect to $\pi(y \mid x)$ and set it to zero:

$$\log \frac{\pi(y \mid x)}{q(y \mid x)} + 1 - \beta R(x,y) + \lambda = 0.$$

Solving for $\pi$ yields:

$$\pi(y \mid x) \propto q(y \mid x) \cdot \exp(\beta R(x,y)).$$

**2) Soft-regularized formulation (dual perspective).** Alternatively, assume RLVR solves the entropy-regularized objective

$$\pi_\theta = \arg\max_{\pi \ll q} \mathbb{E}_{y \sim \pi}[R(x,y)] - \beta^{-1} \text{KL}(\pi \| q),$$

for some inverse temperature parameter $\beta > 0$. Here, the constraint $\pi \ll q$ denotes that $\pi$ is absolutely continuous with respect to $q$, meaning $\pi(y \mid x) > 0$ only if $q(y \mid x) > 0$.[1] The objective is equivalent to the following minimization:

$$\pi_\theta = \arg\min_{\pi \in \Delta(\mathcal{Y})} \text{KL}(\pi \| q) - \beta \mathbb{E}_{y \sim \pi}[R(x,y)].$$

The Lagrangian becomes

$$\mathcal{L}(\pi, \lambda) = \sum_{y \in \mathcal{Y}} \pi(y) \log \frac{\pi(y)}{q(y)} - \beta \sum_{y \in \mathcal{Y}} \pi(y) R(x,y) + \lambda \left( \sum_{y \in \mathcal{Y}} \pi(y) - 1 \right),$$

where $\lambda \in \mathbb{R}$ is the Lagrange multiplier enforcing the normalization constraint.

Taking the derivative with respect to $\pi(y)$ and setting it to zero:

$$\frac{\partial \mathcal{L}}{\partial \pi(y)} = \log \frac{\pi(y)}{q(y)} + 1 - \beta R(x,y) + \lambda = 0.$$

Solving for $\pi(y)$ gives:

$$\pi(y) = q(y) \cdot \exp\left( \beta R(x,y) - \lambda - 1 \right).$$

Letting the normalization constant be:

$$Z = \sum_{y' \in \mathcal{Y}} q(y') \cdot \exp(\beta R(x,y')),$$

we absorb constants into $Z$ and write:

$$\pi_\theta(y \mid x) = \frac{q(y \mid x) \cdot \exp(\beta R(x,y))}{Z}.$$

Both derivations recover the same *exponentially tilted* distribution that emphasizes high-reward completions relative to the base model. In the hard-constrained view, $\beta$ is a Lagrange multiplier tuned to meet the target reward $\rho$; in the soft-regularized view, $\beta$ sets the strength of the trade-off between reward and divergence. This completes the constructive proof of Prop. C.4.

$\square$

---

[1] Formally, absolute continuity $\pi \ll q$ ensures that the KL divergence $\text{KL}(\pi \| q)$ is finite. If $\pi$ assigns positive mass to any output that $q$ assigns zero probability, the divergence becomes infinite. This condition also enforces support preservation: $\text{supp}(\pi) \subseteq \text{supp}(q)$.

Notably, by standard convex duality, this solution also arises as the optimizer of the entropy-regularized problem $\max_{\pi \ll q} \; \mathbb{E}_\pi\big[R(x,y)\big] - \frac{1}{\beta}\,\mathrm{KL}\big(\pi \,\|\, q\big)$, which softens the constraint into a penalty. Thus, RLVR can be interpreted either as a *hard projection* onto the closest distribution achieving the reward target, or as a *soft trade-off* that balances expected reward with closeness to the base model. Similar exponential tilting policy improvement oracles have been analyzed in the context of KL-regularized contextual bandits and RLHF (Zhao et al., 2024), though their focus is on sample complexity under coverage.

**KL-Free Limit.** A relevant special case is the KL-free limit, where explicit KL regularization is removed ($\beta \to \infty$) (Wei et al., 2023; Yu et al., 2025; Luo et al., 2025a; Yue et al., 2025b). In this regime, RLVR simplifies to a hard-filtered projection onto reward-maximizing completions.

**Corollary C.5** (KL-Free Projection). *In the limit $\beta \to \infty$, the RLVR update converges to the renormalized restriction of the base model to the correct completion set:*

$$\lim_{\beta \to \infty} \pi_\beta(y \mid x) \;=\; \frac{q(y \mid x)\,\mathbf{1}\{y \in \mathcal{C}\}}{\sum_{y' \in \mathcal{C}} q(y' \mid x)}.$$

*Proof.* Since $R(x,y) \in \{0,1\}$, we have

$$\exp\big(\beta R(x,y)\big) = \begin{cases} e^\beta & \text{if } R(x,y) = 1, \\ 1 & \text{if } R(x,y) = 0. \end{cases}$$

Thus, the RLVR distribution becomes

$$\pi_\beta(y \mid x) = \frac{q(y \mid x)\,\exp\big(\beta R(x,y)\big)}{Z_\beta(x)} ni = \frac{q(y \mid x)\,\big[e^\beta \mathbf{1}\{R(x,y)=1\} + \mathbf{1}\{R(x,y)=0\}\big]}{Z_\beta(x)},$$

where

$$Z_\beta(x) = e^\beta \sum_{y':R(x,y')=1} q(y' \mid x) + \sum_{y':R(x,y')=0} q(y' \mid x).$$

As $\beta \to \infty$, the term with $e^\beta$ dominates whenever there exists at least one $y$ with $R(x,y) = 1$. Thus

$$Z_\beta(x) \approx e^\beta \sum_{y' \in \mathcal{C}} q(y' \mid x).$$

Similarly, in the numerator, we have

$$q(y \mid x)\,\exp\big(\beta R(x,y)\big) = \begin{cases} q(y \mid x)\,e^\beta & \text{if } y \in \mathcal{C}, \\ q(y \mid x) & \text{otherwise.} \end{cases}$$

Dividing by $Z_\beta(x)$ and taking $\beta \to \infty$, the probabilities assigned to $y$ with $R(x,y) = 0$ vanish:

$$\pi_\beta(y \mid x) \approx \begin{cases} \dfrac{q(y \mid x)\,e^\beta}{e^\beta \sum_{y' \in \mathcal{C}} q(y' \mid x)} = \dfrac{q(y \mid x)}{\sum_{y' \in \mathcal{C}} q(y' \mid x)} & \text{if } y \in \mathcal{C}, \\ 0 & \text{otherwise.} \end{cases}$$

Thus we obtain

$$\lim_{\beta \to \infty} \pi_\beta(y \mid x) = \frac{q(y \mid x)\,\mathbf{1}\{y \in \mathcal{C}\}}{\sum_{y' \in \mathcal{C}} q(y' \mid x)},$$

$\square$

Together, Prop. C.4 and Cor. C.5 illustrate a continuum of RLVR behaviors—from softly regularized reweighting (small $\beta$) to sharply constrained filtering (large $\beta$). Even in the KL-free limit, updates remain fundamentally anchored to the base model's distribution, preserving relative probabilities within the reward-consistent subset. Consequently, while this projection ensures stable, efficient updates, it inherently limits RLVR's exploratory capacity. As established in Thm. C.1, RLVR remains confined to the initial support of the base model unless explicit mechanisms introduce meaningful probability mass to new regions. Thus, the variational interpretation clarifies RLVR's strengths in improving precision and efficiency within existing capabilities, alongside its limitations in fundamentally expanding model reasoning.

## C.3 ENTROPY–REWARD TRADE-OFF: PRECISION AT THE COST OF ANSWER DIVERSITY

Another structural property of RLVR is its tendency to systematically reduce the entropy of the answer distribution. This behavior arises naturally from reward optimization, which statistically favors sharper distributions concentrated on high-reward completions. While such entropy reduction is beneficial in domains like board games or math—where precision is paramount—it may also suppress valuable diversity in contexts that benefit from broader coverage or multiple valid outputs, such as story or dialogue generation (Chen et al., 2023) and coding copilots (Peng et al., 2023).

**Theorem C.6** (Entropy Reduction and Precision–Coverage Trade-off). *Assume a finite output space $\mathcal{Y}$ and define the Shannon entropy of a distribution as $\mathcal{H}[p] := -\sum_{y \in \mathcal{Y}} p(y \mid x) \log p(y \mid x)$. Then the following statements hold:*

(a) ***Entropy reduction.*** *Any RLVR update $\pi_\theta$ satisfies*

$$\mathcal{H}[\pi_\theta] \ \leq \ \mathcal{H}[q],$$

*with equality only if the reward is constant on the support of $q$.*

(b) ***Trade-off with coverage.*** *Lower entropy increases sampling precision for small budgets, but for large $k$, reduces the diversity of explored outputs—potentially missing alternative correct completions.*

*Proof.* **(a) Entropy reduction.** Consider the exponentially tilted distribution

$$\pi_\theta(y \mid x) = \frac{q(y \mid x) \exp(\beta R(x, y))}{Z}, \quad \text{with} \quad Z = \sum_{y \in \mathcal{Y}} q(y \mid x) \exp(\beta R(x, y)).$$

By standard properties of KL divergence,

$$\mathrm{KL}(\pi_\theta \| q) = \sum_y \pi_\theta(y \mid x) \log \frac{\pi_\theta(y \mid x)}{q(y \mid x)} \ \geq \ 0.$$

Rearranging gives

$$\mathcal{H}[\pi_\theta] = \mathcal{H}[q] - \mathrm{KL}(\pi_\theta \| q) \ \leq \ \mathcal{H}[q].$$

Thus, any such RLVR update decreases entropy relative to the base distribution, unless the reward is constant (in which case $\pi_\theta = q$).

**(b) Trade-off with diversity at different sampling budgets.** The RLVR-trained policy sharpens the probability mass around high-reward completions. Explicitly,

$$\pi_\theta(y \mid x) \ \propto \ q(y \mid x) \exp(\beta R(x, y)),$$

where $\beta > 0$ controls concentration.

- **Small sampling budgets ($k = 1$):** The increased probability on high-reward outputs generally improves single-shot success rates. Formally,

$$\texttt{pass@1}_{\pi_\theta}(x) = \sum_{y:R(x,y)=1} \pi_\theta(y \mid x) \ > \ \sum_{y:R(x,y)=1} q(y \mid x) = \texttt{pass@1}_q(x),$$

provided the reweighting boosts correct completions relative to incorrect ones.

- **Large sampling budgets ($k \gg 1$):** However, reduced entropy leads to concentration on fewer modes. As $\beta$ grows, $\pi_\theta$ may collapse onto a narrow subset of correct completions, neglecting other valid solutions accessible under the more dispersed $q$. Thus,

$$\limsup_{k \to \infty} \texttt{pass@}k_{\pi_\theta}(x) < \limsup_{k \to \infty} \texttt{pass@}k_q(x),$$

under typical conditions of entropy reduction and selective mass shifting.

- **Loss of tail coverage:** In particular, if there exist rare but correct completions that have small mass under $q$ but are further downweighted (or eliminated) by the tilting, then the total mass on correct completions can decrease:

$$\pi_\theta(C) < q(C), \quad C = \{y : R(x, y) = 1\}.$$

This restricts the long-run probability of recovering diverse solutions via large $k$ sampling.

**Conclusion.** This establishes a trade-off: RLVR improves sampling efficiency by concentrating probability on high-reward outputs (increasing `pass@1`), but this comes at the cost of reduced entropy and narrower exploration of the solution space (potentially lowering `pass@k` for large $k$). Empirical studies confirm this phenomenon in settings like code generation and symbolic reasoning, where many semantically distinct correct completions exist. $\qquad\square$

This trade-off underpins RLVR's empirical strengths in tasks with narrowly defined optimal solutions, such as mathematical proofs or tactical game endgames (where precision is paramount), while also emphasizing the need for explicit diversity mechanisms in more open-ended domains, such as code generation, creative writing (Feizi et al., 2023; Ding et al., 2024), or brainstorming (Chang & Li, 2025). Importantly, entropy reduction is not inherently undesirable: when a task admits a unique correct solution, lower answer-level entropy simply reflects desirable convergence. Importantly, even in multi-solution domains, concentrating mass on a narrower set may still be desirable under constrained compute budgets. However, our results show that entropy reduction can still lead to empirical support shrinkage even in predominantly single-solution domains like math, where RLVR sometimes fails to recover valid completions still accessible to the more diverse base model. This highlights that entropy-induced narrowing is a general phenomenon, not limited to multi-solution tasks, underscoring the broader need for explicit exploration or diversity-promoting strategies.

### C.4 ESTIMATING THE SAMPLING THRESHOLD $\epsilon$ FROM `PASS@K`

We provide a statistical analysis of the threshold $\epsilon$ in the `pass@k` sampling. Suppose we sample $k$ times from a model $\pi(\cdot \mid x)$, and let $y^* \in \mathcal{C}$ be a correct completion with unknown probability $p = \pi(y^* \mid x)$. If $y^*$ is not observed in any of those $k$ samples, we can upper bound $p$ using the following argument.

The probability of *not* sampling $y^*$ in a single trial is $1 - p$, so the probability of missing it in all $k$ independent trials is $(1-p)^k$. To ensure this event occurs with probability at most $\zeta$, we solve:

$$(1-p)^k \le \zeta.$$

Taking logarithms of both sides:
$$k \cdot \log(1-p) \le \log \zeta.$$

Using the inequality $\log(1-p) \le -p$ for $p \in (0,1)$, we get:

$$k \cdot (-p) \ge \log \zeta \quad \Rightarrow \quad p \le \frac{-\log \zeta}{k}.$$

Consequently, if the correct completion $y^*$ is not observed in $k$ samples, then with confidence $1 - \zeta$, its probability satisfies:

$$\boxed{\pi(y^* \mid x) \le \frac{-\log \zeta}{k}.}$$

**Example.** If $k = 8192$ in the math reasoning tasks and we desire 95% confidence (i.e., $\zeta = 0.05$), then

$$\pi(y^* \mid x) \le \frac{-\log(0.05)}{8192} \approx \frac{2.996}{8192} \approx 3.66 \times 10^{-4}.$$

## D THE USE OF LLMS

This study utilized large language models solely for providing assistance with minor language enhancements. All content has undergone human review, verification, and further modification.

Table 13: **Effect of RLVR (DAPO) training on DeepSeek 1.5B across math benchmarks.** `Pass@256` values are percentages. Step = 0 corresponds to the non-RLVR baseline.

| Dataset | Step | Pass@256 RLVR | SRR | NDR | SDS | NSCR | P | E | S | O |
|---|---|---|---|---|---|---|---|---|---|---|
| AIME24 | 0 | 83.33 | - | - | - | - | - | - | - | - |
| | 30 | 80.00 | 0.960 | 0.000 | 0.000 | -0.040 | 24 | 0 | 1 | 5 |
| | 60 | 80.00 | 0.960 | 0.000 | 0.000 | -0.040 | 24 | 0 | 1 | 5 |
| | 90 | 83.33 | 1.000 | 0.000 | 0.000 | 0.000 | 25 | 0 | 0 | 5 |
| | 120 | 80.00 | 0.960 | 0.000 | 0.000 | -0.040 | 24 | 0 | 1 | 5 |
| | 150 | 80.00 | 0.960 | 0.000 | 0.000 | -0.040 | 24 | 0 | 1 | 5 |
| | 180 | 83.33 | 1.000 | 0.000 | 0.000 | 0.000 | 25 | 0 | 0 | 5 |
| | 210 | 76.67 | 0.920 | 0.000 | 0.000 | -0.080 | 23 | 0 | 2 | 5 |
| | 240 | 73.33 | 0.880 | 0.000 | 0.000 | -0.120 | 22 | 0 | 3 | 5 |
| | 270 | 73.33 | 0.880 | 0.000 | 0.000 | -0.120 | 22 | 0 | 3 | 5 |
| | 300 | 73.33 | 0.880 | 0.000 | 0.000 | -0.120 | 22 | 0 | 3 | 5 |
| AIME25 | 0 | 70.00 | - | - | - | - | - | - | - | - |
| | 30 | 70.00 | 0.905 | 0.095 | 0.172 | 0.000 | 19 | 2 | 2 | 7 |
| | 60 | 73.33 | 0.905 | 0.136 | 0.237 | 0.042 | 19 | 3 | 2 | 6 |
| | 90 | 70.00 | 0.905 | 0.095 | 0.172 | 0.000 | 19 | 2 | 2 | 7 |
| | 120 | 73.33 | 0.905 | 0.136 | 0.237 | 0.042 | 19 | 3 | 2 | 6 |
| | 150 | 73.33 | 0.905 | 0.136 | 0.237 | 0.042 | 19 | 3 | 2 | 6 |
| | 180 | 63.33 | 0.905 | 0.000 | 0.000 | -0.095 | 19 | 0 | 2 | 9 |
| | 210 | 66.67 | 0.905 | 0.050 | 0.095 | -0.045 | 19 | 1 | 2 | 8 |
| | 240 | 66.67 | 0.905 | 0.050 | 0.095 | -0.045 | 19 | 1 | 2 | 8 |
| | 270 | 63.33 | 0.905 | 0.000 | 0.000 | -0.095 | 19 | 0 | 2 | 9 |
| | 300 | 66.67 | 0.864 | 0.095 | 0.172 | -0.042 | 18 | 2 | 3 | 7 |
| AMC23 | 0 | 97.50 | - | - | - | - | - | - | - | - |
| | 30 | 97.50 | 1.000 | 0.000 | 0.000 | 0.000 | 39 | 0 | 0 | 1 |
| | 60 | 97.50 | 1.000 | 0.000 | 0.000 | 0.000 | 39 | 0 | 0 | 1 |
| | 90 | 97.50 | 1.000 | 0.000 | 0.000 | 0.000 | 39 | 0 | 0 | 1 |
| | 120 | 97.50 | 1.000 | 0.000 | 0.000 | 0.000 | 39 | 0 | 0 | 1 |
| | 150 | 97.50 | 1.000 | 0.000 | 0.000 | 0.000 | 39 | 0 | 0 | 1 |
| | 180 | 97.50 | 1.000 | 0.000 | 0.000 | 0.000 | 39 | 0 | 0 | 1 |
| | 210 | 97.50 | 1.000 | 0.000 | 0.000 | 0.000 | 39 | 0 | 0 | 1 |
| | 240 | 97.50 | 1.000 | 0.000 | 0.000 | 0.000 | 39 | 0 | 0 | 1 |
| | 270 | 97.50 | 1.000 | 0.000 | 0.000 | 0.000 | 39 | 0 | 0 | 1 |
| | 300 | 97.50 | 1.000 | 0.000 | 0.000 | 0.000 | 39 | 0 | 0 | 1 |
| MATH500 | 0 | 99.20 | - | - | - | - | - | - | - | - |
| | 30 | 98.80 | 0.996 | 0.000 | 0.000 | -0.004 | 494 | 0 | 2 | 4 |
| | 60 | 98.80 | 0.996 | 0.000 | 0.000 | -0.004 | 494 | 0 | 2 | 4 |
| | 90 | 99.20 | 0.998 | 0.002 | 0.004 | 0.000 | 495 | 1 | 1 | 3 |
| | 120 | 99.00 | 0.996 | 0.002 | 0.004 | -0.002 | 494 | 1 | 2 | 3 |
| | 150 | 98.80 | 0.996 | 0.000 | 0.000 | -0.004 | 494 | 0 | 2 | 4 |
| | 180 | 99.00 | 0.998 | 0.000 | 0.000 | -0.002 | 495 | 0 | 1 | 4 |
| | 210 | 98.80 | 0.994 | 0.002 | 0.004 | -0.004 | 493 | 1 | 3 | 3 |
| | 240 | 98.60 | 0.992 | 0.002 | 0.004 | -0.006 | 492 | 1 | 4 | 3 |
| | 270 | 98.40 | 0.992 | 0.000 | 0.000 | -0.008 | 492 | 0 | 4 | 4 |
| | 300 | 97.80 | 0.984 | 0.002 | 0.004 | -0.014 | 488 | 1 | 8 | 3 |
| Minerva | 0 | 62.50 | - | - | - | - | - | - | - | - |
| | 30 | 62.13 | 0.935 | 0.059 | 0.111 | -0.006 | 159 | 10 | 11 | 92 |
| | 60 | 59.93 | 0.929 | 0.031 | 0.059 | -0.040 | 158 | 5 | 12 | 97 |
| | 90 | 61.76 | 0.924 | 0.065 | 0.122 | -0.011 | 157 | 11 | 13 | 91 |
| | 120 | 62.87 | 0.953 | 0.053 | 0.100 | 0.006 | 162 | 9 | 8 | 93 |
| | 150 | 62.50 | 0.947 | 0.053 | 0.100 | 0.000 | 161 | 9 | 9 | 93 |
| | 180 | 59.56 | 0.906 | 0.049 | 0.094 | -0.045 | 154 | 8 | 16 | 94 |
| | 210 | 60.29 | 0.924 | 0.043 | 0.082 | -0.034 | 157 | 7 | 13 | 95 |
| | 240 | 61.40 | 0.941 | 0.042 | 0.080 | -0.017 | 160 | 7 | 10 | 95 |
| | 270 | 59.19 | 0.906 | 0.043 | 0.083 | -0.051 | 154 | 7 | 16 | 95 |
| | 300 | 58.46 | 0.912 | 0.025 | 0.049 | -0.063 | 155 | 4 | 15 | 98 |
| Olympiad | 0 | 88.00 | - | - | - | - | - | - | - | - |
| | 30 | 86.67 | 0.963 | 0.022 | 0.043 | -0.015 | 572 | 13 | 22 | 68 |
| | 60 | 86.81 | 0.960 | 0.027 | 0.053 | -0.013 | 570 | 16 | 24 | 65 |
| | 90 | 86.07 | 0.961 | 0.017 | 0.034 | -0.022 | 571 | 10 | 23 | 71 |
| | 120 | 86.52 | 0.961 | 0.022 | 0.044 | -0.016 | 571 | 13 | 23 | 68 |
| | 150 | 85.48 | 0.955 | 0.017 | 0.034 | -0.028 | 567 | 10 | 27 | 71 |
| | 180 | 84.00 | 0.944 | 0.011 | 0.021 | -0.045 | 561 | 6 | 33 | 75 |
| | 210 | 82.81 | 0.926 | 0.016 | 0.032 | -0.058 | 550 | 9 | 44 | 72 |
| | 240 | 82.81 | 0.931 | 0.011 | 0.021 | -0.058 | 553 | 6 | 41 | 75 |
| | 270 | 81.33 | 0.912 | 0.013 | 0.025 | -0.075 | 542 | 7 | 52 | 74 |
| | 300 | 79.70 | 0.897 | 0.009 | 0.018 | -0.093 | 533 | 5 | 61 | 76 |

Table 14: Models evaluated in the entropy analysis.

| Name | Full Model Name | Type | Parameters |
|------|-----------------|------|------------|
| DeepSeek-1.5B | `DeepSeek-R1-Distill-Qwen-1.5B` | Base | 1.5B |
| ProRL-1.5B | `Nemotron-Research-Reasoning-Qwen-1.5B` | RLVR | 1.5B |
| DeepSeek-7B | `DeepSeek-R1-Distill-Qwen-7B` | Base | 7B |
| AceReason-7B | `AceReason-Nemotron-7B` | RLVR | 7B |
| Skywork-OR1-7B | `Skywork-OR1-7B` | RLVR | 7B |
| DeepSeek-14B | `DeepSeek-R1-Distill-Qwen-14B` | Base | 14B |
| AceReason-14B | `AceReason-Nemotron-14B` | RLVR | 14B |
| Qwen2.5-32B | `Qwen2.5-32B` | Base | 32B |
| DAPO-32B | `DAPO-Qwen-32B` | RLVR | 32B |

