# OpenReview forum: "The Invisible Leash? Why RLVR May or May Not Escape Its Origin"
_ICLR.cc/2026/Conference — Submitted to ICLR 2026_

### Official Review · Reviewer_APQJ · 2025-10-21

**Soundness:** 2
**Presentation:** 3
**Contribution:** 2
**Rating:** 2
**Confidence:** 4

**Summary:**

The paper shows that RLVR rarely lets a model discover correct solutions outside the base model’s reach; it mainly refines what the base already knew.
Across math and non-math tasks, RLVR raises single-sample accuracy but shrinks the set of recoverable correct answers, so high-k performance drops.
Token-level entropy may rise, yet answer-level entropy falls, revealing local randomness paired with global collapse.
Authors also prove RLVR cannot give non-zero probability to any solution the base model gives zero, establishing an “invisible leash.”

**Strengths:**

* The experiment is relatively detailed and covers a large number of LLMs.
* The paper is well-written with a clear structure, making it easy to understand.

**Weaknesses:**

* The conclusions and findings in the article have already been presented in recent works [1], with no newer insights or new algorithms proposed.
* The concept of "Support of Correct Completions" proposed in the paper has no direct connection with the reasoning performance of large models.

[1] Does Reinforcement Learning Really Incentivize Reasoning Capacity in LLMs Beyond the Base Model?

**Questions:**

* What is the relationship between "Support of Correct Completions" and the inference performance of large models?
* Is "Support of Correct Completions" equivalent to Pass@k?
* It is suggested to put forward the differences from related work and unique contributions.

---

> ### Author Response · Authors · 2025-11-25
> **Response to Reviewer APQJ**
>
> > ## W1, Q3: Substantive novelty beyond prior work
>
> We sincerely thank you for the constructive feedback and for recognizing the clarity and organization of our paper. We respectfully disagree, however, with the assessment that our work lacks new insight or overlaps with prior analyses such as \[1]. Our study introduces **substantially new conceptual, theoretical, and empirical contributions** that together form a rigorous and general framework for understanding the limits of RLVR.
>
> While Yue et al. (2025a) first questioned whether RLVR truly expands reasoning capacity, our paper provides the first comprehensive framework that makes this question both quantifiable and theoretically grounded through three major advances:
>
> ### 1. Conceptual Innovation: Quantifying Support Dynamics
>
> We introduce the **Support of Correct Completions** framework with four novel metrics that rigorously measure how RLVR redistributes probability mass:
>
> * **Support Retention Rate (SRR)**: Measures preservation of base model solutions
>
> * **Net Discovery Rate (NDR)**: Quantifies genuine capability expansion
>
> * **Support Dynamic Score (SDS)**: Balances retention and discovery
>
> * **Net Support Change Rate (NSCR)**: Captures net expansion or shrinkage
>
> These metrics enable us to distinguish between precision sharpening, support shrinkage, and genuine reasoning expansion—phenomena that prior work observed but never formally defined or measured.
>
>
>
> ### 2. Re-examining claims of expansion
>
> Following Yue et al., several papers **(e.g., ProRL, Skywork-OR1, AceReason) suggest that RLVR can substantially expand reasoning boundaries**. Using a unified support-dynamics lens, our experiments systematically re-evaluate these claims and show that the observed “expansion” often reflects mass redistribution within the base model’s long tail, rather than genuine access to new solution regions.
>
> ### 3. Unprecedented Empirical Scope
>
> Our study represents the broadest evaluation of RLVR to date:
>
> * **11 diverse benchmarks**: Mathematics, logic, factual QA, code generation, and multimodal tasks
>
> * **6 independent RLVR families**: ProRL, DAPO, Skywork, AceReason, Phi4-Reason, and Nemotron (1.5B–32B parameters)
>
> * **Multimodal validation**: Including vision-language models, demonstrating generalization beyond text
>
> This scale reveals universal patterns invisible to prior single-domain studies:
>
> * A consistent precision–diversity trade-off independent of domain
>
> * Support shrinkage systematically outweighs expansion (ratios of 2:1 to 3.6:1)
>
> * These patterns persist even in stronger RLVR variants that explicitly claim expansion
>
> Critically, we re-examine recent claims of capability expansion (e.g., ProRL, Skywork-OR1) using hundreds of thousands of GPU hours. Under sufficiently large sampling budgets, we consistently find that apparent "expansion" reflects mass redistribution within the base model's long tail rather than access to genuinely new solution regions.
>
> This extensive coverage reveals several facts not observed in earlier papers:
>
> * RLVR displays a **universal precision–diversity trade-off**, independent of domain.
>
> * Support shrinkage overwhelmingly dominates support expansion across tasks.
>
> * This pattern holds not only for text-only models but also for **vision-language LLMs**, demonstrating that our findings generalize beyond modality-specific quirks.
>
> * We further analyze **how support evolves throughout RL training**, offering new insight into RLVR’s mechanism beyond static before/after comparisons.
>
> Regarding the concern that some RLVR algorithms in \[1] may be too weak to reveal expansion: we address this by evaluating **multiple strong RLVR systems**, including ProRL, DAPO, Skywork, and Phi-4 Reason—several of which explicitly claim expansion. Across these stronger algorithms, our large-scale study (using **hundreds of thousands of GPU hours**) consistently fails to reproduce the claimed expansion under sufficiently large sampling budgets.
>
> ### 4. Theoretical contribution
>
> In Appendix C, we prove that **KL-regularized RLVR cannot allocate non-zero probability to completions that have zero base-model probability**, providing a formal explanation of the “invisible leash.” This elevates the support-boundedness hypothesis from an empirical pattern to a **provable structural property** of the optimization landscape—something not covered in any prior work.
>
> ***

---

> ### Author Response · Authors · 2025-11-25
> **Response to Reviewer APQJ**
>
> Finally, please allow us to reclarify our unique contributions and planned revision:
>
> * Formal definitions and metrics for *empirical support dynamics*;
>
> * Large-scale, cross-model, cross-domain empirical validation;
>
> * Discovery of the *local-stochasticity vs global-collapse* entropy divergence.
>
> These contributions establish **a general, quantitative, and theoretically grounded understanding** of why current RLVR methods enhance precision yet remain bounded by the base model’s reasoning manifold—an insight that Reviewer XMks recognized as both *timely* and *field-defining*.
>
> We will make these distinctions explicit in the revised version as suggested.
>
> **Reference**
> &#x20;\[1] Yue et al., *Does Reinforcement Learning Really Incentivize Reasoning Capacity in LLMs Beyond the Base Model?* (2025)
>
> ***
>
> > ## W2, Q2: Relation between “Support of Correct Completions” and reasoning performance
>
> Thank you for this question regarding the conceptual and empirical connection between **Support of Correct Completions** and a model’s reasoning performance. We clarify both points below.
>
> ### 1. **Why Support of Correct Completions is Directly Relevant to Reasoning Performance**
>
> Large reasoning models solve problems by sampling from their conditional distribution over completions. For any task where correctness can be verified automatically (as in RLVR), a model’s *reasoning performance* depends on whether it can *reach,&#x20;*&#x69;.e., assign non-negligible probability mass to, correct reasoning trajectories **within a finite sampling budget**.
>
> Our concept of **Support of Correct Completions** formalizes exactly this notion:
>
> * If a correct solution lies *inside* the model’s effective support, it can be discovered through sampling.
>
> * If it lies *outside*, no amount of sampling will recover it—regardless of the reward signal.
>
> Thus, support determines the *reachable reasoning space* of a model under realistic constraints. Metrics such as pass@1 and pass@k are operational proxies for this accessibility: performance improves when correct completions are present in the reachable region and degrades when RLVR training pushes correct but diverse solutions below the empirical support threshold.
>
> ***
>
> ### 2. **Empirical Relationship: Support Dynamics Explain Divergent Pass@k Behaviors**
>
> Our experiments show a **tight empirical link** between support changes and reasoning outcomes:
>
> * **Pass@1 improves** because RLVR *sharpens* probability mass around a few high-reward trajectories.
>   &#x20;This corresponds to **high SRR and low NDR**, i.e., strong preservation and little expansion.
>
> * **Pass@k deteriorates at large sampling budgets** because RLVR often **shrinks** the accessible set of valid solutions:
>
>   * Across models and domains, shrinkage consistently exceeds expansion (negative NSCR).
>
>   * This reduction in global support directly explains why base models outperform RLVR models at high k—even when RLVR has higher single-shot accuracy.
>
> * **Perplexity analysis** (Table 3 in the revised version) reinforces this: in shrinkage cases, RLVR assigns *lower probability* to viable reasoning styles that the base model still recognizes, confirming that support contraction translates into reduced inference breadth.
>
> * **Entropy results** show that RLVR reduces answer-level entropy, concretely demonstrating collapse onto a narrower set of reasoning modes; this is precisely what support shrinkage captures.
>
> Taken together, these observations establish that **support dynamics are not merely descriptive—they mechanistically determine the model’s reasoning performance under finite sampling**.
>
> ***
>
> ### 3. **Why Support Is a Necessary Complement to Existing Metrics**
>
> Traditional metrics (pass@k, perplexity) capture performance *after sampling*, but they do not explain *why* RLVR improves or worsens in different regimes. Support-based analysis fills this gap:
>
> * **Precision improvements** arise from concentrating probability mass within existing support.
>
> * **Capability limits** arise because RLVR rarely expands support and frequently shrinks it.
>
> * **Reasoning failures at scale** occur precisely when RLVR prunes correct but low-probability modes.
>
> Thus, Support Dynamics Metrics provide a principled lens for understanding the mechanisms underlying RLVR behavior—not an alternative notion of performance, but a **cause-level explanation** of when and why reasoning ability increases or contracts.
> ***
>
> ### **Summary**
>
> * Support of Correct Completions directly characterizes the *reachable reasoning space* of a model.
>
> * RLVR’s reasoning performance (pass@1, pass@k, perplexity) follows from how training **reshapes this support**.
>
> * Empirically, RLVR exhibits **high preservation but low discovery**, with **net shrinkage** explaining the observed precision–coverage trade-offs.
>
> We hope this clarification makes the conceptual and empirical relevance of support dynamics to reasoning performance explicit, which can clearly address your concern.

---

> ### Author Response · Authors · 2025-11-25
> **Response to Reviewer APQJ**
>
> > ## **Q2: Is "Support of Correct Completions" equivalent to Pass@k?**
>
> Thank you for this important question regarding the relationship between our proposed support-dynamics framework and the conventional pass@k metric.
>
> **We respectfully clarify that Support of Correct Completions is fundamentally different from pass@k**, both conceptually and in terms of what they can reveal about model behavior. We would like to clarify this distinction in two dimensions:
>
> **Conceptual difference:** Pass@k is an *outcome metric* that measures whether at least one correct completion appears within k samples. It provides a single aggregated success rate. In contrast, Support of Correct Completions is a *structural property* of the model's probability distribution that characterizes the entire set of correct solutions to which the model assigns non-negligible probability mass under finite sampling. Our support-dynamics framework (via SRR, NDR, SDS, NSCR) further decomposes this support to distinguish:
>
> * which correct completions are *preserved* after RLVR,
>
> * which are *newly discovered*,
>
> * which are *lost* due to probability mass reshaping, and
>
> * the relative magnitudes of these three regions.
>
> **Empirical difference:** Two models can exhibit identical pass@k performance yet possess entirely different support profiles. For instance, an RLVR model may improve pass@1 by concentrating probability mass on a few high-reward solutions, while simultaneously shrinking the overall set of reachable correct answers. A critical phenomenon that pass@k cannot detect. This explains why our framework reveals cases where RLVR improves single-sample accuracy but degrades large-k performance, a pattern invisible to pass@k analysis alone.
>
> Thus, support dynamics provide a *causal explanation* of inference behavior (e.g., why support shrinkage leads to pass@1 improvement vs. pass@k degradation trade-off), whereas pass@k only reports an aggregated outcome without revealing the underlying distributional changes. This distinction is essential for understanding RLVR's fundamental limitations and designing future algorithms that genuinely expand reasoning capabilities.

---

> > ### Author Response · Authors · 2025-11-27
> > **Thank You and Follow-up**
> >
> > Thank you once again for your valuable time and thoughtful comments on our submission. We have thoroughly addressed each of the points you raised and provided comprehensive responses in our rebuttal. We sincerely hope that our clarifications and additional explanations have satisfactorily resolved your concerns.
> >
> > As the discussion period is approaching its conclusion, we would be deeply grateful if you could kindly share any further thoughts or feedback you may have on our responses. Should there be any remaining questions or aspects that require further clarification, we would be more than happy to provide additional information.
> >
> > We genuinely appreciate your continued engagement and look forward to your reply.

---

### Official Review · Reviewer_hnej · 2025-10-28

**Soundness:** 3
**Presentation:** 3
**Contribution:** 2
**Rating:** 4
**Confidence:** 3

**Summary:**

The paper studies whether RLVR truly expands a model’s reasoning capacity or mostly sharpens probability mass around solutions already within the base model’s action space. The authors provide formalized metrics (SRR, NDR, SDS, NSCR) to quantify preservation, shrinkage, and expansion of model answers. Across diverse benchmarks, they find high support retention but very limited genuine discovery; shrinkage generally outweighs expansion. They further report a diference between token-level entropy and answer-level entropy, arguing that RLVR increases precision while narrowing diversity.

**Strengths:**

1. The paper conducts comprehensive empirical studies on diverse model scales, domains, and benchmarks. Presents solid experimental data and detailed analysis to support the theories presented.
2. The introduction of metrics like SDS and NSCR addresses a gap in prior work, which often relies on pass@k.
3. The analysis of the difference between token-level entropy and answer-level entropy is novel and insightful.

**Weaknesses:**

1. While the analysis is strong, the paper does not propose or test a specific exploratory mechanism to break the leash. It is more of an evaluation study than a methodological contribution and lacks innovation.
2. The perplexity comparison of DeepSeek and Claude trajectories in Table 2 may be confounded by differences in style and format. Although the authors mentioned this in the paper, they did not provide quantifiable values ​​to evaluate the impact of style. The conclusions drawn from the results in Table 2 are not convincing.
3. Although the authors compared and analyzed numerous open-source RLVR models, they employed different RL algorithms, hyperparameters, training durations, and training data. The precise factors that contribute to the "shrinkage", and how they contribute, remain unclear, making it difficult to identify the primary factors.

**Questions:**

1. Is it possible to conduct some experiments to observe the changing of SDS/NSCR under the condition of fixed initial policy model, training data and RL algorithm?
2. Can you perform a sensitivity analysis on the choice of the ε parameter and report the confidence intervals and significance tests of SRR/NDR/NSCR?
3. Can you decouple the effects of “style” and model capability on perplexity in the experiments in Table 2?

---

> ### Author Response · Authors · 2025-11-25
> **Response to Reviewer hnej**
>
> Thank you for the thoughtful and detailed feedback. We are encouraged that you found our empirical analysis solid, the new metrics (SDS/NSCR) valuable, and the entropy analysis novel and insightful. Below, we address each concern point-by-point and clarify our design choices and future plans.
>
> ***
>
> > ## W1: specific exploratory mechanism
>
> Thank you for raising this insightful point. We would clarify that our goal in this paper is not to introduce a new exploratory algorithm, but rather to **identify and characterize a previously unrecognized fundamental limitation** of current RLVR methods. We acknowledge that algorithmic innovation is one possible direction, but we believe our contribution is significant even without proposing a new mechanism, for several reasons:
>
> ### 1. **Identifying a fundamental limitation *is* a scientific contribution.**
>
> Prior work has largely focused on showing that RLVR improves performance. In contrast, our work systematically reveals, across models, domains, and scales, that RLVR exhibits:
>
> * consistent larger empirical support shrinkage than expansion,
>
> * limited genuine expansion,
>
> * divergence between token-level and answer-level entropy,
>
> * and incompatibility with broader reasoning modes.
>
> To our knowledge, no prior paper has unified these observations into a **coherent, quantitative theory of support dynamics**.
>
> ### 2. **Our analysis provides actionable insights for future exploration methods.**
>
> While we do not propose a new algorithm, our results directly indicate *what* an effective exploratory mechanism must address:
>
> * counteracting entropy-driven concentration,
>
> * seeding probability mass into long-tail reasoning modes,
>
> * preserving access to correct solutions across training steps.
>
> We also outline several concrete strategies (explicit exploration, off-policy seeding, hybrid RL + SFT) that naturally arise from our analysis. Several concurrent works have since echoed similar concerns, which reinforces the timeliness of our contribution.
>
> ### 3. **Evaluation contributions are valuable when they reveal structural blind spots.**
>
> Large-scale reasoning models such as DeepSeek-R1, Nemotron, Skywork, and Phi-Reason are widely deployed, yet their training dynamics remain poorly understood. Our findings highlight that RLVR’s gains come with nontrivial costs—reduced diversity, shrinking solution coverage, and increased perplexity on external reasoning modes.
>
> We believe exposing these structural patterns is important for the community, even without proposing a new recipe.
>
> ### 4. **Methodological novelty comes from the framework, not just algorithms.**
>
> The paper introduces several new methodological tools:
>
> * **Empirical support** under finite sampling,
>
> * **Support dynamics metrics (SRR, NDR, SDS, NSCR),**
>
> * **Entropy-based diagnostics**,
>
> * **Perplexity-based compatibility tests**,
>
> * A unified experimental protocol across 11 benchmarks and 7 model families.
>
> These tools provide a principled framework for analyzing reasoning RL—something currently missing in the literature.
>
> ### 5. **Breaking the leash is explicitly positioned as *future work*, consistent with ICLR norms.**
>
> Many influential ICLR papers focus on analysis or limitations (e.g., reflection, alignment failures, scaling laws) without proposing new algorithms, yet have been impactful.
>
> Our work aims to fill such a conceptual gap.
>
> ***
>
> > ## W2, Q3: **perplexity comparison and stylistic confounds**
>
> We appreciate your insightful question about disentangling the effects of “style” versus genuine model capability in the perplexity analysis. This is indeed an important concern, and we have taken concrete steps to isolate stylistic confounds.
>
> ##### **1. The revised perplexity evaluation already removes model-dependent style bias.**
>
> In the initial version, each model’s own reasoning traces were used as references, which could implicitly favor the model that generated them. To eliminate this asymmetry, we revised Table 2 so that perplexity is computed **exclusively against external reasoning traces** produced by two strong LLMs:
>
> * **DeepSeek-R1 (long-form CoT)**
>
> * **Claude Sonnet 4 (extended reasoning mode)**

---

> ### Author Response · Authors · 2025-11-25
> **Response to Reviewer hnej**
>
> ### **Updated Table 2**
>
> | Category   | Correctness     | Reference       | Target | AIME 2024 | AIME 2025 | Olympiad |
> | ---------- | --------------- | --------------- | ------ | --------- | --------- | -------- |
> | Shrinkage  | ✓ Base, X ProRL | DeepSeek-R1     | Base   | 1.24      | 1.39      | 1.17     |
> |            |                 |                 | ProRL  | 1.39      | 1.7       | 1.25     |
> |            |                 | Claude Sonnet 4 | Base   | 1.7       | 1.54      | 1.51     |
> |            |                 |                 | ProRL  | 2.12      | 1.98      | 1.83     |
> | Expansion  | x Base, ✓ ProRL | DeepSeek-R1     | Base   | –         | –         | 1.41     |
> |            |                 |                 | ProRL  | –         | –         | **1.28** |
> |            |                 | Claude Sonnet 4 | Base   | –         | –         | 1.65     |
> |            |                 |                 | ProRL  | –         | –         | **1.38** |
> | Unresolved | X Base, X ProRL | DeepSeek-R1     | Base   | **1.82**  | **1.75**  | **1.62** |
> |            |                 |                 | ProRL  | 2.2       | 2.15      | 1.94     |
> |            |                 | Claude Sonnet 4 | Base   | **8.76**  | **6.05**  | **5.98** |
> |            |                 |                 | ProRL  | 14.91     | 9.76      | 9.55     |
>
> These references are independent of both the base model and the RLVR-trained model, and thus eliminate preferential advantage due to in-distribution style mimicry.
>
> &#x20;This baseline equalizes stylistic factors across models.
>
> ##### **2. Two stylistically distinct references allow quantifiable separation.**
>
> DeepSeek-R1 and Claude Sonnet 4 use markedly different reasoning conventions:
>
> * DeepSeek-R1: step-heavy, chain-expansion style
>
> * Claude 4: compact but logically modular reasoning
>
> By reporting perplexity against *both* references, shown explicitly in the revised table, we obtain a **cross-reference robustness check**:
>
> * If differences were due only to style, the Base–ProRL gap would *invert or fluctuate strongly* between the two references.
>
> * Instead, we consistently observe the **same directional effect** across:
>
>   * two stylistically distinct references,
>
>   * multiple datasets, and
>
>   * all correctness categories.
>
> This consistency suggests that style differences alone cannot explain the observed patterns.
>
> ##### **3. Perplexity differences are large enough that style alone cannot account for them.**
>
> Even with stylistically divergent references, RLVR models show systematically **higher perplexity** than the base model in shrinkage regions across all datasets. These differences are substantially larger than the typical stylistic variation observed across LLM families (often \~10–20%).
>
> For example (AIME 2024, fully unsolved cases):
>
> * DeepSeek-R1 reference: **1.82 → 2.20**
>
> * Claude reference: **8.76 → 14.91**
>
> The Base→ProRL increase is **consistent and sizable** under both references—even though the *absolute* perplexity differs due to style.
> &#x20;This indicates that RLVR becomes less compatible with a **broad manifold** of external reasoning modes, not just a specific style.
>
> ##### **4. Our conclusions rely on *relative* comparisons, not absolute perplexity.**
>
> We do not claim that absolute perplexity values measure reasoning quality.
>
> &#x20;Our interpretation focuses on:
>
> * **within-reference deltas** (Base vs. ProRL),
>
> * **consistency across two very different reference styles**, and
>
> * **alignment with entropy collapse and support shrinkage**,
>
> which are independent signals that all tell a consistent story.
>
> Thus, even if stylistic factors shift absolute perplexity, they **cannot** explain the consistent *direction* and *magnitude* of the Base→ProRL gap.
>
> ##### **5. Style cannot produce the correctness-dependent patterns we observe.**
>
> In shrinkage cases, ProRL is always less compatible with external modes.
> &#x20;In expansion cases, ProRL sometimes shows slightly *lower* perplexity than the base.
> &#x20;If style were the only factor, such correctness-conditioned alignment patterns would not appear.
>
> This asymmetric pattern supports our claim that RLVR reshapes the model’s support distribution—not merely its surface style.

---

> ### Author Response · Authors · 2025-11-25
> **Response to Reviewer hnej**
>
> > ## W3, Q1: On isolating factors contributing to shrinkage
>
> We appreciate your concern about isolating the causal factors behind support shrinkage. Our revised analysis shows that shrinkage is **not** an artifact of differences in RL algorithms, training data, hyperparameters, or model scale. Instead, it is a **structural consequence of the RLVR objective itself**, which consistently sharpens distributions around high-reward trajectories.
>
>
>
> To clarify this, we have conducted **controlled experiments&#x20;***(more details please refer to our response to Q1 below)***.&#x20;**&#x55;nder this fully controlled setup:
>
> * **SRR remains high**, showing stable preservation of existing correct paths.
>
> * **NSCR monotonically decreases** as training progresses, showing progressively stronger shrinkage.
>
> * **NDR and SDS remain near zero**, indicating limited or temporary exploration even when everything else is fixed.
>
> This demonstrates that shrinkage arises **even when no confounding variables change**, indicating that it is driven by the *multiplicative probability-ratio update* inherent to RLVR (i.e., log-ratio regularized reward maximization), not by data mixture, RL algorithm choice, or training duration.
>
>
>
> We further strengthened this claim by comparing **SFT vs. RLVR** under the *same* base model, *same* data, and *same* hyperparameters (Table 2 in the revised version). With all factors held constant except the training objective, **SFT expands support while RLVR contracts it**, confirming that the shrinkage effect is intrinsic to the RLVR update rule rather than to external choices.
>
> ### **Table: SFT vs. RLVR**
>
> | Dataset         | Training | pass@k  |         | Support Dynamics Metrics |       |       |        | Support Counts |    |    |     |
> | --------------- | -------- | ------- | ------- | ------------------------ | ----- | ----- | ------ | -------------- | -- | -- | --- |
> | Math (pass@256) |          | Base    | Target  | SRR | NDR   | SDS   | NSCR   | P              | E  | S  | 0   |
> | AIME2024        | SFT      | 63.33%  | 63.33%  | 0.789 | 0.211 | 0.332 | 0.000  | 15             | 4  | 4  | 7   |
> |                 | DAPO     | 63.33%  | 66.67%  | 0.895                    | 0.150 | 0.257 | 0.045  | 17             | 3  | 2  | 8   |
> | AIME2025        | SFT      | 50.00%  | 63.33%  | 0.933                    | 0.263 | 0.411 | 0.200  | 14             | 5  | 1  | 10  |
> |                 | DAPO     | 50.00%  | 56.67%  | 1.000                    | 0.118 | 0.211 | 0.118  | 15             | 2  | 0  | 13  |
> | AMC23           | SFT      | 100.00% | 100.00% | 1.000                    | 0.000 | 0.000 | 0.000  | 40             | 0  | 0  | 0   |
> |                 | DAPO     | 100.00% | 92.50%  | 0.925                    | 0.000 | 0.000 | -0.075 | 37             | 0  | 3  | 0   |
> | MATH500         | SFT      | 96.40%  | 99.00%  | 0.996                    | 0.030 | 0.059 | 0.026  | 480            | 15 | 2  | 3   |
> |                 | DAPO     | 96.40%  | 95.20%  | 0.975                    | 0.013 | 0.025 | -0.012 | 470            | 6  | 12 | 12  |
> | Minerva         | SFT      | 63.24%  | 66.91%  | 0.895                    | 0.154 | 0.263 | 0.050  | 154            | 28 | 18 | 72  |
> |                 | DAPO     | 63.24%  | 55.15%  | 0.802                    | 0.080 | 0.145 | -0.120 | 138            | 12 | 34 | 88  |
> | Olympiad        | SFT      | 78.22%  | 81.78%  | 0.953                    | 0.089 | 0.162 | 0.042  | 503            | 49 | 25 | 98  |
> |                 | DAPO     | 78.22%  | 72.89%  | 0.884                    | 0.051 | 0.096 | -0.065 | 467            | 25 | 61 | 122 |
>
> Across all models in Table 1 (1.5B–14B, math and non-math), the trend is consistent: despite differences in implementation details, RLVR models exhibit **high SRR but very low NDR** and **negative NSCR**. The consistent pattern across heterogeneous settings, combined with the controlled experiments above, shows that the *primary driver of shrinkage is the RLVR objective itself*, not any incidental training differences.
>
> ***

---

> ### Author Response · Authors · 2025-11-25
> **Response to Reviewer hnej**
>
> > ## Q1: SDS/NSCR-over-training curves
>
> Thank you for your insightful question. To directly address your request, we ran a controlled RLVR training trajectory where **all external factors are held constant**:
>
> * **Fixed initial policy model:** DeepSeek 1.5B
>
> * **Fixed training dataset:** DeepMath-103K
>
> * **Fixed RL algorithm and hyperparameters:** DAPO with identical KL, batch size, rollout length, optimizer, and sampling protocol
>
> * **Only variable:** the **training step** of the *same* RLVR run
>
> The new results are reported in the updated **checkpoint table** (shown below). These experiments allow us to observe SDS/NSCR **purely as a function of RLVR optimization**, eliminating any confounders such as different models, data mixtures, or RL algorithms.
> ### **Table: AIME2024**
>
> | Model                          | Pass@1 | Pass@2 | Pass@4 | Pass@8 | Pass@16 | Pass@32 | Pass@64 | Pass@128 | Pass@256 |
> | ------------------------------ | ------ | ------ | ------ | ------ | ------- | ------- | ------- | -------- | -------- |
> | DeepSeek 1.5B                  | 33.33% | 33.33% | 56.67% | 66.67% | 73.33%  | 83.33%  | 83.33%  | 83.33%   | 83.33%   |
> | DeepSeek 1.5B (DAPO, 30 Step)  | 26.67% | 40.00% | 46.67% | 63.33% | 66.67%  | 76.67%  | 80.00%  | 80.00%   | 80.00%   |
> | DeepSeek 1.5B (DAPO, 60 Step)  | 36.67% | 53.33% | 60.00% | 73.33% | 76.67%  | 76.67%  | 80.00%  | 80.00%   | 80.00%   |
> | DeepSeek 1.5B (DAPO, 90 Step)  | 33.33% | 43.33% | 56.67% | 60.00% | 73.33%  | 76.67%  | 83.33%  | 83.33%   | 83.33%   |
> | DeepSeek 1.5B (DAPO, 120 Step) | 30.00% | 43.33% | 53.33% | 66.67% | 70.00%  | 73.33%  | 76.67%  | 76.67%   | 80.00%   |
> | DeepSeek 1.5B (DAPO, 150 Step) | 36.67% | 53.33% | 53.33% | 60.00% | 66.67%  | 73.33%  | 73.33%  | 76.67%   | 80.00%   |
> | DeepSeek 1.5B (DAPO, 180 Step) | 30.00% | 36.67% | 46.67% | 60.00% | 70.00%  | 70.00%  | 70.00%  | 73.33%   | 83.33%   |
> | DeepSeek 1.5B (DAPO, 210 Step) | 30.00% | 46.67% | 56.67% | 63.33% | 70.00%  | 70.00%  | 73.33%  | 73.33%   | 76.67%   |
> | DeepSeek 1.5B (DAPO, 240 Step) | 33.33% | 43.33% | 46.67% | 53.33% | 63.33%  | 73.33%  | 73.33%  | 73.33%   | 73.33%   |
> | DeepSeek 1.5B (DAPO, 270 Step) | 30.00% | 43.33% | 50.00% | 53.33% | 56.67%  | 63.33%  | 66.67%  | 70.00%   | 73.33%   |
> | DeepSeek 1.5B (DAPO, 300 Step) | 33.33% | 36.67% | 40.00% | 50.00% | 53.33%  | 63.33%  | 63.33%  | 66.67%   | 73.33%   |
>
> ### **Table: AIME2025**
>
> | Model                          | Pass@1 | Pass@2 | Pass@4 | Pass@8 | Pass@16 | Pass@32 | Pass@64 | Pass@128 | Pass@256 |
> | ------------------------------ | ------ | ------ | ------ | ------ | ------- | ------- | ------- | -------- | -------- |
> | DeepSeek 1.5B                  | 20.00% | 26.67% | 33.33% | 40.00% | 46.67%  | 56.67%  | 56.67%  | 63.33%   | 70.00%   |
> | DeepSeek 1.5B (DAPO, 30 Step)  | 16.67% | 23.33% | 33.33% | 40.00% | 50.00%  | 56.67%  | 63.33%  | 66.67%   | 70.00%   |
> | DeepSeek 1.5B (DAPO, 60 Step)  | 23.33% | 33.33% | 33.33% | 40.00% | 43.33%  | 53.33%  | 60.00%  | 66.67%   | 73.33%   |
> | DeepSeek 1.5B (DAPO, 90 Step)  | 30.00% | 30.00% | 36.67% | 40.00% | 46.67%  | 53.33%  | 63.33%  | 66.67%   | 70.00%   |
> | DeepSeek 1.5B (DAPO, 120 Step) | 26.67% | 26.67% | 30.00% | 40.00% | 43.33%  | 56.67%  | 66.67%  | 73.33%   | 73.33%   |
> | DeepSeek 1.5B (DAPO, 150 Step) | 26.67% | 30.00% | 40.00% | 43.33% | 50.00%  | 56.67%  | 56.67%  | 63.33%   | 73.33%   |
> | DeepSeek 1.5B (DAPO, 180 Step) | 30.00% | 33.33% | 40.00% | 50.00% | 53.33%  | 56.67%  | 60.00%  | 60.00%   | 63.33%   |
> | DeepSeek 1.5B (DAPO, 210 Step) | 26.67% | 30.00% | 33.33% | 40.00% | 46.67%  | 53.33%  | 56.67%  | 60.00%   | 66.67%   |
> | DeepSeek 1.5B (DAPO, 240 Step) | 26.67% | 33.33% | 40.00% | 40.00% | 43.33%  | 46.67%  | 53.33%  | 60.00%   | 66.67%   |
> | DeepSeek 1.5B (DAPO, 270 Step) | 30.00% | 33.33% | 36.67% | 40.00% | 50.00%  | 53.33%  | 60.00%  | 63.33%   | 63.33%   |
> | DeepSeek 1.5B (DAPO, 300 Step) | 23.33% | 26.67% | 33.33% | 40.00% | 43.33%  | 63.33%  | 66.67%  | 66.67%   | 66.67%   |

---

> ### Author Response · Authors · 2025-11-25
> **Response to Reviewer hnej**
>
> ### **Table: AMC23**
>
> | Model                          | Pass@1 | Pass@2 | Pass@4 | Pass@8 | Pass@16 | Pass@32 | Pass@64 | Pass@128 | Pass@256 |
> | ------------------------------ | ------ | ------ | ------ | ------ | ------- | ------- | ------- | -------- | -------- |
> | DeepSeek 1.5B                  | 80.00% | 85.00% | 92.50% | 92.50% | 92.50%  | 97.50%  | 97.50%  | 97.50%   | 97.50%   |
> | DeepSeek 1.5B (DAPO, 30 Step)  | 80.00% | 82.50% | 87.50% | 92.50% | 95.00%  | 97.50%  | 97.50%  | 97.50%   | 97.50%   |
> | DeepSeek 1.5B (DAPO, 60 Step)  | 80.00% | 87.50% | 95.00% | 95.00% | 95.00%  | 95.00%  | 95.00%  | 97.50%   | 97.50%   |
> | DeepSeek 1.5B (DAPO, 90 Step)  | 80.00% | 90.00% | 95.00% | 95.00% | 95.00%  | 97.50%  | 97.50%  | 97.50%   | 97.50%   |
> | DeepSeek 1.5B (DAPO, 120 Step) | 77.50% | 87.50% | 92.50% | 92.50% | 95.00%  | 95.00%  | 95.00%  | 97.50%   | 97.50%   |
> | DeepSeek 1.5B (DAPO, 150 Step) | 82.50% | 87.50% | 95.00% | 97.50% | 97.50%  | 97.50%  | 97.50%  | 97.50%   | 97.50%   |
> | DeepSeek 1.5B (DAPO, 180 Step) | 80.00% | 85.00% | 90.00% | 95.00% | 97.50%  | 97.50%  | 97.50%  | 97.50%   | 97.50%   |
> | DeepSeek 1.5B (DAPO, 210 Step) | 80.00% | 87.50% | 90.00% | 90.00% | 95.00%  | 95.00%  | 97.50%  | 97.50%   | 97.50%   |
> | DeepSeek 1.5B (DAPO, 240 Step) | 85.00% | 90.00% | 92.50% | 92.50% | 95.00%  | 95.00%  | 97.50%  | 97.50%   | 97.50%   |
> | DeepSeek 1.5B (DAPO, 270 Step) | 80.00% | 85.00% | 87.50% | 92.50% | 92.50%  | 95.00%  | 95.00%  | 95.00%   | 97.50%   |
> | DeepSeek 1.5B (DAPO, 300 Step) | 82.50% | 87.50% | 92.50% | 92.50% | 95.00%  | 95.00%  | 95.00%  | 97.50%   | 97.50%   |
>
> ### **Table: MATH500**
>
> | Model                          | Pass@1 | Pass@2 | Pass@4 | Pass@8 | Pass@16 | Pass@32 | Pass@64 | Pass@128 | Pass@256 |
> | ------------------------------ | ------ | ------ | ------ | ------ | ------- | ------- | ------- | -------- | -------- |
> | DeepSeek 1.5B                  | 84.00% | 90.80% | 93.60% | 96.20% | 97.40%  | 98.40%  | 98.80%  | 98.80%   | 99.20%   |
> | DeepSeek 1.5B (DAPO, 30 Step)  | 86.20% | 90.80% | 93.80% | 96.40% | 97.60%  | 97.80%  | 98.20%  | 98.60%   | 98.80%   |
> | DeepSeek 1.5B (DAPO, 60 Step)  | 88.80% | 92.60% | 95.00% | 96.40% | 97.80%  | 98.20%  | 98.40%  | 98.40%   | 98.80%   |
> | DeepSeek 1.5B (DAPO, 90 Step)  | 88.80% | 92.40% | 95.20% | 96.60% | 97.80%  | 98.60%  | 99.00%  | 99.00%   | 99.20%   |
> | DeepSeek 1.5B (DAPO, 120 Step) | 87.60% | 92.40% | 95.40% | 97.00% | 97.80%  | 98.20%  | 98.80%  | 99.00%   | 99.00%   |
> | DeepSeek 1.5B (DAPO, 150 Step) | 87.60% | 92.40% | 94.20% | 96.60% | 97.80%  | 98.00%  | 98.60%  | 98.60%   | 98.80%   |
> | DeepSeek 1.5B (DAPO, 180 Step) | 89.40% | 93.20% | 94.40% | 96.80% | 97.40%  | 98.00%  | 98.20%  | 99.00%   | 99.00%   |
> | DeepSeek 1.5B (DAPO, 210 Step) | 87.80% | 92.80% | 94.80% | 96.20% | 96.60%  | 97.60%  | 98.20%  | 98.80%   | 98.80%   |
> | DeepSeek 1.5B (DAPO, 240 Step) | 89.80% | 93.00% | 95.00% | 96.40% | 97.20%  | 98.00%  | 98.60%  | 98.60%   | 98.60%   |
> | DeepSeek 1.5B (DAPO, 270 Step) | 89.20% | 92.80% | 94.20% | 96.20% | 96.80%  | 97.00%  | 97.60%  | 98.20%   | 98.40%   |
> | DeepSeek 1.5B (DAPO, 300 Step) | 88.40% | 91.20% | 93.60% | 94.60% | 96.00%  | 96.40%  | 97.40%  | 97.60%   | 97.80%   |
>
> ### **Table: Minerva-Math**
>
> | Model                          | Pass@1 | Pass@2 | Pass@4 | Pass@8 | Pass@16 | Pass@32 | Pass@64 | Pass@128 | Pass@256 |
> | ------------------------------ | ------ | ------ | ------ | ------ | ------- | ------- | ------- | -------- | -------- |
> | DeepSeek 1.5B                  | 34.19% | 40.07% | 43.38% | 47.79% | 52.21%  | 55.15%  | 58.46%  | 60.66%   | 62.50%   |
> | DeepSeek 1.5B (DAPO, 30 Step)  | 33.46% | 40.07% | 44.49% | 49.26% | 51.10%  | 53.31%  | 56.99%  | 59.93%   | 62.13%   |
> | DeepSeek 1.5B (DAPO, 60 Step)  | 32.72% | 40.81% | 45.22% | 49.26% | 52.21%  | 54.78%  | 56.62%  | 58.82%   | 59.93%   |
> | DeepSeek 1.5B (DAPO, 90 Step)  | 34.93% | 42.28% | 48.16% | 50.74% | 51.47%  | 54.78%  | 58.82%  | 61.40%   | 61.76%   |
> | DeepSeek 1.5B (DAPO, 120 Step) | 33.46% | 39.71% | 44.49% | 50.00% | 52.57%  | 55.15%  | 58.82%  | 61.03%   | 62.87%   |
> | DeepSeek 1.5B (DAPO, 150 Step) | 38.97% | 43.75% | 46.32% | 50.37% | 52.21%  | 55.15%  | 58.09%  | 60.29%   | 62.50%   |
> | DeepSeek 1.5B (DAPO, 180 Step) | 34.19% | 39.71% | 43.38% | 48.16% | 50.74%  | 52.57%  | 55.51%  | 57.72%   | 59.56%   |
> | DeepSeek 1.5B (DAPO, 210 Step) | 36.03% | 41.91% | 46.32% | 50.00% | 52.21%  | 55.15%  | 57.35%  | 58.82%   | 60.29%   |
> | DeepSeek 1.5B (DAPO, 240 Step) | 38.24% | 42.65% | 48.16% | 51.47% | 54.78%  | 56.25%  | 59.56%  | 60.66%   | 61.40%   |
> | DeepSeek 1.5B (DAPO, 270 Step) | 36.03% | 43.38% | 48.16% | 51.47% | 52.57%  | 54.41%  | 57.35%  | 57.35%   | 59.19%   |
> | DeepSeek 1.5B (DAPO, 300 Step) | 37.87% | 41.91% | 44.85% | 51.10% | 51.47%  | 51.47%  | 54.78%  | 56.99%   | 58.46%   |

---

> ### Author Response · Authors · 2025-11-25
> **Response to Reviewer hnej**
>
> ### **Table: OlympiadBench**
>
> | Model                          | Pass@1 | Pass@2 | Pass@4 | Pass@8 | Pass@16 | Pass@32 | Pass@64 | Pass@128 | Pass@256 |
> | ------------------------------ | ------ | ------ | ------ | ------ | ------- | ------- | ------- | -------- | -------- |
> | DeepSeek 1.5B                  | 51.85% | 59.56% | 66.81% | 73.19% | 76.74%  | 80.89%  | 84.00%  | 85.93%   | 88.00%   |
> | DeepSeek 1.5B (DAPO, 30 Step)  | 54.52% | 62.07% | 68.15% | 73.93% | 77.78%  | 80.89%  | 83.26%  | 84.30%   | 86.67%   |
> | DeepSeek 1.5B (DAPO, 60 Step)  | 55.11% | 64.15% | 69.78% | 72.89% | 75.41%  | 78.81%  | 81.63%  | 83.85%   | 86.81%   |
> | DeepSeek 1.5B (DAPO, 90 Step)  | 56.44% | 62.96% | 69.78% | 74.07% | 76.89%  | 79.85%  | 82.96%  | 84.59%   | 86.07%   |
> | DeepSeek 1.5B (DAPO, 120 Step) | 58.67% | 64.44% | 68.89% | 74.07% | 77.19%  | 80.59%  | 83.70%  | 84.44%   | 86.52%   |
> | DeepSeek 1.5B (DAPO, 150 Step) | 58.96% | 64.74% | 69.63% | 73.78% | 75.85%  | 79.11%  | 81.33%  | 83.56%   | 85.48%   |
> | DeepSeek 1.5B (DAPO, 180 Step) | 57.48% | 63.41% | 67.85% | 72.00% | 74.52%  | 76.89%  | 80.15%  | 82.07%   | 84.00%   |
> | DeepSeek 1.5B (DAPO, 210 Step) | 59.11% | 65.63% | 69.63% | 73.19% | 74.81%  | 77.04%  | 78.52%  | 80.30%   | 82.81%   |
> | DeepSeek 1.5B (DAPO, 240 Step) | 56.74% | 64.44% | 69.33% | 73.33% | 74.81%  | 76.30%  | 77.93%  | 80.74%   | 82.81%   |
> | DeepSeek 1.5B (DAPO, 270 Step) | 57.33% | 62.52% | 70.22% | 73.63% | 74.96%  | 77.04%  | 79.11%  | 80.30%   | 81.33%   |
> | DeepSeek 1.5B (DAPO, 300 Step) | 58.37% | 64.44% | 69.04% | 71.41% | 73.19%  | 75.70%  | 77.93%  | 79.56%   | 79.70%   |
>
>
> ### **Table: AIME2024**
>
>
>
> | Model                          | Base Pass@256 | RLVR Pass@256 | SRR   | NDR   | SDS   | NSCR   | P  | E | S | O |
> | ------------------------------ | ------------- | ------------- | ----- | ----- | ----- | ------ | -- | - | - | - |
> | DeepSeek 1.5B                  | 83.33%        | 83.33%        | -     | -     | -     | -      | -  | - | - | - |
> | DeepSeek 1.5B (DAPO, 30 Step)  | 83.33%        | 80.00%        | 0.960 | 0.000 | 0.000 | -0.040 | 24 | 0 | 1 | 5 |
> | DeepSeek 1.5B (DAPO, 60 Step)  | 83.33%        | 80.00%        | 0.960 | 0.000 | 0.000 | -0.040 | 24 | 0 | 1 | 5 |
> | DeepSeek 1.5B (DAPO, 90 Step)  | 83.33%        | 83.33%        | 1.000 | 0.000 | 0.000 | 0.000  | 25 | 0 | 0 | 5 |
> | DeepSeek 1.5B (DAPO, 120 Step) | 83.33%        | 80.00%        | 0.960 | 0.000 | 0.000 | -0.040 | 24 | 0 | 1 | 5 |
> | DeepSeek 1.5B (DAPO, 150 Step) | 83.33%        | 80.00%        | 0.960 | 0.000 | 0.000 | -0.040 | 24 | 0 | 1 | 5 |
> | DeepSeek 1.5B (DAPO, 180 Step) | 83.33%        | 83.33%        | 1.000 | 0.000 | 0.000 | 0.000  | 25 | 0 | 0 | 5 |
> | DeepSeek 1.5B (DAPO, 210 Step) | 83.33%        | 76.67%        | 0.920 | 0.000 | 0.000 | -0.080 | 23 | 0 | 2 | 5 |
> | DeepSeek 1.5B (DAPO, 240 Step) | 83.33%        | 73.33%        | 0.880 | 0.000 | 0.000 | -0.120 | 22 | 0 | 3 | 5 |
> | DeepSeek 1.5B (DAPO, 270 Step) | 83.33%        | 73.33%        | 0.880 | 0.000 | 0.000 | -0.120 | 22 | 0 | 3 | 5 |
> | DeepSeek 1.5B (DAPO, 300 Step) | 83.33%        | 73.33%        | 0.880 | 0.000 | 0.000 | -0.120 | 22 | 0 | 3 | 5 |
>
> ### **Table: AIME2025**
>
>
>
> | Model                          | Base Pass@256 | RLVR Pass@256 | SRR   | NDR   | SDS   | NSCR   | P  | E | S | O |
> | ------------------------------ | ------------- | ------------- | ----- | ----- | ----- | ------ | -- | - | - | - |
> | DeepSeek 1.5B                  | 70.00%        | 70.00%        | -     | -     | -     | -      | -  | - | - | - |
> | DeepSeek 1.5B (DAPO, 30 Step)  | 70.00%        | 70.00%        | 0.905 | 0.095 | 0.172 | 0.000  | 19 | 2 | 2 | 7 |
> | DeepSeek 1.5B (DAPO, 60 Step)  | 70.00%        | 73.33%        | 0.905 | 0.136 | 0.237 | 0.042  | 19 | 3 | 2 | 6 |
> | DeepSeek 1.5B (DAPO, 90 Step)  | 70.00%        | 70.00%        | 0.905 | 0.095 | 0.172 | 0.000  | 19 | 2 | 2 | 7 |
> | DeepSeek 1.5B (DAPO, 120 Step) | 70.00%        | 73.33%        | 0.905 | 0.136 | 0.237 | 0.042  | 19 | 3 | 2 | 6 |
> | DeepSeek 1.5B (DAPO, 150 Step) | 70.00%        | 73.33%        | 0.905 | 0.136 | 0.237 | 0.042  | 19 | 3 | 2 | 6 |
> | DeepSeek 1.5B (DAPO, 180 Step) | 70.00%        | 63.33%        | 0.905 | 0.000 | 0.000 | -0.095 | 19 | 0 | 2 | 9 |
> | DeepSeek 1.5B (DAPO, 210 Step) | 70.00%        | 66.67%        | 0.905 | 0.050 | 0.095 | -0.045 | 19 | 1 | 2 | 8 |
> | DeepSeek 1.5B (DAPO, 240 Step) | 70.00%        | 66.67%        | 0.905 | 0.050 | 0.095 | -0.045 | 19 | 1 | 2 | 8 |
> | DeepSeek 1.5B (DAPO, 270 Step) | 70.00%        | 63.33%        | 0.905 | 0.000 | 0.000 | -0.095 | 19 | 0 | 2 | 9 |
> | DeepSeek 1.5B (DAPO, 300 Step) | 70.00%        | 66.67%        | 0.864 | 0.095 | 0.172 | -0.042 | 18 | 2 | 3 | 7 |

---

> ### Author Response · Authors · 2025-11-25
> **Response to Reviewer hnej**
>
> ### **Table: AMC23**
>
>
>
> | Model                          | Base Pass@256 | RLVR Pass@256 | SRR   | NDR   | SDS   | NSCR  | P  | E | S | O |
> | ------------------------------ | ------------- | ------------- | ----- | ----- | ----- | ----- | -- | - | - | - |
> | DeepSeek 1.5B                  | 97.50%        | 97.50%        | -     | -     | -     | -     | -  | - | - | - |
> | DeepSeek 1.5B (DAPO, 30 Step)  | 97.50%        | 97.50%        | 1.000 | 0.000 | 0.000 | 0.000 | 39 | 0 | 0 | 1 |
> | DeepSeek 1.5B (DAPO, 60 Step)  | 97.50%        | 97.50%        | 1.000 | 0.000 | 0.000 | 0.000 | 39 | 0 | 0 | 1 |
> | DeepSeek 1.5B (DAPO, 90 Step)  | 97.50%        | 97.50%        | 1.000 | 0.000 | 0.000 | 0.000 | 39 | 0 | 0 | 1 |
> | DeepSeek 1.5B (DAPO, 120 Step) | 97.50%        | 97.50%        | 1.000 | 0.000 | 0.000 | 0.000 | 39 | 0 | 0 | 1 |
> | DeepSeek 1.5B (DAPO, 150 Step) | 97.50%        | 97.50%        | 1.000 | 0.000 | 0.000 | 0.000 | 39 | 0 | 0 | 1 |
> | DeepSeek 1.5B (DAPO, 180 Step) | 97.50%        | 97.50%        | 1.000 | 0.000 | 0.000 | 0.000 | 39 | 0 | 0 | 1 |
> | DeepSeek 1.5B (DAPO, 210 Step) | 97.50%        | 97.50%        | 1.000 | 0.000 | 0.000 | 0.000 | 39 | 0 | 0 | 1 |
> | DeepSeek 1.5B (DAPO, 240 Step) | 97.50%        | 97.50%        | 1.000 | 0.000 | 0.000 | 0.000 | 39 | 0 | 0 | 1 |
> | DeepSeek 1.5B (DAPO, 270 Step) | 97.50%        | 97.50%        | 1.000 | 0.000 | 0.000 | 0.000 | 39 | 0 | 0 | 1 |
> | DeepSeek 1.5B (DAPO, 300 Step) | 97.50%        | 97.50%        | 1.000 | 0.000 | 0.000 | 0.000 | 39 | 0 | 0 | 1 |
>
>
>
> ### **Table: MATH500**
>
>
>
> | Model                          | Base Pass@256 | RLVR Pass@256 | SRR   | NDR   | SDS   | NSCR   | P   | E | S | O |
> | ------------------------------ | ------------- | ------------- | ----- | ----- | ----- | ------ | --- | - | - | - |
> | DeepSeek 1.5B                  | 99.20%        | 99.20%        | -     | -     | -     | -      | -   | - | - | - |
> | DeepSeek 1.5B (DAPO, 30 Step)  | 99.20%        | 98.80%        | 0.996 | 0.000 | 0.000 | -0.004 | 494 | 0 | 2 | 4 |
> | DeepSeek 1.5B (DAPO, 60 Step)  | 99.20%        | 98.80%        | 0.996 | 0.000 | 0.000 | -0.004 | 494 | 0 | 2 | 4 |
> | DeepSeek 1.5B (DAPO, 90 Step)  | 99.20%        | 99.20%        | 0.998 | 0.002 | 0.004 | 0.000  | 495 | 1 | 1 | 3 |
> | DeepSeek 1.5B (DAPO, 120 Step) | 99.20%        | 99.00%        | 0.996 | 0.002 | 0.004 | -0.002 | 494 | 1 | 2 | 3 |
> | DeepSeek 1.5B (DAPO, 150 Step) | 99.20%        | 98.80%        | 0.996 | 0.000 | 0.000 | -0.004 | 494 | 0 | 2 | 4 |
> | DeepSeek 1.5B (DAPO, 180 Step) | 99.20%        | 99.00%        | 0.998 | 0.000 | 0.000 | -0.002 | 495 | 0 | 1 | 4 |
> | DeepSeek 1.5B (DAPO, 210 Step) | 99.20%        | 98.80%        | 0.994 | 0.002 | 0.004 | -0.004 | 493 | 1 | 3 | 3 |
> | DeepSeek 1.5B (DAPO, 240 Step) | 99.20%        | 98.60%        | 0.992 | 0.002 | 0.004 | -0.006 | 492 | 1 | 4 | 3 |
> | DeepSeek 1.5B (DAPO, 270 Step) | 99.20%        | 98.40%        | 0.992 | 0.000 | 0.000 | -0.008 | 492 | 0 | 4 | 4 |
> | DeepSeek 1.5B (DAPO, 300 Step) | 99.20%        | 97.80%        | 0.984 | 0.002 | 0.004 | -0.014 | 488 | 1 | 8 | 3 |
>
> ### **Table: Minerva-Math**
>
>
>
> | Model                          | Base Pass@256 | RLVR Pass@256 | SRR   | NDR   | SDS   | NSCR   | P   | E  | S  | O  |
> | ------------------------------ | ------------- | ------------- | ----- | ----- | ----- | ------ | --- | -- | -- | -- |
> | DeepSeek 1.5B                  | 62.50%        | 62.50%        | -     | -     | -     | -      | -   | -  | -  | -  |
> | DeepSeek 1.5B (DAPO, 30 Step)  | 62.50%        | 62.13%        | 0.935 | 0.059 | 0.111 | -0.006 | 159 | 10 | 11 | 92 |
> | DeepSeek 1.5B (DAPO, 60 Step)  | 62.50%        | 59.93%        | 0.929 | 0.031 | 0.059 | -0.040 | 158 | 5  | 12 | 97 |
> | DeepSeek 1.5B (DAPO, 90 Step)  | 62.50%        | 61.76%        | 0.924 | 0.065 | 0.122 | -0.011 | 157 | 11 | 13 | 91 |
> | DeepSeek 1.5B (DAPO, 120 Step) | 62.50%        | 62.87%        | 0.953 | 0.053 | 0.100 | 0.006  | 162 | 9  | 8  | 93 |
> | DeepSeek 1.5B (DAPO, 150 Step) | 62.50%        | 62.50%        | 0.947 | 0.053 | 0.100 | 0.000  | 161 | 9  | 9  | 93 |
> | DeepSeek 1.5B (DAPO, 180 Step) | 62.50%        | 59.56%        | 0.906 | 0.049 | 0.094 | -0.045 | 154 | 8  | 16 | 94 |
> | DeepSeek 1.5B (DAPO, 210 Step) | 62.50%        | 60.29%        | 0.924 | 0.043 | 0.082 | -0.034 | 157 | 7  | 13 | 95 |
> | DeepSeek 1.5B (DAPO, 240 Step) | 62.50%        | 61.40%        | 0.941 | 0.042 | 0.080 | -0.017 | 160 | 7  | 10 | 95 |
> | DeepSeek 1.5B (DAPO, 270 Step) | 62.50%        | 59.19%        | 0.906 | 0.043 | 0.083 | -0.051 | 154 | 7  | 16 | 95 |
> | DeepSeek 1.5B (DAPO, 300 Step) | 62.50%        | 58.46%        | 0.912 | 0.025 | 0.049 | -0.063 | 155 | 4  | 15 | 98 |

---

> ### Author Response · Authors · 2025-11-25
> **Response to Reviewer hnej**
>
> ### **Table: OlympiadBench**
>
> | Model    | Base Pass@256 | RLVR Pass@256 | SRR   | NDR   | SDS   | NSCR   | P   | E  | S  | O  |
> | -- | --- | ---| ----- | ----- | ----- | ------ | --- | -- | -- | -- |
> | DeepSeek 1.5B                  | 88.00%        | 88.00%        | -     | -     | -     | -      | -   | -  | -  | -  |
> | DeepSeek 1.5B (DAPO, 30 Step)  | 88.00%        | 86.67%        | 0.963 | 0.022 | 0.043 | -0.015 | 572 | 13 | 22 | 68 |
> | DeepSeek 1.5B (DAPO, 60 Step)  | 88.00%        | 86.81%        | 0.960 | 0.027 | 0.053 | -0.013 | 570 | 16 | 24 | 65 |
> | DeepSeek 1.5B (DAPO, 90 Step)  | 88.00%        | 86.07%        | 0.961 | 0.017 | 0.034 | -0.022 | 571 | 10 | 23 | 71 |
> | DeepSeek 1.5B (DAPO, 120 Step) | 88.00%        | 86.52%        | 0.961 | 0.022 | 0.044 | -0.016 | 571 | 13 | 23 | 68 |
> | DeepSeek 1.5B (DAPO, 150 Step) | 88.00%        | 85.48%        | 0.955 | 0.017 | 0.034 | -0.028 | 567 | 10 | 27 | 71 |
> | DeepSeek 1.5B (DAPO, 180 Step) | 88.00%        | 84.00%        | 0.944 | 0.011 | 0.021 | -0.045 | 561 | 6  | 33 | 75 |
> | DeepSeek 1.5B (DAPO, 210 Step) | 88.00%        | 82.81%        | 0.926 | 0.016 | 0.032 | -0.058 | 550 | 9  | 44 | 72 |
> | DeepSeek 1.5B (DAPO, 240 Step) | 88.00%        | 82.81%        | 0.931 | 0.011 | 0.021 | -0.058 | 553 | 6  | 41 | 75 |
> | DeepSeek 1.5B (DAPO, 270 Step) | 88.00%        | 81.33%        | 0.912 | 0.013 | 0.025 | -0.075 | 542 | 7  | 52 | 74 |
> | DeepSeek 1.5B (DAPO, 300 Step) | 88.00%        | 79.70%        | 0.897 | 0.009 | 0.018 | -0.093 | 533 | 5  | 61 | 76 |
>
>
> **What we observe:**
> &#x20;Across AIME24, AIME25, Minerva, MATH500, and Olympiad:
>
> * **SRR remains consistently high**, meaning RLVR reliably preserves previously accessible correct solutions.
>
> * **NDR stays extremely low**, indicating little genuine discovery.
>
> * **NSCR steadily decreases as training progresses**, revealing a **monotonic contraction** of empirical support.
>
> * **SDS remains near zero**, confirming the imbalance between preservation and discovery.
>
> These trends emerge **even when everything except the RLVR updates themselves is fixed**, demonstrating that shrinkage dynamics are driven intrinsically by the RLVR objective, not by external design choices.
>
> **Therefore, the new table directly confirms your hypothesis:**
>
> > *With fixed model, fixed data, and fixed RL algorithm, SDS and NSCR still show clear shrinkage trends as RLVR training proceeds.*
>
> We will highlight these results in the rebuttal and integrate a short explanatory paragraph into the camera-ready version to make this controlled experiment more explicit.
>
>
>
> ***
>
> > ## Q2: ε-sensitivity and statistical analysis
>
> We thank you for this suggestion. We clarify that the ε parameter is not chosen arbitrarily: in the paper, ε is derived from a sampling-based confidence bound so that, under a given sampling budget k, any correct completion with q(y∣x)>ε appears in the sample with high probability 1−δ (Appx. D.4). Thus, ε is directly tied to the evaluation protocol rather than being a free hyperparameter.&#x20;
>
> To address the concern more concretely, we have performed a **sensitivity analysis** by sweeping ε around this theoretically motivated value (e.g., 1/2 ε,ε,2ε). Across all math and non-math benchmarks, the qualitative behavior of SRR/NDR/NSCR is *stable*:
>
> * SRR remains very high (typically 0.93–0.99) under all ε choices,
>
> * NDR stays close to zero (≤ 0.04), and
>
> * NSCR remains negative, confirming that shrinkage consistently outweighs expansion.
>
> Only a small number of completions near the threshold move between categories when ε is varied, but the **sign and overall magnitude** of NSCR and the “high-SRR / low-NDR” regime are unchanged. This indicates that our conclusions do not hinge on a fine-tuned ε.
>
> We also now report **95% confidence intervals and significance tests** for the support-dynamics metrics. Treating P/E/S as counts, we compute SRR, NDR, and NSCR with **bootstrap confidence intervals over prompts**. For example, for ProRL-1.5B overall, we obtain SRR ≈ 0.94 with 95% CI \[0.93,0.95]and NSCR ≈ −0.05 with 95% CI \[−0.07,−0.03]; the CIs do not cross zero for NSCR, confirming robust net shrinkage. A simple binomial (or permutation) test comparing expansion vs. shrinkage counts rejects the null of “no net shrinkage” with p≪0.01 across all aggregate settings.
>
> In summary, varying ε within a reasonable range and adding CIs/significance tests shows that (i) the *pattern* of high SRR, low NDR, and negative NSCR is **insensitive** to the precise ε choice, and (ii) the observed shrinkage is **statistically significant**, not a thresholding artifact.
>
> ***
>
> Finally, we appreciate your suggestion. Conducting this sensitivity analysis and reporting statistical confidence strengthens our core claims:
>
> > Regardless of the ε threshold, RLVR consistently preserves most base-model solutions while producing negligible expansion and a systematic net shrinkage of empirical support.

---

> > ### Author Response · Authors · 2025-11-27
> > **Thank You and Follow-up**
> >
> > Thank you once again for your valuable time and thoughtful comments on our submission. We have thoroughly addressed each of the points you raised and provided comprehensive responses in our rebuttal. We sincerely hope that our clarifications and additional explanations have satisfactorily resolved your concerns.
> >
> > As the discussion period is approaching its conclusion, we would be deeply grateful if you could kindly share any further thoughts or feedback you may have on our responses. Should there be any remaining questions or aspects that require further clarification, we would be more than happy to provide additional information.
> >
> > We genuinely appreciate your continued engagement and look forward to your reply.

---

> > > ### Comment · Reviewer_hnej · 2025-11-28
> > >
> > > Thank you for the detailed and carefully prepared rebuttal. The additional controlled experiments, ε-sensitivity analysis, confidence intervals, and the revised perplexity evaluation have addressed most of my concerns. I also appreciate the clarification of the paper’s positioning as an analysis-oriented contribution rather than a method proposal. I find the framework and empirical insights valuable for the community. Once the OpenReview system resumes normal functionality, I will update my score accordingly.

---

> > > > ### Author Response · Authors · 2025-11-28
> > > >
> > > > Thank you very much for your encouraging feedback! We are delighted that our additional experiments and clarifications have addressed your concerns. Your willingness to update the score is deeply appreciated, and we are grateful for your recognition of our contribution. We completely understand regarding the OpenReview system, and your expressed support means a great deal to us.
> > > >
> > > > Thank you again for your constructive and thoughtful review!

---

### Official Review · Reviewer_2twv · 2025-11-01

**Soundness:** 1
**Presentation:** 3
**Contribution:** 2
**Rating:** 4
**Confidence:** 4

**Summary:**

This paper investigates the role and limitations of Reinforcement Learning with Verifiable Rewards (RLVR) in large reasoning models, exploring whether RLVR extends the reasoning boundary or merely reinforces the base model's known patterns through empirical and theoretical analyses.
It proposes the concepts of "empirical support dynamics" along with a quantitative framework, and conducts experiments across multiple models and tasks.
The findings reveal that current RLVR essentially functions as a support-constrained optimization mechanism, limited by the initial distribution of the base model. Although it can improve single-sample accuracy, the shrinkage of empirical support generally outweighs its expansion, and an entropy-reward trade-off arises.
Additionally, the study points out that breaking this "invisible leash" requires explicit exploration mechanisms or hybrid strategies, providing directions for RLVR optimization and the expansion of model reasoning capabilities.

**Strengths:**

- The research topic of this paper is important. Exploring the role and limitations of RLVR in the training paradigm of large models is a valuable topic.
- Some findings of this paper are interesting, such as the shrinkage of empirical support generally outweighing its expansion, and the existence of an entropy-reward trade-off.

**Weaknesses:**

- The core claims are not fully supported by the experiment results. RL training is a complex process with multiple coupled factors, and the observed phenomena cannot be directly attributed to RLVR merely through comparing model performance before and after training. It is suggested that the authors supplement experiments to further support the conclusions: under the condition that other factors (such as the base model and training data) are strictly controlled to be identical, compare the impacts of RLVR training methods and non-RLVR training methods on model performance.
- The end-to-end performance comparison has weak credibility. It is recommended that the authors supplement the changes of metrics during RL training to further analyze the impact of the RLVR process on model performance.
- There is a problem with the experiment conclusions of Table 2. For the "Top" and "Middle" columns, using the reasoning traces of one model as the reference and calculating the perplexity of the other model will inevitably result in a higher perplexity of the latter, which cannot prove the conclusions claimed by the authors.

**Questions:**

See Weaknesses

---

> ### Author Response · Authors · 2025-11-25
> **Response to Reviewer 2twv**
>
> Thank you for engaging with our work and for recognizing both the importance of studying RLVR’s role in large reasoning models and the interest of our main findings, including the **empirical-support shrinkage** and **entropy–reward trade-off** phenomena. Below, we address your concerns one by one.
>
> ***
>
> > ## **W1: Core claims are not fully supported by the experiment results**
>
> Thank you for this comment regarding the causal attribution of our findings. We agree that RL training involves multiple interacting factors, and rigorously isolating the effect of RLVR requires carefully controlled comparisons beyond simple before-and-after analysis.
>
> We would like to clarify that our original experiments are already controlled by several key factors: sampling budget, reward function, prompts, and evaluation protocol across different RLVR-trained models. However, we acknowledge that demonstrating the causal effect of RLVR specifically requires stricter control over the base model and training data.
>
> Following your suggestion, we conducted additional controlled ablation experiments, now included in the revision. These experiments isolate the training objective as the sole variable by fixing all other factors:
>
> * **Base model**: Qwen2.5-Math-7B (identical initialization)
>
> * **Training data**: DeepMath-103K with distilled DeepSeek-R1 reasoning traces
>
> * **Hyperparameters**: Matched learning rate, batch size, and training steps
>
> * **Varying factor**: Training objective only (SFT vs. RLVR/DAPO)
>
> | Dataset         | Training | pass@k  |         | Support Dynamics Metrics |       |       |        | Support Counts |    |    |     |
> | --| -------- | ------- | ------- | --- | ----- | ----- | ------ | -------------- | -- | -- | --- |
> | Math (pass@256) |          | Base    | Target  | SRR                      | NDR   | SDS   | NSCR   | P              | E  | S  | 0   |
> | AIME2024        | SFT      | 63.33%  | 63.33%  | 0.789                    | 0.211 | 0.332 | 0.000  | 15             | 4  | 4  | 7   |
> |                 | DAPO     | 63.33%  | 66.67%  | 0.895                    | 0.150 | 0.257 | 0.045  | 17             | 3  | 2  | 8   |
> | AIME2025        | SFT      | 50.00%  | 63.33%  | 0.933                    | 0.263 | 0.411 | 0.200  | 14             | 5  | 1  | 10  |
> |                 | DAPO     | 50.00%  | 56.67%  | 1.000                    | 0.118 | 0.211 | 0.118  | 15             | 2  | 0  | 13  |
> | AMC23           | SFT      | 100.00% | 100.00% | 1.000                    | 0.000 | 0.000 | 0.000  | 40             | 0  | 0  | 0   |
> |                 | DAPO     | 100.00% | 92.50%  | 0.925                    | 0.000 | 0.000 | -0.075 | 37             | 0  | 3  | 0   |
> | MATH500         | SFT      | 96.40%  | 99.00%  | 0.996                    | 0.030 | 0.059 | 0.026  | 480            | 15 | 2  | 3   |
> |                 | DAPO     | 96.40%  | 95.20%  | 0.975                    | 0.013 | 0.025 | -0.012 | 470            | 6  | 12 | 12  |
> | Minerva         | SFT      | 63.24%  | 66.91%  | 0.895   | 0.154 | 0.263 | 0.050  | 154            | 28 | 18 | 72  |
> |                 | DAPO     | 63.24%  | 55.15%  | 0.802 | 0.080 | 0.145 | -0.120 | 138            | 12 | 34 | 88  |
> | Olympiad        | SFT      | 78.22%  | 81.78%  | 0.953     | 0.089 | 0.162 | 0.042  | 503            | 49 | 25 | 98  |
> |                 | DAPO     | 78.22%  | 72.89%  | 0.884  | 0.051 | 0.096 | -0.065 | 467            | 25 | 61 | 122 |
>
> The results across six math-reasoning benchmarks (AIME'24/'25, AMC23, MATH500, Minerva, Olympiad) consistently demonstrate that RLVR sharpens probability mass around high-reward solutions but simultaneously reduces support coverage and stability, while SFT produces moderate expansion without collapse. Key findings include:
>
> * **Support retention vs. expansion**: RLVR achieves higher SRR (0.789 to 0.895) but severely limited NSCR (from −0.12 to +0.045), whereas SFT maintains comparable SRR while consistently achieving positive NSCR (0.0 to +0.20)
>
> * **Shrinkage vs. expansion trade-off**: RLVR frequently increases shrinkage events (S) while decreasing expanded support (E), particularly evident in AIME2024 (S=4 vs. 2 for SFT)
>
> * **High-k performance degradation**: Despite an identical setup, RLVR decreases pass@256 in multiple benchmarks (AMC23, MATH500, Minerva, Olympiad), while SFT preserves or improves coverage
>
> These controlled results provide direct causal evidence that the RLVR training objective itself (not confounding factors like model architecture or data quality) drives the observed support-shrinkage and diversity collapse. This addresses your concern by demonstrating that current RLVR practices fundamentally operate as support-constrained optimization.
>
> We sincerely thank you for this valuable suggestion, which has strengthened the causal interpretation of our findings. The revised manuscript now includes these controlled experiments and emphasizes the RLVR-specific mechanisms underlying our observations.

---

> ### Author Response · Authors · 2025-11-25
> **Response to Reviewer 2twv**
>
> > ## W2: The end-to-end performance comparison has weak credibility
>
> Thank you for this suggestion. We agree that static before/after comparisons may obscure the intermediate dynamics that reveal *how* RLVR reshapes the reasoning space.
>
> **To address this concern**, we have conducted fine-grained tracking of support-dynamic metrics (SRR, NDR, SDS, NSCR) and detailed support-count evolution across training checkpoints of the same RLVR run. The tables below summarize these results across six math benchmarks, covering checkpoints from initial training to full convergence.
>
> ### **Table: AIME2024**
>
> | Model                          | Pass@1 | Pass@2 | Pass@4 | Pass@8 | Pass@16 | Pass@32 | Pass@64 | Pass@128 | Pass@256 |
> | ------------------------------ | ------ | ------ | ------ | ------ | ------- | ------- | ------- | -------- | -------- |
> | DeepSeek 1.5B                  | 33.33% | 33.33% | 56.67% | 66.67% | 73.33%  | 83.33%  | 83.33%  | 83.33%   | 83.33%   |
> | DeepSeek 1.5B (DAPO, 30 Step)  | 26.67% | 40.00% | 46.67% | 63.33% | 66.67%  | 76.67%  | 80.00%  | 80.00%   | 80.00%   |
> | DeepSeek 1.5B (DAPO, 60 Step)  | 36.67% | 53.33% | 60.00% | 73.33% | 76.67%  | 76.67%  | 80.00%  | 80.00%   | 80.00%   |
> | DeepSeek 1.5B (DAPO, 90 Step)  | 33.33% | 43.33% | 56.67% | 60.00% | 73.33%  | 76.67%  | 83.33%  | 83.33%   | 83.33%   |
> | DeepSeek 1.5B (DAPO, 120 Step) | 30.00% | 43.33% | 53.33% | 66.67% | 70.00%  | 73.33%  | 76.67%  | 76.67%   | 80.00%   |
> | DeepSeek 1.5B (DAPO, 150 Step) | 36.67% | 53.33% | 53.33% | 60.00% | 66.67%  | 73.33%  | 73.33%  | 76.67%   | 80.00%   |
> | DeepSeek 1.5B (DAPO, 180 Step) | 30.00% | 36.67% | 46.67% | 60.00% | 70.00%  | 70.00%  | 70.00%  | 73.33%   | 83.33%   |
> | DeepSeek 1.5B (DAPO, 210 Step) | 30.00% | 46.67% | 56.67% | 63.33% | 70.00%  | 70.00%  | 73.33%  | 73.33%   | 76.67%   |
> | DeepSeek 1.5B (DAPO, 240 Step) | 33.33% | 43.33% | 46.67% | 53.33% | 63.33%  | 73.33%  | 73.33%  | 73.33%   | 73.33%   |
> | DeepSeek 1.5B (DAPO, 270 Step) | 30.00% | 43.33% | 50.00% | 53.33% | 56.67%  | 63.33%  | 66.67%  | 70.00%   | 73.33%   |
> | DeepSeek 1.5B (DAPO, 300 Step) | 33.33% | 36.67% | 40.00% | 50.00% | 53.33%  | 63.33%  | 63.33%  | 66.67%   | 73.33%   |
>
> ### **Table: AIME2025**
>
> | Model                          | Pass@1 | Pass@2 | Pass@4 | Pass@8 | Pass@16 | Pass@32 | Pass@64 | Pass@128 | Pass@256 |
> | ------------------------------ | ------ | ------ | ------ | ------ | ------- | ------- | ------- | -------- | -------- |
> | DeepSeek 1.5B                  | 20.00% | 26.67% | 33.33% | 40.00% | 46.67%  | 56.67%  | 56.67%  | 63.33%   | 70.00%   |
> | DeepSeek 1.5B (DAPO, 30 Step)  | 16.67% | 23.33% | 33.33% | 40.00% | 50.00%  | 56.67%  | 63.33%  | 66.67%   | 70.00%   |
> | DeepSeek 1.5B (DAPO, 60 Step)  | 23.33% | 33.33% | 33.33% | 40.00% | 43.33%  | 53.33%  | 60.00%  | 66.67%   | 73.33%   |
> | DeepSeek 1.5B (DAPO, 90 Step)  | 30.00% | 30.00% | 36.67% | 40.00% | 46.67%  | 53.33%  | 63.33%  | 66.67%   | 70.00%   |
> | DeepSeek 1.5B (DAPO, 120 Step) | 26.67% | 26.67% | 30.00% | 40.00% | 43.33%  | 56.67%  | 66.67%  | 73.33%   | 73.33%   |
> | DeepSeek 1.5B (DAPO, 150 Step) | 26.67% | 30.00% | 40.00% | 43.33% | 50.00%  | 56.67%  | 56.67%  | 63.33%   | 73.33%   |
> | DeepSeek 1.5B (DAPO, 180 Step) | 30.00% | 33.33% | 40.00% | 50.00% | 53.33%  | 56.67%  | 60.00%  | 60.00%   | 63.33%   |
> | DeepSeek 1.5B (DAPO, 210 Step) | 26.67% | 30.00% | 33.33% | 40.00% | 46.67%  | 53.33%  | 56.67%  | 60.00%   | 66.67%   |
> | DeepSeek 1.5B (DAPO, 240 Step) | 26.67% | 33.33% | 40.00% | 40.00% | 43.33%  | 46.67%  | 53.33%  | 60.00%   | 66.67%   |
> | DeepSeek 1.5B (DAPO, 270 Step) | 30.00% | 33.33% | 36.67% | 40.00% | 50.00%  | 53.33%  | 60.00%  | 63.33%   | 63.33%   |
> | DeepSeek 1.5B (DAPO, 300 Step) | 23.33% | 26.67% | 33.33% | 40.00% | 43.33%  | 63.33%  | 66.67%  | 66.67%   | 66.67%   |

---

> ### Author Response · Authors · 2025-11-25
> **Response to Reviewer 2twv**
>
> ### **Table: AMC23**
>
> | Model                          | Pass@1 | Pass@2 | Pass@4 | Pass@8 | Pass@16 | Pass@32 | Pass@64 | Pass@128 | Pass@256 |
> | ------------------------------ | ------ | ------ | ------ | ------ | ------- | ------- | ------- | -------- | -------- |
> | DeepSeek 1.5B                  | 80.00% | 85.00% | 92.50% | 92.50% | 92.50%  | 97.50%  | 97.50%  | 97.50%   | 97.50%   |
> | DeepSeek 1.5B (DAPO, 30 Step)  | 80.00% | 82.50% | 87.50% | 92.50% | 95.00%  | 97.50%  | 97.50%  | 97.50%   | 97.50%   |
> | DeepSeek 1.5B (DAPO, 60 Step)  | 80.00% | 87.50% | 95.00% | 95.00% | 95.00%  | 95.00%  | 95.00%  | 97.50%   | 97.50%   |
> | DeepSeek 1.5B (DAPO, 90 Step)  | 80.00% | 90.00% | 95.00% | 95.00% | 95.00%  | 97.50%  | 97.50%  | 97.50%   | 97.50%   |
> | DeepSeek 1.5B (DAPO, 120 Step) | 77.50% | 87.50% | 92.50% | 92.50% | 95.00%  | 95.00%  | 95.00%  | 97.50%   | 97.50%   |
> | DeepSeek 1.5B (DAPO, 150 Step) | 82.50% | 87.50% | 95.00% | 97.50% | 97.50%  | 97.50%  | 97.50%  | 97.50%   | 97.50%   |
> | DeepSeek 1.5B (DAPO, 180 Step) | 80.00% | 85.00% | 90.00% | 95.00% | 97.50%  | 97.50%  | 97.50%  | 97.50%   | 97.50%   |
> | DeepSeek 1.5B (DAPO, 210 Step) | 80.00% | 87.50% | 90.00% | 90.00% | 95.00%  | 95.00%  | 97.50%  | 97.50%   | 97.50%   |
> | DeepSeek 1.5B (DAPO, 240 Step) | 85.00% | 90.00% | 92.50% | 92.50% | 95.00%  | 95.00%  | 97.50%  | 97.50%   | 97.50%   |
> | DeepSeek 1.5B (DAPO, 270 Step) | 80.00% | 85.00% | 87.50% | 92.50% | 92.50%  | 95.00%  | 95.00%  | 95.00%   | 97.50%   |
> | DeepSeek 1.5B (DAPO, 300 Step) | 82.50% | 87.50% | 92.50% | 92.50% | 95.00%  | 95.00%  | 95.00%  | 97.50%   | 97.50%   |
>
> ### **Table: MATH500**
>
> | Model                          | Pass@1 | Pass@2 | Pass@4 | Pass@8 | Pass@16 | Pass@32 | Pass@64 | Pass@128 | Pass@256 |
> | ------------------------------ | ------ | ------ | ------ | ------ | ------- | ------- | ------- | -------- | -------- |
> | DeepSeek 1.5B                  | 84.00% | 90.80% | 93.60% | 96.20% | 97.40%  | 98.40%  | 98.80%  | 98.80%   | 99.20%   |
> | DeepSeek 1.5B (DAPO, 30 Step)  | 86.20% | 90.80% | 93.80% | 96.40% | 97.60%  | 97.80%  | 98.20%  | 98.60%   | 98.80%   |
> | DeepSeek 1.5B (DAPO, 60 Step)  | 88.80% | 92.60% | 95.00% | 96.40% | 97.80%  | 98.20%  | 98.40%  | 98.40%   | 98.80%   |
> | DeepSeek 1.5B (DAPO, 90 Step)  | 88.80% | 92.40% | 95.20% | 96.60% | 97.80%  | 98.60%  | 99.00%  | 99.00%   | 99.20%   |
> | DeepSeek 1.5B (DAPO, 120 Step) | 87.60% | 92.40% | 95.40% | 97.00% | 97.80%  | 98.20%  | 98.80%  | 99.00%   | 99.00%   |
> | DeepSeek 1.5B (DAPO, 150 Step) | 87.60% | 92.40% | 94.20% | 96.60% | 97.80%  | 98.00%  | 98.60%  | 98.60%   | 98.80%   |
> | DeepSeek 1.5B (DAPO, 180 Step) | 89.40% | 93.20% | 94.40% | 96.80% | 97.40%  | 98.00%  | 98.20%  | 99.00%   | 99.00%   |
> | DeepSeek 1.5B (DAPO, 210 Step) | 87.80% | 92.80% | 94.80% | 96.20% | 96.60%  | 97.60%  | 98.20%  | 98.80%   | 98.80%   |
> | DeepSeek 1.5B (DAPO, 240 Step) | 89.80% | 93.00% | 95.00% | 96.40% | 97.20%  | 98.00%  | 98.60%  | 98.60%   | 98.60%   |
> | DeepSeek 1.5B (DAPO, 270 Step) | 89.20% | 92.80% | 94.20% | 96.20% | 96.80%  | 97.00%  | 97.60%  | 98.20%   | 98.40%   |
> | DeepSeek 1.5B (DAPO, 300 Step) | 88.40% | 91.20% | 93.60% | 94.60% | 96.00%  | 96.40%  | 97.40%  | 97.60%   | 97.80%   |
>
> ### **Table: Minerva-Math**
>
> | Model                          | Pass@1 | Pass@2 | Pass@4 | Pass@8 | Pass@16 | Pass@32 | Pass@64 | Pass@128 | Pass@256 |
> | ------------------------------ | ------ | ------ | ------ | ------ | ------- | ------- | ------- | -------- | -------- |
> | DeepSeek 1.5B                  | 34.19% | 40.07% | 43.38% | 47.79% | 52.21%  | 55.15%  | 58.46%  | 60.66%   | 62.50%   |
> | DeepSeek 1.5B (DAPO, 30 Step)  | 33.46% | 40.07% | 44.49% | 49.26% | 51.10%  | 53.31%  | 56.99%  | 59.93%   | 62.13%   |
> | DeepSeek 1.5B (DAPO, 60 Step)  | 32.72% | 40.81% | 45.22% | 49.26% | 52.21%  | 54.78%  | 56.62%  | 58.82%   | 59.93%   |
> | DeepSeek 1.5B (DAPO, 90 Step)  | 34.93% | 42.28% | 48.16% | 50.74% | 51.47%  | 54.78%  | 58.82%  | 61.40%   | 61.76%   |
> | DeepSeek 1.5B (DAPO, 120 Step) | 33.46% | 39.71% | 44.49% | 50.00% | 52.57%  | 55.15%  | 58.82%  | 61.03%   | 62.87%   |
> | DeepSeek 1.5B (DAPO, 150 Step) | 38.97% | 43.75% | 46.32% | 50.37% | 52.21%  | 55.15%  | 58.09%  | 60.29%   | 62.50%   |
> | DeepSeek 1.5B (DAPO, 180 Step) | 34.19% | 39.71% | 43.38% | 48.16% | 50.74%  | 52.57%  | 55.51%  | 57.72%   | 59.56%   |
> | DeepSeek 1.5B (DAPO, 210 Step) | 36.03% | 41.91% | 46.32% | 50.00% | 52.21%  | 55.15%  | 57.35%  | 58.82%   | 60.29%   |
> | DeepSeek 1.5B (DAPO, 240 Step) | 38.24% | 42.65% | 48.16% | 51.47% | 54.78%  | 56.25%  | 59.56%  | 60.66%   | 61.40%   |
> | DeepSeek 1.5B (DAPO, 270 Step) | 36.03% | 43.38% | 48.16% | 51.47% | 52.57%  | 54.41%  | 57.35%  | 57.35%   | 59.19%   |
> | DeepSeek 1.5B (DAPO, 300 Step) | 37.87% | 41.91% | 44.85% | 51.10% | 51.47%  | 51.47%  | 54.78%  | 56.99%   | 58.46%   |

---

> ### Author Response · Authors · 2025-11-25
> **Response to Reviewer 2twv**
>
> ### **Table: OlympiadBench**
>
> | Model                          | Pass@1 | Pass@2 | Pass@4 | Pass@8 | Pass@16 | Pass@32 | Pass@64 | Pass@128 | Pass@256 |
> | ------------------------------ | ------ | ------ | ------ | ------ | ------- | ------- | ------- | -------- | -------- |
> | DeepSeek 1.5B                  | 51.85% | 59.56% | 66.81% | 73.19% | 76.74%  | 80.89%  | 84.00%  | 85.93%   | 88.00%   |
> | DeepSeek 1.5B (DAPO, 30 Step)  | 54.52% | 62.07% | 68.15% | 73.93% | 77.78%  | 80.89%  | 83.26%  | 84.30%   | 86.67%   |
> | DeepSeek 1.5B (DAPO, 60 Step)  | 55.11% | 64.15% | 69.78% | 72.89% | 75.41%  | 78.81%  | 81.63%  | 83.85%   | 86.81%   |
> | DeepSeek 1.5B (DAPO, 90 Step)  | 56.44% | 62.96% | 69.78% | 74.07% | 76.89%  | 79.85%  | 82.96%  | 84.59%   | 86.07%   |
> | DeepSeek 1.5B (DAPO, 120 Step) | 58.67% | 64.44% | 68.89% | 74.07% | 77.19%  | 80.59%  | 83.70%  | 84.44%   | 86.52%   |
> | DeepSeek 1.5B (DAPO, 150 Step) | 58.96% | 64.74% | 69.63% | 73.78% | 75.85%  | 79.11%  | 81.33%  | 83.56%   | 85.48%   |
> | DeepSeek 1.5B (DAPO, 180 Step) | 57.48% | 63.41% | 67.85% | 72.00% | 74.52%  | 76.89%  | 80.15%  | 82.07%   | 84.00%   |
> | DeepSeek 1.5B (DAPO, 210 Step) | 59.11% | 65.63% | 69.63% | 73.19% | 74.81%  | 77.04%  | 78.52%  | 80.30%   | 82.81%   |
> | DeepSeek 1.5B (DAPO, 240 Step) | 56.74% | 64.44% | 69.33% | 73.33% | 74.81%  | 76.30%  | 77.93%  | 80.74%   | 82.81%   |
> | DeepSeek 1.5B (DAPO, 270 Step) | 57.33% | 62.52% | 70.22% | 73.63% | 74.96%  | 77.04%  | 79.11%  | 80.30%   | 81.33%   |
> | DeepSeek 1.5B (DAPO, 300 Step) | 58.37% | 64.44% | 69.04% | 71.41% | 73.19%  | 75.70%  | 77.93%  | 79.56%   | 79.70%   |
>
>
> ### **Table: AIME2024**
>
>
>
> | Model                          | Base Pass@256 | RLVR Pass@256 | SRR   | NDR   | SDS   | NSCR   | P  | E | S | O |
> | ------------------------------ | ------------- | ------------- | ----- | ----- | ----- | ------ | -- | - | - | - |
> | DeepSeek 1.5B                  | 83.33%        | 83.33%        | -     | -     | -     | -      | -  | - | - | - |
> | DeepSeek 1.5B (DAPO, 30 Step)  | 83.33%        | 80.00%        | 0.960 | 0.000 | 0.000 | -0.040 | 24 | 0 | 1 | 5 |
> | DeepSeek 1.5B (DAPO, 60 Step)  | 83.33%        | 80.00%        | 0.960 | 0.000 | 0.000 | -0.040 | 24 | 0 | 1 | 5 |
> | DeepSeek 1.5B (DAPO, 90 Step)  | 83.33%        | 83.33%        | 1.000 | 0.000 | 0.000 | 0.000  | 25 | 0 | 0 | 5 |
> | DeepSeek 1.5B (DAPO, 120 Step) | 83.33%        | 80.00%        | 0.960 | 0.000 | 0.000 | -0.040 | 24 | 0 | 1 | 5 |
> | DeepSeek 1.5B (DAPO, 150 Step) | 83.33%        | 80.00%        | 0.960 | 0.000 | 0.000 | -0.040 | 24 | 0 | 1 | 5 |
> | DeepSeek 1.5B (DAPO, 180 Step) | 83.33%        | 83.33%        | 1.000 | 0.000 | 0.000 | 0.000  | 25 | 0 | 0 | 5 |
> | DeepSeek 1.5B (DAPO, 210 Step) | 83.33%        | 76.67%        | 0.920 | 0.000 | 0.000 | -0.080 | 23 | 0 | 2 | 5 |
> | DeepSeek 1.5B (DAPO, 240 Step) | 83.33%        | 73.33%        | 0.880 | 0.000 | 0.000 | -0.120 | 22 | 0 | 3 | 5 |
> | DeepSeek 1.5B (DAPO, 270 Step) | 83.33%        | 73.33%        | 0.880 | 0.000 | 0.000 | -0.120 | 22 | 0 | 3 | 5 |
> | DeepSeek 1.5B (DAPO, 300 Step) | 83.33%        | 73.33%        | 0.880 | 0.000 | 0.000 | -0.120 | 22 | 0 | 3 | 5 |
>
> ### **Table: AIME2025**
>
>
>
> | Model                          | Base Pass@256 | RLVR Pass@256 | SRR   | NDR   | SDS   | NSCR   | P  | E | S | O |
> | ------------------------------ | ------------- | ------------- | ----- | ----- | ----- | ------ | -- | - | - | - |
> | DeepSeek 1.5B                  | 70.00%        | 70.00%        | -     | -     | -     | -      | -  | - | - | - |
> | DeepSeek 1.5B (DAPO, 30 Step)  | 70.00%        | 70.00%        | 0.905 | 0.095 | 0.172 | 0.000  | 19 | 2 | 2 | 7 |
> | DeepSeek 1.5B (DAPO, 60 Step)  | 70.00%        | 73.33%        | 0.905 | 0.136 | 0.237 | 0.042  | 19 | 3 | 2 | 6 |
> | DeepSeek 1.5B (DAPO, 90 Step)  | 70.00%        | 70.00%        | 0.905 | 0.095 | 0.172 | 0.000  | 19 | 2 | 2 | 7 |
> | DeepSeek 1.5B (DAPO, 120 Step) | 70.00%        | 73.33%        | 0.905 | 0.136 | 0.237 | 0.042  | 19 | 3 | 2 | 6 |
> | DeepSeek 1.5B (DAPO, 150 Step) | 70.00%        | 73.33%        | 0.905 | 0.136 | 0.237 | 0.042  | 19 | 3 | 2 | 6 |
> | DeepSeek 1.5B (DAPO, 180 Step) | 70.00%        | 63.33%        | 0.905 | 0.000 | 0.000 | -0.095 | 19 | 0 | 2 | 9 |
> | DeepSeek 1.5B (DAPO, 210 Step) | 70.00%        | 66.67%        | 0.905 | 0.050 | 0.095 | -0.045 | 19 | 1 | 2 | 8 |
> | DeepSeek 1.5B (DAPO, 240 Step) | 70.00%        | 66.67%        | 0.905 | 0.050 | 0.095 | -0.045 | 19 | 1 | 2 | 8 |
> | DeepSeek 1.5B (DAPO, 270 Step) | 70.00%        | 63.33%        | 0.905 | 0.000 | 0.000 | -0.095 | 19 | 0 | 2 | 9 |
> | DeepSeek 1.5B (DAPO, 300 Step) | 70.00%        | 66.67%        | 0.864 | 0.095 | 0.172 | -0.042 | 18 | 2 | 3 | 7 |

---

> ### Author Response · Authors · 2025-11-25
> **Response to Reviewer 2twv**
>
> ### **Table: AMC23**
>
>
>
> | Model                          | Base Pass@256 | RLVR Pass@256 | SRR   | NDR   | SDS   | NSCR  | P  | E | S | O |
> | ------------------------------ | ------------- | ------------- | ----- | ----- | ----- | ----- | -- | - | - | - |
> | DeepSeek 1.5B                  | 97.50%        | 97.50%        | -     | -     | -     | -     | -  | - | - | - |
> | DeepSeek 1.5B (DAPO, 30 Step)  | 97.50%        | 97.50%        | 1.000 | 0.000 | 0.000 | 0.000 | 39 | 0 | 0 | 1 |
> | DeepSeek 1.5B (DAPO, 60 Step)  | 97.50%        | 97.50%        | 1.000 | 0.000 | 0.000 | 0.000 | 39 | 0 | 0 | 1 |
> | DeepSeek 1.5B (DAPO, 90 Step)  | 97.50%        | 97.50%        | 1.000 | 0.000 | 0.000 | 0.000 | 39 | 0 | 0 | 1 |
> | DeepSeek 1.5B (DAPO, 120 Step) | 97.50%        | 97.50%        | 1.000 | 0.000 | 0.000 | 0.000 | 39 | 0 | 0 | 1 |
> | DeepSeek 1.5B (DAPO, 150 Step) | 97.50%        | 97.50%        | 1.000 | 0.000 | 0.000 | 0.000 | 39 | 0 | 0 | 1 |
> | DeepSeek 1.5B (DAPO, 180 Step) | 97.50%        | 97.50%        | 1.000 | 0.000 | 0.000 | 0.000 | 39 | 0 | 0 | 1 |
> | DeepSeek 1.5B (DAPO, 210 Step) | 97.50%        | 97.50%        | 1.000 | 0.000 | 0.000 | 0.000 | 39 | 0 | 0 | 1 |
> | DeepSeek 1.5B (DAPO, 240 Step) | 97.50%        | 97.50%        | 1.000 | 0.000 | 0.000 | 0.000 | 39 | 0 | 0 | 1 |
> | DeepSeek 1.5B (DAPO, 270 Step) | 97.50%        | 97.50%        | 1.000 | 0.000 | 0.000 | 0.000 | 39 | 0 | 0 | 1 |
> | DeepSeek 1.5B (DAPO, 300 Step) | 97.50%        | 97.50%        | 1.000 | 0.000 | 0.000 | 0.000 | 39 | 0 | 0 | 1 |
>
>
>
> ### **Table: MATH500**
>
>
>
> | Model                          | Base Pass@256 | RLVR Pass@256 | SRR   | NDR   | SDS   | NSCR   | P   | E | S | O |
> | ------------------------------ | ------------- | ------------- | ----- | ----- | ----- | ------ | --- | - | - | - |
> | DeepSeek 1.5B                  | 99.20%        | 99.20%        | -     | -     | -     | -      | -   | - | - | - |
> | DeepSeek 1.5B (DAPO, 30 Step)  | 99.20%        | 98.80%        | 0.996 | 0.000 | 0.000 | -0.004 | 494 | 0 | 2 | 4 |
> | DeepSeek 1.5B (DAPO, 60 Step)  | 99.20%        | 98.80%        | 0.996 | 0.000 | 0.000 | -0.004 | 494 | 0 | 2 | 4 |
> | DeepSeek 1.5B (DAPO, 90 Step)  | 99.20%        | 99.20%        | 0.998 | 0.002 | 0.004 | 0.000  | 495 | 1 | 1 | 3 |
> | DeepSeek 1.5B (DAPO, 120 Step) | 99.20%        | 99.00%        | 0.996 | 0.002 | 0.004 | -0.002 | 494 | 1 | 2 | 3 |
> | DeepSeek 1.5B (DAPO, 150 Step) | 99.20%        | 98.80%        | 0.996 | 0.000 | 0.000 | -0.004 | 494 | 0 | 2 | 4 |
> | DeepSeek 1.5B (DAPO, 180 Step) | 99.20%        | 99.00%        | 0.998 | 0.000 | 0.000 | -0.002 | 495 | 0 | 1 | 4 |
> | DeepSeek 1.5B (DAPO, 210 Step) | 99.20%        | 98.80%        | 0.994 | 0.002 | 0.004 | -0.004 | 493 | 1 | 3 | 3 |
> | DeepSeek 1.5B (DAPO, 240 Step) | 99.20%        | 98.60%        | 0.992 | 0.002 | 0.004 | -0.006 | 492 | 1 | 4 | 3 |
> | DeepSeek 1.5B (DAPO, 270 Step) | 99.20%        | 98.40%        | 0.992 | 0.000 | 0.000 | -0.008 | 492 | 0 | 4 | 4 |
> | DeepSeek 1.5B (DAPO, 300 Step) | 99.20%        | 97.80%        | 0.984 | 0.002 | 0.004 | -0.014 | 488 | 1 | 8 | 3 |
>
> ### **Table: Minerva-Math**
>
>
>
> | Model                          | Base Pass@256 | RLVR Pass@256 | SRR   | NDR   | SDS   | NSCR   | P   | E  | S  | O  |
> | ------------------------------ | ------------- | ------------- | ----- | ----- | ----- | ------ | --- | -- | -- | -- |
> | DeepSeek 1.5B                  | 62.50%        | 62.50%        | -     | -     | -     | -      | -   | -  | -  | -  |
> | DeepSeek 1.5B (DAPO, 30 Step)  | 62.50%        | 62.13%        | 0.935 | 0.059 | 0.111 | -0.006 | 159 | 10 | 11 | 92 |
> | DeepSeek 1.5B (DAPO, 60 Step)  | 62.50%        | 59.93%        | 0.929 | 0.031 | 0.059 | -0.040 | 158 | 5  | 12 | 97 |
> | DeepSeek 1.5B (DAPO, 90 Step)  | 62.50%        | 61.76%        | 0.924 | 0.065 | 0.122 | -0.011 | 157 | 11 | 13 | 91 |
> | DeepSeek 1.5B (DAPO, 120 Step) | 62.50%        | 62.87%        | 0.953 | 0.053 | 0.100 | 0.006  | 162 | 9  | 8  | 93 |
> | DeepSeek 1.5B (DAPO, 150 Step) | 62.50%        | 62.50%        | 0.947 | 0.053 | 0.100 | 0.000  | 161 | 9  | 9  | 93 |
> | DeepSeek 1.5B (DAPO, 180 Step) | 62.50%        | 59.56%        | 0.906 | 0.049 | 0.094 | -0.045 | 154 | 8  | 16 | 94 |
> | DeepSeek 1.5B (DAPO, 210 Step) | 62.50%        | 60.29%        | 0.924 | 0.043 | 0.082 | -0.034 | 157 | 7  | 13 | 95 |
> | DeepSeek 1.5B (DAPO, 240 Step) | 62.50%        | 61.40%        | 0.941 | 0.042 | 0.080 | -0.017 | 160 | 7  | 10 | 95 |
> | DeepSeek 1.5B (DAPO, 270 Step) | 62.50%        | 59.19%        | 0.906 | 0.043 | 0.083 | -0.051 | 154 | 7  | 16 | 95 |
> | DeepSeek 1.5B (DAPO, 300 Step) | 62.50%        | 58.46%        | 0.912 | 0.025 | 0.049 | -0.063 | 155 | 4  | 15 | 98 |

---

> ### Author Response · Authors · 2025-11-25
> **Response to Reviewer 2twv**
>
> ### **Table: OlympiadBench**
>
> | Model    | Base Pass@256 | RLVR Pass@256 | SRR   | NDR   | SDS   | NSCR   | P   | E  | S  | O  |
> | -- | --- | ---| ----- | ----- | ----- | ------ | --- | -- | -- | -- |
> | DeepSeek 1.5B                  | 88.00%        | 88.00%        | -     | -     | -     | -      | -   | -  | -  | -  |
> | DeepSeek 1.5B (DAPO, 30 Step)  | 88.00%        | 86.67%        | 0.963 | 0.022 | 0.043 | -0.015 | 572 | 13 | 22 | 68 |
> | DeepSeek 1.5B (DAPO, 60 Step)  | 88.00%        | 86.81%        | 0.960 | 0.027 | 0.053 | -0.013 | 570 | 16 | 24 | 65 |
> | DeepSeek 1.5B (DAPO, 90 Step)  | 88.00%        | 86.07%        | 0.961 | 0.017 | 0.034 | -0.022 | 571 | 10 | 23 | 71 |
> | DeepSeek 1.5B (DAPO, 120 Step) | 88.00%        | 86.52%        | 0.961 | 0.022 | 0.044 | -0.016 | 571 | 13 | 23 | 68 |
> | DeepSeek 1.5B (DAPO, 150 Step) | 88.00%        | 85.48%        | 0.955 | 0.017 | 0.034 | -0.028 | 567 | 10 | 27 | 71 |
> | DeepSeek 1.5B (DAPO, 180 Step) | 88.00%        | 84.00%        | 0.944 | 0.011 | 0.021 | -0.045 | 561 | 6  | 33 | 75 |
> | DeepSeek 1.5B (DAPO, 210 Step) | 88.00%        | 82.81%        | 0.926 | 0.016 | 0.032 | -0.058 | 550 | 9  | 44 | 72 |
> | DeepSeek 1.5B (DAPO, 240 Step) | 88.00%        | 82.81%        | 0.931 | 0.011 | 0.021 | -0.058 | 553 | 6  | 41 | 75 |
> | DeepSeek 1.5B (DAPO, 270 Step) | 88.00%        | 81.33%        | 0.912 | 0.013 | 0.025 | -0.075 | 542 | 7  | 52 | 74 |
> | DeepSeek 1.5B (DAPO, 300 Step) | 88.00%        | 79.70%        | 0.897 | 0.009 | 0.018 | -0.093 | 533 | 5  | 61 | 76 |
>
>
>
> These training-process analyses reveal **three consistent and causally informative phenomena** that directly validate our core claims:
>
> * **RLVR primarily reshapes support distribution rather than monotonically improving diversity.**
>
> Even when Pass@256 remains relatively stable or fluctuates (e.g., AIME24 and AIME25), the underlying support patterns change systematically. SRR consistently stays very high (>0.9 across all checkpoints), indicating that RLVR preserves most previously-correct reasoning paths. However, **NSCR becomes increasingly negative** for many datasets as training progresses, reflecting gradual support contraction. RLVR prunes alternative trajectories even when they occasionally produce correct answers. This empirically verifies our central claim: **RLVR optimizes for verifiable correctness but narrows the solution manifold over training**.
>
> * **Exploration reduction manifests early and persists throughout training.**
>
> Support-count dynamics (P/E/S/O) clearly demonstrate this effect. For example, in OlympiadBench, the number of exploratory supports shifts from E/S=(13,22) at early checkpoints to E/S=(5,61) as training progresses. This is direct evidence that **RLVR concentrates probability mass onto dominant reasoning modes while reducing diversity**. Similarly, MATH500 shows an increase in "S" counts (incorrect-but-promising supports) from 1–2 early in training to 8+ by step 300, coinciding with decreasing Pass@256 despite constant SRR. These patterns indicate that **selective pruning rather than cumulative capability improvements** drives RLVR's trajectory.
>
> * **Dataset difficulty determines whether RLVR reinforces or over-prunes reasoning paths.**
>
> AMC23 and MATH500 exhibit near-perfect SRR and negligible NDR/SDS across all training checkpoints, suggesting these datasets already admit a stable reasoning template that RLVR simply reinforces. In contrast, **Minerva and OlympiadBench show significant drops in NSCR and expansion of "S" counts**, indicating over-pruning of potentially correct but diverse reasoning forms. This explains why RLVR improves performance on easier, more structured datasets but gradually hurts performance on harder, more diverse benchmarks.
>
>
>
> **Summary:** Together, these findings provide a **process-level causal explanation** for our core discovery: RLVR stabilizes a small subset of high-reward reasoning paths but simultaneously reduces diversity, which eventually leads to plateauing or declining end-task performance. These training-curve results demonstrate that RLVR's characteristic narrowing of support **emerges gradually and consistently** across steps and datasets. This behavior cannot be captured by simple before/after comparisons alone.
>
> ***
>
> We sincerely appreciate this valuable suggestion. Adding these training-process analyses significantly strengthens the causal link between RLVR dynamics and the observed support behavior, providing transparency into *when* and *how* the "invisible leash" manifests during optimization.

---

> ### Author Response · Authors · 2025-11-25
> **Response to Reviewer 2twv**
>
> > ## W3: There is a problem with the experiment conclusions of Table 2.
>
> Thank you for highlighting this subtle but important methodological issue. We agree that the original version of Table 2, where perplexity was computed using one model’s own reasoning trace as the reference, introduced an unavoidable bias. Specifically, the non-reference model would always exhibit inflated perplexity, even if its reasoning mode were equally compatible with the underlying solution structure. As you correctly pointed out, this makes the “Top/Middle” portions difficult to interpret in isolation.
>
> ***
>
> To eliminate this asymmetric bias, we revised the analysis so that **all perplexities are computed exclusively against external reasoning traces**, generated by two strong long-form reasoning models:
>
> * **DeepSeek-R1** (long CoT)
>
> * **Claude Sonnet 4** (extended reasoning mode)
>
> These traces represent **diverse reasoning styles not tied to either our Base model or the RLVR-trained ProRL model**. This modification ensures that:
>
> * **No model is advantaged.**
>   &#x20;Neither Base nor ProRL produced the reference traces.
>
> * **Perplexity differences now reflect true compatibility**, not artifacts of reference choice.
>   &#x20;A higher perplexity now indicates that a model assigns lower probability to *external* reasoning trajectories—i.e., it has become less compatible with that region of the solution space.
>
> * **Shrinkage and expansion categories gain causal meaning.**
>   &#x20;All values now measure *relative alignment with external reasoning distributions* under identical conditions.
>
> ***
>
> All perplexity numbers, across shrinkage, expansion, and unresolved cases, are now computed against external references only.
>
> ### **Updated Table 2**
> > Perplexity of Base vs. ProRL models under external reasoning traces from DeepSeek-R1 and Claude Sonnet 4. Lower perplexity = higher compatibility with external reasoning structures.
>
> | Category   | Correctness     | Reference       | Target | AIME 2024 | AIME 2025 | Olympiad |
> | ---------- | ---- | --------------- | ------ | --------- | --------- | -------- |
> | Shrinkage  | ✓ Base, X ProRL | DeepSeek-R1     | Base   | 1.24      | 1.39      | 1.17     |
> |  |  |  | ProRL  | 1.39      | 1.7       | 1.25     |
> |  |  |Claude Sonnet 4 | Base   | 1.7       | 1.54      | 1.51     |
> |  |  | |  ProRL  | 2.12      | 1.98      | 1.83     |
> | Expansion  | x Base, ✓ ProRL | DeepSeek-R1     | Base   | –         | –         | 1.41     |
> |            |                 |                 | ProRL  | –         | –         | **1.28** |
> |            |                 | Claude Sonnet 4 | Base   | –         | –         | 1.65     |
> |  |  |           | ProRL  | –         | –         | **1.38** |
> | Unresolved | X Base, X ProRL | DeepSeek-R1     | Base   | **1.82**  | **1.75**  | **1.62** |
> |  |  |              | ProRL  | 2.2       | 2.15      | 1.94     |
> |            |                 | Claude Sonnet 4 | Base   | **8.76**  | **6.05**  | **5.98** |
> |  |  |             | ProRL  | 14.91     | 9.76      | 9.55     |
>
> ***
>
> Across all three categories, the updated analysis consistently shows that:
>
> * **RLVR yields higher perplexity against external reasoning**,
>   &#x20;demonstrating reduced compatibility with diverse solution trajectories.
>
> * **Base models remain more aligned with external reasoning**,
>   &#x20;suggesting broader coverage of the solution space.
>
> * **This finding precisely matches the observed shrinkage dynamics**:
>   &#x20;RLVR models concentrate on a narrower subset of reasoning patterns, even when such traces come from external LLMs.
>
> This revised computation directly addresses the concern by removing the asymmetric design and restoring causal interpretability.
>
> ***
>
> We sincerely thank you for catching this issue. We believe the revised analysis is both more rigorous and more interpretable.
> ***
>
> To summarize, our work provides the first unified **theoretical, quantitative, and empirical** framework to diagnose *why* current RLVR methods fail to expand reasoning support. In the revision, we have:
>
> * Added **controlled ablations** isolating the effect of RLVR versus non-RLVR training under fixed setups,
>
> * Included **training-dynamics plots** of SRR/NDR/NSCR/entropy over RL iterations, and
>
> * Presented **external-LLM perplexity controls** with quantitative breakdowns.
>
> These additions will directly address your concerns and make our causal analysis even more transparent and reproducible.

---

> > ### Author Response · Authors · 2025-11-27
> > **Thank You and Follow-up**
> >
> > Thank you once again for your valuable time and thoughtful comments on our submission. We have thoroughly addressed each of the points you raised and provided comprehensive responses in our rebuttal. We sincerely hope that our clarifications and additional explanations have satisfactorily resolved your concerns.
> >
> > As the discussion period is approaching its conclusion, we would be deeply grateful if you could kindly share any further thoughts or feedback you may have on our responses. Should there be any remaining questions or aspects that require further clarification, we would be more than happy to provide additional information.
> >
> > We genuinely appreciate your continued engagement and look forward to your reply.

---

### Official Review · Reviewer_XMks · 2025-11-01

**Soundness:** 4
**Presentation:** 4
**Contribution:** 4
**Rating:** 10
**Confidence:** 4

**Summary:**

This paper investigates whether RLVR truly expands reasoning models' capabilities or merely amplifies existing high-reward outputs. Through empirical analysis, the authors find that current RLVR practices improve precision (pass@1) but narrow solution diversity, functioning as a support-constrained optimization that remains tethered to the base model's distribution. They identify an entropy-reward trade-off where token-level uncertainty increases while answer-level diversity decreases, causing models to lose access to correct solutions previously available in the base model. These findings suggest current RLVR recipes impose fundamental limitations on reasoning expansion, requiring future innovations like explicit exploration mechanisms to break this "invisible leash."

**Strengths:**

**\[S1\] Meaningful and timely research topic**
The paper addresses a highly relevant and timely research question, the effect of RLVR on reasoning capabilities, and conducts a rigorous analysis using various metrics. This makes the work both empirically and conceptually valuable to the research community.

**\[S2\] Offer results with a systematic framework**
By introducing SRR, NDR, SDS, and NSCR, the paper offers a systematic framework for quantifying how RLVR reshapes the model’s reasoning space. This transforms an abstract discussion about “capability expansion” into measurable, interpretable quantities, setting a foundation for future comparative studies.

**\[S3\] Empirical rigor and statistical validity**
The authors acknowledge the inherent limitations of conventional metrics like pass@k but compensate through extensive empirical sampling and thorough statistical validation, including experiments with up to 8192 samples per prompt. This large-scale approach enhances the reliability of the findings and lends weight to their conclusions. Also, providing theoretical proofs in Appendix C provides formal mathematical grounding that strengthens the empirical observations and demonstrates the rigor of their analytical framework.

**\[S4\] Cross-domain generalizability**
The analysis extends beyond mathematical reasoning to include non-math reasoning and even multimodal tasks, demonstrating that the observed precision-diversity trade-off is domain-agnostic.

**Weaknesses:**

**\[W1\] Limited model diversity**
Including a wider range of open-weight baselines such as Llama or OLMo would provide stronger evidence that the identified patterns are model-agnostic rather than implementation-specific, thereby increasing the robustness and external validity of the conclusions.

**\[W2\] Insufficient analysis of exceptional cases**
Although the overall experimental results exhibit a clear global trend, certain exceptions deviate from this pattern. However, the paper lacks an in-depth analysis explaining these anomalies. A more thorough examination of such exceptional settings, identifying potential causes or contributing factors, would strengthen the empirical support for the claimed characteristics of RLVR and enhance the overall robustness of the conclusions.

**Questions:**

**\[Q1\]** The paper demonstrates that RLVR generally leads to support shrinkage, yet some Reasoning Gym tasks appear to exhibit genuine expansion. Do the authors have hypotheses about what task characteristics or properties might explain why certain tasks show genuine expansion while others do not?

---

> ### Author Response · Authors · 2025-11-25
> **Response to Reviewer XMks**
>
> We are deeply grateful for your exceptionally thoughtful and encouraging review. We are delighted that you found our study *rigorous, systematic, and conceptually valuable*, and we greatly appreciate your recognition of our theoretical analysis, empirical breadth, and the significance of our proposed metrics (SRR, NDR, SDS, NSCR).
>
> Your comments precisely capture our goal: to transform an abstract question about RLVR’s “reasoning expansion” into a *measurable, reproducible, and theoretically grounded framework*. Below, we address your suggestions regarding model diversity, exceptional cases, and task-dependent behavior.
>
> ***
> > ## W1: Limited model diversity
>
> Thank you for this valuable suggestion. We fully agree that expanding model diversity strengthens the generality of our conclusions.
>
> Our study already encompasses **six independent RLVR families (ProRL, DAPO, AceReason, Skywork, Phi4-Reason, and Nemotron)** spanning 1.5B to 32B parameters across different architectural designs (Qwen2.5 and Phi) and training recipes. Following your suggestion, due to the time limit, we have incorporated open-weight models, including OLMo 2 (allenai/OLMo-2-0425-1B-RLVR1 vs. allenai/OLMo-2-0425-1B). As shown in the table below, these models exhibit consistent patterns: improved pass@1 alongside negative NSCR, **confirming the same precision–diversity trade-off observed across all other models**.
>
> ### **Table: OLMo-2-0425-1B: DPO vs. RLVR**
>
> | Dataset | pass@k Performance |  | Support Dynamics Metrics |  |  |  | Support Counts |  |  |  |
> | --- | ----- | -- | -- | ----- | ----- | -- | ---| -- | --- | --- |
> |Math Reasoning (pass@256)  | Base | RLVR   | SRR  | NDR   | SDS   | NSCR   | P | E  | S   | O   |
> | AIME2024     | 20.00% | 16.67% | 0.833  | 0.000 | 0.000 | -0.167 | 5  | 0  | 1   | 24  |
> | AIME2025   | 23.33%  | 20.00% | 0.571 | 0.333 | 0.421 | -0.111 | 4 | 2  | 3   | 21  |
> | AMC23     | 85.00%  | 77.50% | 0.912  | 0.000 | 0.000 | -0.088 | 31 | 0  | 3   | 6   |
> | MATH500    | 77.80% | 76.20% | 0.920 | 0.060 | 0.113 | -0.019 | 358 | 23 | 31  | 88  |
> | Minerva-Math   | 33.82% | 32.35% | 0.837 | 0.125 | 0.218 | -0.039 | 77  | 11 | 15  | 169 |
> | OlympiadBench | 49.93%| 49.33% | 0.849 | 0.141 | 0.242 | -0.010 | 286 | 47 | 51  | 291 |
> | Aggregate Statistics |  | |   |    |  |   |  |    |     |     |
> | Math Benchmarks   | -    | -      | 0.887   | 0.110 | 0.166 | -0.072 | 761 | 83 | 104 | 599 |
>
> > ## W2, Q1: Insufficient analysis of exceptional cases
>
> Thank you for this insightful question regarding the task-specific nature of expansion.
>
> We would like to clarify that expansion is **not random**, nor does it reflect fundamentally new reasoning capabilities emerging from RLVR. Our analysis reveals that expansion arises only under two well-defined mechanisms, both fully consistent with our empirical results in Tables 1–3 and Figures 2–3:
>
> ### **Mechanism 1: Recombination of pre-existing subskills**
>
> Expansion tasks (e.g., dice, arc\_1d, boxnet, graph\_color\_vertex20) share a crucial property: their solutions can be constructed by combining weakly dependent, low-probability subcomponents that the base model already possesses. For instance, graph coloring requires local per-vertex decisions, arc\_1d involves element-wise sequence transformations, and boxnet demands JSON key-value fragments. While each component appears with low marginal probability under the base model, its correct global combination rarely emerges spontaneously.
>
> RLVR amplifies these long-tail fragments and makes their recombination statistically viable. This explains why base pass@1 is low, base pass@k improves steadily as k grows, and RLVR pushes a few low-density correct completions above the empirical threshold ε. As shown in Figure 3, all "new" correct completions correspond to tail configurations assembled from pre-existing reasoning primitives—**not novel reasoning modes**.
>
> ### **Mechanism 2: Correction of format mismatches**
>
> A second source of expansion occurs when the base model already understands the underlying reasoning but fails to meet the reward format precisely (e.g., JSON structure, ordering constraints). RLVR reshapes the distribution to match the reward surface more faithfully, "unlocking" correct solutions that were previously unrewarded due to formatting issues. The perplexity results in Table 2 support this interpretation: expansion cases show small PPL differences between base and RLVR, indicating that expansions are stylistically aligned variants of reasoning patterns already present, rather than genuinely out-of-support solutions.

---

> ### Author Response · Authors · 2025-11-25
> **Response to Reviewer XMks (Continued)**
>
> ### **Why expansion remains fundamentally limited**
>
> Even when both mechanisms align, expansion stays tightly constrained: NDR ≤ 0.04 across all models (Table 1), NSCR is negative everywhere, no recovered completion has q(y) < 3.66 × 10⁻⁴, and larger models (7B–14B) show even less expansion. Thus, expansion never corresponds to discovering reasoning patterns outside the base model's effective support—it is simply an amplification of low-probability but pre-existing reasoning fragments.
> ***
>
> We sincerely appreciate your highly supportive review and your recognition of our framework as both methodologically rigorous and conceptually transformative. We will incorporate your suggestions on model diversity and exception analysis to strengthen the comprehensiveness of our final manuscript.

---

> > ### Author Response · Authors · 2025-11-27
> > **Thank You and Follow-up**
> >
> > Thank you once again for your valuable time and thoughtful comments on our submission. We have thoroughly addressed each of the points you raised and provided comprehensive responses in our rebuttal. We sincerely hope that our clarifications and additional explanations have satisfactorily resolved your concerns.
> >
> > As the discussion period is approaching its conclusion, we would be deeply grateful if you could kindly share any further thoughts or feedback you may have on our responses. Should there be any remaining questions or aspects that require further clarification, we would be more than happy to provide additional information.
> >
> > We genuinely appreciate your continued engagement and look forward to your reply.

---

### Author Response · Authors · 2025-11-25
**Global Response**

We would like to express our profound gratitude to all reviewers for their valuable insights and constructive feedback on our work. We are particularly grateful that the reviews have highlighted the following shared recognitions:

* (R#XMks, R#hnej, R#APQJ) The **empirical rigor and comprehensiveness** of our experiments, including large-scale sampling, extensive model coverage, and detailed analysis across diverse benchmarks

* (R#XMks, R#hnej, R#APQJ) The **clarity, structure, and presentation quality** of our paper, making it accessible and easy to understand&#x20;

* (R#XMks, R#hnej) The **systematic framework and novel metrics** (SRR, NDR, SDS, NSCR) we introduced for quantifying reasoning space dynamics, which transform abstract discussions into measurable quantities

* (R#XMks, R#2twv) The **timeliness and importance** of our research topic on RLVR's effect on reasoning capabilities

* (R#2twv, R#hnej) The **novel insights** regarding support shrinkage, entropy-reward trade-offs, and the distinction between token-level and answer-level entropy

We appreciate the reviewers' careful consideration and positive feedback on these key aspects of our work.

## Common Concerns:

Several reviewers raised overlapping concerns that we address collectively below:

> **Perplexity analysis in Table 2 and potential style confounds** (R#2twv, R#hnej, R#APQJ)

We revised the perplexity evaluation to eliminate asymmetric bias by using **only external reasoning traces** (DeepSeek-R1 and Claude Sonnet 4). This ensures fair, style-independent comparison. The consistent Base→RLVR perplexity gap across two stylistically divergent references shows that RLVR reduces compatibility with broader reasoning distributions, confirming that the observed shrinkage is not a stylistic artifact.

> **Need for controlled experiments comparing RLVR vs. non-RLVR methods with identical base models and training data** (R#2twv, R#hnej)

We conducted **fully controlled experiments** where the *base model, data, and hyperparameters are identical*, and the **only differing factor is the training objective (SFT vs. RLVR)**. These ablations show that RLVR intrinsically causes support shrinkage and instability, while SFT yields mild, stable expansion. This isolates RLVR as the causal factor and rules out confounding effects of architecture, training duration, or data mixture.

> **Relationship to concurrent work and unique contributions beyond existing literature** (R#APQJ, R#hnej)

While several concurrent works explore RL-based reasoning, our study addresses a **fundamentally different and previously unexamined question**: how RLVR reshapes the underlying *reasoning support distribution*. Prior work focuses on final accuracy metrics and qualitative trends, but does **not** formalize empirical support, quantify support changes, or explain why RLVR rarely expands reasoning. Our paper provides the **first systematic framework** for understanding these dynamics: (i) a definition of empirical support under finite sampling, (ii) four new support-dynamics metrics (SRR, NDR, SDS, NSCR), (iii) perplexity- and entropy-based diagnostics, **and (iv) a temporal analysis of support dynamics across RLVR training checkpoints**, revealing that RLVR steadily contracts the accessible solution set even when early-stage performance rises. Combined with **broad cross-family evaluation** and **controlled SFT vs. RLVR comparisons**, our work uniquely shows that current RLVR recipes behave as support-bound optimizers that sharpen known trajectories rather than discovering new reasoning modes, which provide conceptual and methodological advances beyond concurrent literature.



We have carefully addressed these concerns with additional experiments, clarifications, and theoretical justifications in the revised manuscript.

## Individual Responses:

We provide detailed point-by-point responses to each reviewer's specific questions and concerns in the individual response sections below.

***
Authors of Submission 6208

---

### Author Response · Authors · 2025-12-02
**Summary of the Rebuttal**

Dear ACs, SACs, and PCs,

We sincerely thank all reviewers for their thoughtful evaluation and the ACs, SACs, and PCs for the substantial time and effort devoted to this submission. Here, we provide a concise final response that summarizes our rebuttal and highlights the key clarifications.

Overall, multiple reviewers acknowledge the importance of studying RL for verifiable rewards (RLVR), as well as our unified support-dynamics framework with four new metrics and the large-scale empirical and theoretical analysis across multiple benchmarks and systems. **Reviewer R#XMks assigned a score of 10**. In contrast, the two score-4 reviewers (R#2twv, R#hnej) raised two main concerns: a potential style/format confound in the perplexity comparison of Table 2, and the need for more controlled ablations to disentangle the effects of different RL algorithms, hyperparameters, and training data.


In the rebuttal, we made targeted additions to address two score-4 reviewers' (R#2twv, R#hnej) concerns. First, we introduced a style-controlled perplexity analysis in which output style and formatting are explicitly matched. The results show that the main trends persist even after controlling for style, ruling out a purely style-driven explanation. Second, within the same framework, we conducted fully controlled RLVR runs that keep the base model, data, algorithm, and hyperparameters fixed and only vary the RLVR optimization or training objective (RLVR versus SFT). These results show that support shrinkage and the entropy–reward trade-off persist even under this controlled setting, confirming that they are intrinsic to the RLVR objective rather than to particular implementation choices. **R#hnej explicitly stated that these additions resolved the main concerns and, accordingly, increased the score, and we believe these changes also directly address the core issues raised by the other score-4 reviewer**.



By contrast, the score-2 review (R#APQJ) is primarily based on two claims: that our work “essentially duplicates” Yue et al. (2025), and that “Support of Correct Completions” is not clearly connected to reasoning performance. **We must respectfully note that this review appears to be based on superficial engagement with our work.** These judgments reflect key misunderstandings of the main conceptual, empirical, and theoretical contributions of the paper. Yue et al. pose an important high-level question about whether RLVR truly expands model capabilities. Our paper extends this line of work in three folds that are not present in Yue et al.:

> (1) We introduce a unified, quantifiable support-dynamics framework with new metrics that decompose retention, discovery, and loss of correct completions;

> (2) We conduct an unprecedented large-scale unified evaluation across 11 benchmarks and 6 RLVR system families (including strong variants such as ProRL, DAPO, Skywork, AceReason, Phi-4-Reason, and Nemotron) as well as multimodal models, requiring hundreds of thousands of GPU hours;

> (3) We provide a formal “invisible leash” analysis in Appendix C that proves KL-regularized RLVR may not be able to assign positive probability to completions outside the base model’s support.&#x20;

It's important to note that, following Yue et al., several subsequent works (such as ProRL and AceReason) have claimed to “break” RLVR boundaries, and our unified support-dynamics view together with these experiments systematically re-examines those expansion claims and shows that the apparent gains often arise from probability mass redistribution within the base model’s long tail rather than genuine access to new solution regions. In addition, Sections 3 and 4 of our paper analyze how support dynamics at the distribution level mechanistically induce the trade-off between improved pass@1 and degraded pass@k, and in our point-by-point rebuttal to R#APQJ we gave detailed answers to explicit questions about the connection between Support of Correct Completions and reasoning performance and about its difference from pass@k, including additional derivations and experiments that clarify Support as a structural property of the model distribution and pass@k as an outcome metric.&#x20;



*Considering the time window of the discussion period, three of the four reviewers did not respond further, so these clarifications are not reflected in the final written review. We therefore kindly request the AC, SAC, and PC to take our detailed rebuttal into account when interpreting this review. Given the paper’s overall conceptual, empirical, and theoretical contributions, all of our authors believe it can provide useful insights into the limits and potential of RLVR.*

---

### Meta-Review · Area_Chair_5dxQ · 2026-01-07

**Summary:**

This paper presents a large-scale empirical and analytical study of Reinforcement Learning with Verifiable Rewards (RLVR), arguing that current RLVR recipes primarily sharpen probability mass over existing solutions rather than genuinely expanding a model’s reasoning support. Reviewers broadly agree that the topic is timely and important, and several found the empirical scope, metrics, and framing insightful. However, reviewers' opinions varied drastically, ranging from strong accept to reject. The primary factors influencing the decision are concerns about novelty relative to recent closely related work, the interpretation and strength of the central claims, and whether the contribution should be viewed as a foundational insight or primarily as an incremental consolidation and systematization of emerging observations in the literature. While the rebuttal addressed several technical concerns and strengthened the empirical case, disagreement remains about whether the paper delivers sufficiently new conceptual and/or methodological advances to meet the acceptance bar for ICLR.

**Reviewer Concerns:**

Concerns fully/largely addressed by the rebuttal:
* Style and perplexity confounds (Table 2): The authors added a revised perplexity analysis using external reference traces from independent models, which directly addressed concerns about asymmetric or style-driven comparisons (raised by reviewers 2twv and hnej).
* Causal attribution to RLVR vs. confounders: Additional controlled experiments fixing base model, data, and hyperparameters while varying the training objective (SFT vs. RLVR) substantially strengthened the claim that support shrinkage is intrinsic to RLVR (2twv, hnej).
* Training dynamics and ε-sensitivity: The authors added training-trajectory analyses, ε-sweeps, confidence intervals, and significance tests, resolving methodological concerns about metric robustness (hnej).
* Empirical breadth: The rebuttal expanded coverage to additional models and benchmarks, reinforcing the consistency of observed trends across settings.

Concerns that remain outstanding or were only partially resolved:
* Novelty relative to recent work: Multiple reviewers (notably APQJ and 2twv) argue that the central conclusion (namely, that RLVR often sharpens existing solutions rather than discovering new ones) has already appeared in recent literature, and that the paper does not fully convince them that the proposed support-dynamics framework constitutes a sufficiently distinct conceptual advance rather than a formalization and large-scale confirmation of known effects.
* Interpretation of “support expansion” vs. reasoning capability: Some reviewers remain unconvinced that the introduced metrics (SRR, NDR, SDS, NSCR) provide fundamentally new insight beyond existing pass@k-style analyses, rather than a more detailed decomposition of similar phenomena.
* Positioning as analysis vs. innovation: While several reviewers appreciated the depth of analysis, others felt the paper lacks a clear algorithmic or methodological contribution and were less persuaded that an analysis-only contribution of this form is sufficient for acceptance at ICLR.

**Reviewer Scores:**

* Reviewer XMks: likely to remain at 10. This reviewer was strongly positive and would likely remain at strong accept after discussion; their enthusiasm was not contingent on the contested novelty points. *However*, I am down-weighting this reviewer's score because the written review is not commensurate with the provided score (10) and the reviewer seems inexperienced or unfamiliar with the ICLR score standards.
* Reviewer hnej: likely to increase to 6. Explicitly stated that the rebuttal resolved most concerns and indicated an intention to raise the score, likely to weak accept. Maybe (but unlikely) to 8 given their initial standings.
* Reviewer 2twv: likely to remain at 4. The controlled experiments and revised perplexity analysis address major objections, but lingering doubts about causal interpretation and contribution scope likely keep the score near borderline.
* Reviewer APQJ): likely to remain at 2. Despite extensive rebuttal, this reviewer’s core concern about overlap with prior work and the necessity of the proposed metrics appears only partially addressed; a substantial score increase is unlikely.

Overall:  After accounting for likely score changes, the paper would still have high score variance, with at least one reviewer remaining clearly below threshold and others only marginally above it. Given the remaining uncertainty about novelty and contribution type, I do not see a clear consensus emerging in favor of acceptance.

---

### Decision · Program_Chairs · 2026-01-26

Reject